# FEDERATED NATURAL POLICY GRADIENT METHODS FOR MULTI-TASK REINFORCEMENT LEARNING

## ABSTRACT

Federated reinforcement learning (RL) enables collaborative decision making of multiple distributed agents without sharing local data trajectories. In this work, we consider a multi-task setting, in which each agent has its own private reward function corresponding to different tasks, while sharing the same transition kernel of the environment. Focusing on infinite-horizon tabular Markov decision processes, the goal is to learn a globally optimal policy that maximizes the sum of the discounted total rewards of all the agents in a decentralized manner, where each agent only communicates with its neighbors over some prescribed graph topology.

We develop federated vanilla and entropy-regularized natural policy gradient (NPG) methods under softmax parameterization, where gradient tracking is applied to the global Q-function to mitigate the impact of imperfect information sharing. We establish non-asymptotic global convergence guarantees under exact policy evaluation, which are nearly independent of the size of the state-action space and illuminate the impacts of network size and connectivity. To the best of our knowledge, this is the first time that global convergence is established for federated multi-task RL using policy optimization. Moreover, the convergence behavior of the proposed algorithms is robust against inexactness of policy evaluation.

## 1 INTRODUCTION

Federated reinforcement learning (FRL) is an emerging paradigm that combines the advantages of federated learning (FL) and reinforcement learning (RL) (Qi et al., 2021; Zhuo et al., 2019), allowing multiple agents to learn a shared policy from local experiences, without exposing their private data to a central server nor other agents. FRL is poised to enable collaborative and efficient decision making in scenarios where data is distributed, heterogeneous, and sensitive, which arise frequently in applications such as edge computing, smart cities, and healthcare (Wang et al., 2023; 2020; Zhuo et al., 2019), to name just a few. As has been observed (Lian et al., 2017), decentralized training can lead to performance improvements in FL by avoiding communication congestions at busy nodes such as the server, especially under high-latency scenarios. This motivates us to design algorithms for the fully decentralized setting, a scenario where the agents can only communicate with their local neighbors over a prescribed network topology.

In this work, we study the problem of *federated multi-task reinforcement learning* (Anwar & Ray-chowdhury, 2021; Qi et al., 2021; Yu et al., 2020), where each agent collects its own reward — possibly unknown to other agents — corresponding to the local task at hand, while having access to the same dynamics (i.e., transition kernel) of the environment. The collective goal is to learn a shared policy that maximizes the total rewards accumulated from all the agents; in other words, one seeks a policy that performs well in terms of overall benefits, rather than biasing towards any individual task, achieving the Pareto frontier in a multi-objective context. There is no shortage of application scenarios where federated multi-task RL becomes highly relevant. For instance, in healthcare (Zerka et al., 2020), different hospitals may be interested in finding an optimal treatment for all patients without disclosing private data, where the effectiveness of the treatment can vary across different hospitals due to demographical differences. As another potential application, to enhance ChatGPT's performance across different tasks or domains (M Alshater, 2022; Rahman et al., 2023), one might consult domain experts to chat and rate ChatGPT's outputs for solving different tasks, and train ChatGPT in a federated manner without exposing private data or feedback of each expert.

Nonetheless, despite the promise, provably efficient algorithms for federated multi-task RL remain substantially under-explored, especially in the fully decentralized setting. The heterogeneity of local tasks leads to a higher degree of disagreements between the global value function and local value functions of individual agents. Due to the lack of global information sharing, care needs to be taken to judiciously balance the use of neighboring information (to facilitate consensus) and local data (to facilitate learning) when updating the policy. To the best of our knowledge, no algorithms are currently available to find the global optimal policy with non-asymptotic convergence guarantees even for tabular infinite-horizon Markov decision processes.

Motivated by the connection with decentralized optimization, it is tempting to take a policy optimization perspective to tackle this challenge. Policy gradient (PG) methods, which seek to learn the policy of interest via first-order optimization methods, play an eminent role in RL due to their simplicity and scalability. In particular, natural policy gradient (NPG) methods (Amari, 1998; Kakade, 2001) are among the most popular variants of PG methods, underpinning default methods used in practice such as trust region policy optimization (TRPO) (Schulman et al., 2015) and proximal policy optimization (PPO) (Schulman et al., 2017). On the theoretical side, it has also been established recently that the NPG algorithm enjoys fast global convergence to the optimal policy in an almost dimension-free manner (Agarwal et al., 2021; Cen et al., 2021), where the iteration complexity is nearly independent of the size of the state-action space. Inspired by the efficacy of NPG methods, it is natural to ask:

> *Can we develop **federated** variants of NPG methods that are easy to implement in the fully decentralized setting with **non-asymptotic global convergence** guarantees for multi-task RL?*

## 1.1 OUR CONTRIBUTIONS

Focusing on infinite-horizon Markov decision processes (MDPs), we provide an affirmative answer to the above question, by developing federated NPG (FedNPG) methods for solving both the vanilla and entropy-regularized multi-task RL problems with finite-time global convergence guarantees. While entropy regularization is often incorporated as an effective strategy to encourage exploration during policy learning, solving the entropy-regularized RL problem is of interest in its own right, as the optimal regularized policy possesses desirable robust properties with respect to reward perturbations (Eysenbach & Levine, 2021; McKelvey & Palfrey, 1995).

Due to the multiplicative update nature of NPG methods under softmax parameterization, it is more convenient to work with the logarithms of local policies in the decentralized setting. In each iteration of the proposed FedNPG method, the logarithms of local policies are updated by a weighted linear combination of two terms (up to normalization): a gossip mixing (Nedic & Ozdaglar, 2009) of the logarithms of neighboring local policies, and a local estimate of the global Q-function tracked via the technique of dynamic average consensus (Zhu & Martínez, 2010), a prevalent idea in decentralized optimization that allows for the use of large constant learning rates (Di Lorenzo & Scutari, 2016; Nedic et al., 2017; Qu & Li, 2017) to accelerate convergence. Our contributions are as follows.

- We propose FedNPG methods for both the vanilla and entropy-regularized multi-task RL problems, where each agent only communicates with its neighbors and performs local computation using its own reward or task information.

- Assuming access to exact policy evaluation, we establish that the average iterate of vanilla FedNPG converges globally at a rate of $\mathcal{O}(1/T^{2/3})$ in terms of the sub-optimality gap for the multi-task RL problem, and that the last iterate of entropy-regularized FedNPG converges globally at a linear rate to the regularized optimal policy. Our convergence theory highlights the impacts of all salient problem parameters (see Table 1 for details), such as the size and connectivity of the communication network. In particular, the iteration complexities of FedNPG are again almost independent of the size of the state-action space, which recover prior results on the centralized NPG methods when the network is fully connected.

- We further demonstrate the stability of the proposed FedNPG methods when policy evaluations are only available in an inexact manner. To be specific, we prove that their convergence rates remain unchanged as long as the approximation errors are sufficiently small in the $\ell_\infty$ sense.

To the best of our knowledge, the proposed federated NPG methods are the first policy optimization methods for multi-task RL that achieve explicit non-asymptotic global convergence guarantees, allowing for fully decentralized communication without any need to share local reward/task information.

### 1.2 RELATED WORK

**Global convergence of NPG methods for tabular MDPs.** Agarwal et al. (2021) first establishes a $\mathcal{O}(1/T)$ last-iterate convergence rate of the NPG method under softmax parameterization with constant step size, assuming access to exact policy evaluation. When entropy regularization is in place, Cen et al. (2021) establishes a global linear convergence to the optimal regularized policy for the entire range of admissible constant learning rates using softmax parameterization and exact policy evaluation, which is further shown to be stable in the presence of $\ell_\infty$ policy evaluation errors. The iteration complexity of NPG methods is nearly independent with the size of the state-action space, which is in sharp contrast to softmax policy gradient methods that may take exponential time to converge (Li et al., 2023; Mei et al., 2020). Lan (2023) proposed a more general framework through the lens of mirror descent for regularized RL with global linear convergence guarantees, which is further generalized in Zhan et al. (2023); Lan et al. (2023). Earlier analysis of regularized MDPs can be found in Shani et al. (2020). Besides, Xiao (2022) proves that vanilla NPG also achieves linear convergence when geometrically increasing learning rates are used; see also Khodadadian et al. (2021); Bhandari & Russo (2021). Zhou et al. (2022) developed an anchor-changing NPG method for multi-task RL under various optimality criteria in the centralized setting.

**Distributed and federated RL.** There have been a variety of settings being set forth for distributed and federated RL. Mnih et al. (2016); Espeholt et al. (2018); Assran et al. (2019); Khodadadian et al. (2022); Woo et al. (2023) focused on developing federated versions of RL algorithms to accelerate training, assuming all agents share the same transition kernel and reward function; in particular, Khodadadian et al. (2022); Woo et al. (2023) established the provable benefits of federated learning in terms of linear speedup. More pertinent to our work, Zhao et al. (2023); Anwar & Raychowdhury (2021) considered the federated multi-task framework, allowing different agents having private reward functions. Zhao et al. (2023) proposed an empirically probabilistic algorithm that can seek an optimal policy under the server-client setting, while Anwar & Raychowdhury (2021) developed new attack methods in the presence of adversarial agents. Different from the FRL framework, Chen et al. (2021; 2022b); Omidshafiei et al. (2017); Kar et al. (2012); Chen et al. (2022a); Zeng et al. (2021) considered the distributed multi-agent RL setting where the agents interact with a dynamic environment through a multi-agent Markov decision process, where each agent can have their own state or action spaces. Zeng et al. (2021) developed a decentralized policy gradient method where different agents have different MDPs.

**Decentralized first-order optimization algorithms.** Early work of consensus-based first-order optimization algorithms for the fully decentralized setting include but are not limited to Lobel & Ozdaglar (2008); Nedic & Ozdaglar (2009); Duchi et al. (2011). Gradient tracking, which leverages the idea of dynamic average consensus (Zhu & Martínez, 2010) to track the gradient of the global objective function, is a popular method to improve the convergence speed (Qu & Li, 2017; Nedic et al., 2017; Di Lorenzo & Scutari, 2016; Pu & Nedić, 2021; Li et al., 2020a).

**Notation.** Boldface small and capital letters denote vectors and matrices, respectively. Sets are denoted with curly capital letters, e.g., $\mathcal{S}, \mathcal{A}$. We let $(\mathbb{R}^d, \|\cdot\|)$ denote the $d$-dimensional real coordinate space equipped with norm $\|\cdot\|$. The $\ell_p$-norm of $\boldsymbol{v}$ is denoted by $\|\boldsymbol{v}\|_p$, where $1 \leq p \leq \infty$, and the spectral norm of a matrix $\boldsymbol{M}$ is denoted by $\|\boldsymbol{M}\|_2$. We let $[N]$ denote $\{1, \ldots, N\}$, use $\mathbf{1}$ to represent the all-one vector, and denote by $\mathbf{0}$ a vector or a matrix consisting of all 0's. We allow the application of functions such as $\log(\cdot)$ and $\exp(\cdot)$ to vectors or matrices, with the understanding that they are applied in an element-wise manner.

## 2 MODEL AND BACKGROUNDS

### 2.1 MARKOV DECISION PROCESSES

**Markov decision processes.** We consider an infinite-horizon discounted Markov decision process (MDP) denoted by $\mathcal{M} = (\mathcal{S}, \mathcal{A}, P, r, \gamma)$, where $\mathcal{S}$ and $\mathcal{A}$ denote the state space and the action space, respectively, $\gamma \in [0, 1)$ indicates the discount factor, $P : \mathcal{S} \times \mathcal{A} \to \Delta(\mathcal{S})$ is the transition kernel, and $r : \mathcal{S} \times \mathcal{A} \to [0, 1]$ stands for the reward function. To be more specific, for each state-action pair $(s, a) \in \mathcal{S} \times \mathcal{A}$ and any state $s' \in \mathcal{S}$, we denote by $P(s'|s, a)$ the transition probability from state $s$ to state $s'$ when action $a$ is taken, and $r(s, a)$ the instantaneous reward received in state $s$ when

| setting | algorithms | iteration complexity | optimality criteria |
|---|---|---|---|
| unregularized | NPG (Agarwal et al., 2021) | $\mathcal{O}\left(\frac{1}{(1-\gamma)^2\varepsilon}\right)$ | $V^\star - V^{\pi^{(t)}} \leq \varepsilon$ |
| | FedNPG (ours) | $\mathcal{O}\left(\frac{\sigma\sqrt{N}\log|\mathcal{A}|}{(1-\gamma)^{\frac{9}{2}}(1-\sigma)\varepsilon^{\frac{3}{2}}} + \frac{1}{(1-\gamma)^2\varepsilon}\right)$ | $\frac{1}{T}\sum_{t=0}^{T-1}\left(V^\star - V^{\bar{\pi}^{(t)}}\right) \leq \varepsilon$ |
| regularized | NPG (Cen et al., 2021) | $\mathcal{O}\left(\frac{1}{\tau\eta}\log\left(\frac{1}{\varepsilon}\right)\right)$ | $V_\tau^\star - V_\tau^{\pi^{(t)}} \leq \varepsilon$ |
| | FedNPG (ours) | $\mathcal{O}\left(\max\left\{\frac{1}{\tau\eta}, \frac{1}{1-\sigma}\right\}\log\left(\frac{1}{\varepsilon}\right)\right)$ | $V_\tau^\star - V_\tau^{\bar{\pi}^{(t)}} \leq \varepsilon$ |

Table 1: Iteration complexities of NPG and FedNPG (ours) methods to reach $\varepsilon$-accuracy of the vanilla and entropy-regularized problems, where we assume exact gradient evaluation, and only keep the dominant terms w.r.t. $\varepsilon$. The policy estimates in the $t$-iteration are $\pi^{(t)}$ and $\bar{\pi}^{(t)}$ for NPG and FedNPG, respectively, where $T$ is the number of iterations. Here, $N$ is the number of agents, $\tau \leq 1$ is the regularization parameter, $\sigma \in [0, 1]$ is the spectral radius of the network, $\gamma \in [0, 1)$ is the discount factor, $|\mathcal{A}|$ is the size of the action space, and $\eta > 0$ is the learning rate. For vanilla FedNPG, the learning rate is set as $\eta = \eta_1 = \mathcal{O}\left(\left(\frac{(1-\gamma)^9(1-\sigma)^2\log|\mathcal{A}|}{TN\sigma}\right)^{1/3}\right)$; for entropy-regularized FedNPG, the learning rate satisfies $0 < \eta < \eta_0 = \mathcal{O}\left(\frac{(1-\gamma)^7(1-\sigma)^2\tau}{\sigma N}\right)$. The iteration complexities of FedNPG reduce to their centralized counterparts when $\sigma = 0$.

action $a$ is taken. Furthermore, a policy $\pi : \mathcal{S} \to \Delta(\mathcal{A})$ specifies an action selection rule, where $\pi(a|s)$ specifies the probability of taking action $a$ in state $s$ for each $(s, a) \in \mathcal{S} \times \mathcal{A}$.

For any given policy $\pi$, we denote by $V^\pi : \mathcal{S} \mapsto \mathbb{R}$ the corresponding value function, which is the expected discounted cumulative reward with an initial state $s_0 = s$, given by

$$\forall s \in \mathcal{S}: \quad V^\pi(s) := \mathbb{E}\left[\sum_{t=0}^\infty \gamma^t r(s_t, a_t)|s_0 = s\right], \tag{1}$$

where the randomness is over the trajectory generated following the policy $a_t \sim \pi(\cdot|s_t)$ and the MDP dynamic $s_{t+1} \sim P(\cdot|s_t, a_t)$. We also overload the notation $V^\pi(\rho)$ to indicate the expected value function of policy $\pi$ when the initial state follows a distribution $\rho$ over $\mathcal{S}$, namely, $V^\pi(\rho) := \mathbb{E}_{s\sim\rho}[V^\pi(s)]$. Similarly, the Q-function $Q^\pi : \mathcal{S} \times \mathcal{A} \mapsto \mathbb{R}$ of policy $\pi$ is defined by

$$\forall (s, a) \in \mathcal{S} \times \mathcal{A}: \quad Q^\pi(s, a) := \mathbb{E}\left[\sum_{t=0}^\infty \gamma^t r(s_t, a_t)|s_0 = s, a_0 = a\right], \tag{2}$$

which measures the expected discounted cumulative reward with an initial state $s_0 = s$ and an initial action $a_0 = a$, with expectation taken over the randomness of the trajectory. The optimal policy $\pi^\star$ refers to the policy that maximizes the value function $V^\pi(s)$ for all states $s \in \mathcal{S}$, which is guaranteed to exist (Puterman, 2014). The corresponding optimal value function and Q-function are denoted as $V^\star$ and $Q^\star$, respectively.

## 2.2 Entropy-regularized RL

Entropy regularization (Williams & Peng, 1991; Ahmed et al., 2019) is a popular technique in practice that encourages stochasticity of the policy to promote exploration, as well as robustness against reward uncertainties. Mathematically, this can be viewed as adjusting the instantaneous reward based the current policy in use as

$$\forall (s, a) \in \mathcal{S} \times \mathcal{A}: \quad r_\tau(s, a) := r(s, a) - \tau\log\pi(a|s), \tag{3}$$

where $\tau \geq 0$ denotes the regularization parameter. Typically, $\tau$ should not be too large to outweigh the actual rewards; for ease of presentation, we assume $\tau \leq 1$. Equivalently, this amounts to the entropy-regularized (also known as "soft") value function, defined as

$$\forall s \in \mathcal{S}: \quad V_\tau^\pi(s) := V^\pi(s) + \tau\mathcal{H}(s, \pi). \tag{4}$$

Here, we define

$$\mathcal{H}(s, \pi) := \mathbb{E}\left[\sum_{t=0}^\infty -\gamma^t\log\pi(a_t|s_t)|s_0 = s\right] = \frac{1}{1-\gamma}\mathbb{E}_{s'\sim d_s^\pi}\left[-\sum_{a\in\mathcal{A}}\pi(a|s')\log\pi(a|s')\right], \tag{5}$$

where $d_{s_0}^\pi$ is the discounted state visitation distribution of policy $\pi$ given an initial state $s_0 \in \mathcal{S}$, denoted by

$$\forall s \in \mathcal{S}: \quad d_{s_0}^\pi(s) := (1 - \gamma) \sum_{t=0}^\infty \gamma^t \mathbb{P}(s_t = s | s_0), \tag{6}$$

with the trajectory generated by following policy $\pi$ in the MDP $\mathcal{M}$ starting from state $s_0$. Analogously, the regularized (or soft) Q-function $Q_\tau^\pi$ of policy $\pi$ is related to the soft value function $V_\tau^\pi(s)$ as

$$\forall (s,a) \in \mathcal{S} \times \mathcal{A}: \quad Q_\tau^\pi(s,a) = r(s,a) + \gamma \mathbb{E}_{s' \in P(\cdot|s,a)} \left[ V_\tau^\pi(s') \right], \tag{7a}$$

$$\forall s \in \mathcal{S}: \quad V_\tau^\pi(s) = \mathbb{E}_{a \sim \pi(\cdot|s)} \left[ -\tau \pi(a|s) + Q_\tau^\pi(s,a) \right]. \tag{7b}$$

The optimal regularized policy, the optimal regularized value function, and the Q-function are denoted by $\pi_\tau^\star$, $V_\tau^\star$, and $Q_\tau^\star$, respectively.

## 2.3 NATURAL POLICY GRADIENT METHODS

Natural policy gradient (NPG) methods lie at the heart of policy optimization, serving as the backbone of popular heuristics such as TRPO (Schulman et al., 2015) and PPO (Schulman et al., 2017). Instead of directly optimizing the policy over the probability simplex, one often adopts the softmax parameterization, which parameterizes the policy as

$$\pi_\theta := \text{softmax}(\theta) \quad \text{or} \quad \forall (s,a) \in \mathcal{S} \times \mathcal{A}: \quad \pi_\theta(a|s) := \frac{\exp \theta(s,a)}{\sum_{a' \in \mathcal{A}} \exp \theta(s,a')} \tag{8}$$

for any $\theta \colon \mathcal{S} \times \mathcal{A} \to \mathbb{R}$.

**Vanilla NPG method.** In the tabular setting, the update rule of vanilla NPG at the $t$-th iteration can be concisely represented as

$$\forall (s,a) \in \mathcal{S} \times \mathcal{A}: \quad \pi^{(t+1)}(a|s) \propto \pi^{(t)}(a|s) \exp \left( \frac{\eta Q^{(t)}(s,a)}{1 - \gamma} \right), \tag{9}$$

where $\eta > 0$ denotes the learning rate, and $Q^{(t)} = Q^{\pi^{(t)}}$ is the Q-function under policy $\pi^{(t)}$. Agarwal et al. (2021) shows that: in order to find an $\varepsilon$-optimal policy, NPG takes at most $\mathcal{O}\left( \frac{1}{(1-\gamma)^2 \varepsilon} \right)$ iterations, assuming exact policy evaluation.

**Entropy-regularized NPG method.** Turning to the regularized problem, we note that the update rule of entropy-regularized NPG becomes

$$\forall (s,a) \in \mathcal{S} \times \mathcal{A}: \quad \pi^{(t+1)}(a|s) \propto (\pi^{(t)}(a|s))^{1 - \frac{\eta \tau}{1-\gamma}} \exp \left( \frac{\eta Q_\tau^{(t)}(s,a)}{1 - \gamma} \right), \tag{10}$$

where $\eta \in (0, \frac{1-\gamma}{\tau}]$ is the learning rate, and $Q_\tau^{(t)} = Q_\tau^{\pi^{(t)}}$ is the soft Q-function of policy $\pi^{(t)}$. Cen et al. (2022) proves that entropy-regularized NPG enjoys fast global linear convergence to the optimal regularized policy: to find an $\varepsilon$-optimal regularized policy, entropy-regularized NPG takes no more than $\mathcal{O}\left( \frac{1}{\eta \tau} \log \left( \frac{1}{\varepsilon} \right) \right)$ iterations.

# 3 FEDERATED NPG METHODS FOR MULTI-TASK RL

## 3.1 FEDERATED MULTI-TASK RL

In this paper, we consider the federated multi-task RL setting, where a set of agents learn collaboratively a single policy that maximizes its average performance over all the tasks using only local computation and communication.

**Multi-task RL.** Each agent $n \in [N]$ has its own private reward function $r_n(s,a)$ — corresponding to different tasks — while sharing the same transition kernel of the environment. The goal is to collectively learn a single policy $\pi$ that maximizes the global value function given by

$$V^\pi(s) = \frac{1}{N} \sum_{n=1}^N V_n^\pi(s), \tag{11}$$

where $V_n^\pi$ is the value function of agent $n \in [N]$, defined by

$$\forall s \in \mathcal{S}: \quad V_n^\pi(s) := \mathbb{E}\left[\sum_{t=0}^\infty \gamma^t r_n(s_t, a_t) | s_0 = s\right]. \tag{12}$$

Clearly, the global value function (11) corresponds to using the average reward of all agents

$$r(s, a) = \frac{1}{N} \sum_{n=1}^N r_n(s, a). \tag{13}$$

The global Q-function $Q^\pi(s, a)$ and the agent Q-functions $Q_n^\pi(s, a)$ can be defined in a similar manner obeying $Q^\pi(s, a) = \frac{1}{N} \sum_{n=1}^N Q_n^\pi(s, a)$.

In parallel, we are interested in the entropy-regularized setting, where each agent $n \in [N]$ is equipped with a regularized reward function given by

$$r_{\tau,n}(s, a) := r_n(s, a) - \tau \log \pi(a|s), \tag{14}$$

and we define similarly the regularized value function and the global regularized value function as

$$\forall s \in \mathcal{S}: \quad V_{\tau,n}^\pi(s) := \mathbb{E}\left[\sum_{t=0}^\infty \gamma^t r_{\tau,n}(s_t, a_t) | s_0 = s\right], \quad \text{and} \quad V_\tau^\pi(s) = \frac{1}{N} \sum_{n=1}^N V_{\tau,n}^\pi(s). \tag{15}$$

The soft Q-function of agent $n$ is given by

$$Q_{\tau,n}^\pi(s, a) = r_n(s, a) + \gamma \mathbb{E}_{s' \in P(\cdot|s,a)}\left[V_{\tau,n}^\pi(s')\right], \tag{16}$$

and the global soft Q-function is given by $Q_\tau^\pi(s, a) = \frac{1}{N} \sum_{n=1}^N Q_{\tau,n}^\pi(s, a)$.

**Federated policy optimization in the fully decentralized setting.** We consider a federated setting with fully decentralized communication, that is, all the agents are synchronized to perform information exchange over some prescribed network topology denoted by an connected and undirected weighted graph $\mathcal{G}([N], E)$. Here, $E$ stands for the edge set of the graph with $N$ nodes — each corresponding to an agent — and two agents can communicate with each other if and only if there is an edge connecting them. The information sharing over the graph is best described by a mixing matrix (Nedic & Ozdaglar, 2009), denoted by $\boldsymbol{W} = [w_{ij}] \in [0, 1]^{N \times N}$, where $w_{ij}$ is a positive number if $(i, j) \in E$ and 0 otherwise. We also make the following standard assumptions on the mixing matrix.

**Assumption 1** (double stochasticity). *The mixing matrix $\boldsymbol{W} = [w_{ij}] \in [0, 1]^{N \times N}$ is symmetric (i.e., $\boldsymbol{W}^\top = \boldsymbol{W}$) and doubly stochastic (i.e., $\boldsymbol{W}\mathbf{1} = \mathbf{1}$, $\mathbf{1}^\top \boldsymbol{W} = \mathbf{1}^\top$).*

The following standard metric measures how fast information propagates over the graph.

**Definition 1** (spectral radius). *The spectral radius of $\boldsymbol{W}$ is defined as*

$$\sigma := \left\|\boldsymbol{W} - \frac{1}{N}\mathbf{1}_N\mathbf{1}_N^\top\right\|_2 \in [0, 1). \tag{17}$$

An immediate consequence is that for any $\boldsymbol{x} \in \mathbb{R}^N$, letting $\overline{x} = \frac{1}{N}\mathbf{1}_N^\top\boldsymbol{x}$ be its average, we have

$$\|\boldsymbol{W}\boldsymbol{x} - \overline{x}\mathbf{1}_N\|_2 \leq \sigma \|\boldsymbol{x} - \overline{x}\mathbf{1}_N\|_2, \tag{18}$$

where the consensus error contracts by a factor of $\sigma$.

### 3.2 PROPOSED FEDERATED NPG ALGORITHMS

Assuming softmax parameterization, the problem can be formulated as decentralized optimization,

$$\text{(unregularized)} \quad \max_\theta V^{\pi_\theta}(s) = \frac{1}{N} \sum_{n=1}^N V_n^{\pi_\theta}(s), \tag{19}$$

$$\text{(regularized)} \quad \max_\theta V_\tau^{\pi_\theta}(s) = \frac{1}{N} \sum_{n=1}^N V_{\tau,n}^{\pi_\theta}(s), \tag{20}$$

---

**Algorithm 1** Federated NPG (FedNPG)

---

1: **Input:** learning rate $\eta > 0$, iteration number $T \in \mathbb{N}_+$, mixing matrix $\boldsymbol{W} \in \mathbb{R}^{N \times N}$.
2: **Initialize:** $\boldsymbol{\pi}^{(0)}, \boldsymbol{T}^{(0)} = \boldsymbol{Q}^{(0)}$.
3: **for** $t = 0, 1, \cdots T - 1$ **do**
4:     Update the policy for each $(s, a) \in \mathcal{S} \times \mathcal{A}$:

$$\log \boldsymbol{\pi}^{(t+1)}(a|s) = \boldsymbol{W} \left( \log \boldsymbol{\pi}^{(t)}(a|s) + \frac{\eta}{1-\gamma} \boldsymbol{T}^{(t)}(s, a) \right) - \log \boldsymbol{z}^{(t)}(s), \qquad (U_\pi^0)$$

    where $\boldsymbol{z}^{(t)}(s) = \sum_{a' \in \mathcal{A}} \exp \left\{ \boldsymbol{W} \left( \log \boldsymbol{\pi}^{(t)}(a'|s) + \frac{\eta}{1-\gamma} \boldsymbol{T}^{(t)}(s, a') \right) \right\}$.
5:     Evaluate $\boldsymbol{Q}^{(t+1)}$.
6:     Update the global Q-function estimate for each $(s, a) \in \mathcal{S} \times \mathcal{A}$:

$$\boldsymbol{T}^{(t+1)}(s, a) = \boldsymbol{W} \Big( \boldsymbol{T}^{(t)}(s, a) + \underbrace{\boldsymbol{Q}^{(t+1)}(s, a) - \boldsymbol{Q}^{(t)}(s, a)}_{\text{Q-tracking}} \Big). \qquad (U_T^0)$$

7: **end for**

---

where $\pi_\theta := \mathrm{softmax}(\theta)$ subject to communication constraints. Motivated by the success of NPG methods, we aim to develop federated NPG methods to achieve our goal. For notational convenience, let $\boldsymbol{\pi}^{(t)} := \big(\pi_1^{(t)}, \cdots, \pi_N^{(t)}\big)^\top$ be the collection of policy estimates at all agents in the $t$-th iteration. Let

$$\overline{\pi}^{(t)} := \mathrm{softmax} \left( \frac{1}{N} \sum_{n=1}^N \log \pi_n^{(t)} \right), \qquad (21)$$

which satisfies that $\overline{\pi}^{(t)}(a|s) \propto \big( \prod_{n=1}^N \pi_n^{(t)}(a|s) \big)^{1/N}$ for each $(s, a) \in \mathcal{S} \times \mathcal{A}$. Therefore, $\overline{\pi}^{(t)}$ could be seen as the normalized geometric mean of $\{\pi_n^{(t)}\}_{n \in [N]}$. Define the collection of Q-function estimates as

$$\boldsymbol{Q}^{(t)} := \Big( Q_1^{\pi_1^{(t)}}, \cdots, Q_N^{\pi_N^{(t)}} \Big)^\top, \qquad \boldsymbol{Q}_\tau^{(t)} := \Big( Q_{\tau,1}^{\pi_1^{(t)}}, \cdots, Q_{\tau,N}^{\pi_N^{(t)}} \Big)^\top.$$

We shall often abuse the notation and treat $\boldsymbol{\pi}^{(t)}, \boldsymbol{Q}_\tau^{(t)}$ as matrices in $\mathbb{R}^{N \times |\mathcal{S}||\mathcal{A}|}$, and treat $\boldsymbol{\pi}^{(t)}(a|s)$, $\boldsymbol{Q}_\tau^{(t)}(a|s)$ as vectors in $\mathbb{R}^N$, for all $(s, a) \in \mathcal{S} \times \mathcal{A}$.

**Vanilla federated NPG methods.** To motivate the algorithm development, observe that the NPG method (cf. (9)) applied to (19) adopts the update rule

$$\pi^{(t+1)}(a|s) \propto \pi^{(t)}(a|s) \exp \left( \frac{\eta Q^{\pi^{(t)}}(s, a)}{1-\gamma} \right) = \pi^{(t)}(a|s) \exp \left( \frac{\eta \sum_{n=1}^N Q_n^{\pi^{(t)}}(s, a)}{N(1-\gamma)} \right)$$

for all $(s, a) \in \mathcal{S} \times \mathcal{A}$. Two challenges arise when executing this update rule: the policy estimates are maintained locally without consensus, and the global Q-function are unavailable in the decentralized setting. To address these challenges, we apply the idea of dynamic average consensus (Zhu & Martínez, 2010), where each agent maintains its own estimate $T_n^{(t)}(s, a)$ of the global Q-function, which are collected as vector $\boldsymbol{T}^{(t)} = \big( T_1^{(t)}, \cdots, T_N^{(t)} \big)^\top$. At each iteration, each agent updates its policy estimates based on its neighbors' information via gossip mixing, in addition to a correction term that tracks the difference $Q_n^{\pi_n^{(t+1)}}(s, a) - Q_n^{\pi_n^{(t)}}(s, a)$ of the local Q-functions between consecutive policy updates. Note that the mixing is applied linearly to the logarithms of local policies, which translates into a multiplicative mixing of the local policies. Algorithm 1 summarizes the detailed procedure of the proposed algorithm written in a compact matrix form, which we dub as federated NPG (FedNPG). Note that the agents do not need to share their reward functions with others, and agent $n \in [N]$ will only be responsible to evaluate the local policy $\pi_n^{(t)}$ using the local reward $r_n$.

**Entropy-regularized federated NPG methods.** Moving onto the entropy regularized case, we adopt similar algorithmic ideas to decentralize (10), and propose the federated NPG (FedNPG) method

with entropy regularization, summarized in Algorithm 2 (see Appendix A). Clearly, the entropy-regularized FedNPG method reduces to the vanilla FedNPG in the absence of the regularization (i.e., when $\tau = 0$).

## 4 THEORETICAL GUARANTEES

### 4.1 GLOBAL CONVERGENCE OF FEDNPG

**Convergence with exact policy evaluation.** We begin with the global convergence of FedNPG (cf. Algorithm 1), stated in the following theorem.

**Theorem 1** (Global sublinear convergence of exact FedNPG (informal)). *Suppose $\pi_n^{(0)}, n \in [N]$ are set as the uniform distribution. Then for $0 < \eta \leq \eta_1 := \frac{(1-\sigma)^2(1-\gamma)^3}{16\sqrt{N}\sigma}$, we have*

$$\frac{1}{T}\sum_{t=0}^{T-1}\left(V^\star(\rho) - V^{\overline{\pi}^{(t)}}(\rho)\right) \leq \frac{V^\star(d_\rho^{\pi^\star})}{(1-\gamma)T} + \frac{\log|\mathcal{A}|}{\eta T} + \frac{32N\sigma\eta^2}{(1-\gamma)^9(1-\sigma)^2}. \tag{22}$$

Theorem 1 characterizes the average-iterate convergence of the average policy $\overline{\pi}^{(t)}$ (cf. (21)) across the agents, which depends logarithmically on the size of the action space, and independently on the size of the state space. When $T \geq \frac{128\sqrt{N}\log|\mathcal{A}|\sigma^2}{(1-\sigma)^4}$, by optimizing the learning rate $\eta = \left(\frac{(1-\gamma)^9(1-\sigma)^2\log|\mathcal{A}|}{32TN\sigma}\right)^{1/3}$ to balance the latter two terms, we arrive at

$$\frac{1}{T}\sum_{t=0}^{T-1}\left(V^\star(\rho) - V^{\overline{\pi}^{(t)}}(\rho)\right) \lesssim \frac{V^\star(d_\rho^{\pi^\star})}{(1-\gamma)T} + \frac{N^{1/3}\sigma^{2/3}}{(1-\gamma)^3(1-\sigma)^{2/3}}\left(\frac{\log|\mathcal{A}|}{T}\right)^{2/3}. \tag{23}$$

When the network is fully connected, i.e., $\sigma = 0$, the convergence rate of FedNPG recovers the $\mathcal{O}(1/T)$ rate, matching that of the centralized NPG established in Agarwal et al. (2021). When the network is relatively well-connected in the sense of $\frac{\sigma^2}{(1-\sigma)^2} \lesssim \frac{1-\gamma}{N^{1/2}}$, FedNPG first converges at the rate of $\mathcal{O}(1/T)$, and then at the slower $\mathcal{O}(1/T^{2/3})$ rate after $T \gtrsim \frac{(1-\gamma)^3(1-\sigma)^2}{N\sigma^2}$. In addition, when the network is poorly connected in the sense of $\frac{\sigma^2}{(1-\sigma)^2} \gtrsim \frac{1-\gamma}{N^{1/2}}$, we see that FedNPG converges at the $\mathcal{O}(1/T^{2/3})$ rate. We state the iteration complexity in Corollary 1.

**Corollary 1** (Iteration complexity of exact FedNPG). *To reach $\frac{1}{T}\sum_{t=0}^{T-1}\left(V^\star(\rho) - V^{\overline{\pi}^{(t)}}(\rho)\right) \leq \varepsilon$, the iteration complexity of FedNPG is $\mathcal{O}\left(\left(\frac{\sigma}{(1-\gamma)^{9/2}(1-\sigma)\varepsilon^{3/2}} + \frac{\sigma^2}{(1-\sigma)^4}\right)\sqrt{N}\log|\mathcal{A}| + \frac{1}{\varepsilon(1-\gamma)^2}\right).$*

**Convergence with inexact policy evaluation.** In practice, the policies need to be evaluated using samples collected by the agents, where the Q-functions are only estimated approximately. We are interested in gauging how the approximation error impacts the performance of FedNPG, as demonstrated in the following theorem.

**Theorem 2** (Global sublinear convergence of inexact FedNPG (informal)). *Suppose that $q_n^{\pi_n^{(t)}}$ are used in replace of $Q_n^{\pi_n^{(t)}}$ in Algorithm 1. Under the assumptions of Theorem 1, we have*

$$\frac{1}{T}\sum_{t=0}^{T-1}\left(V^\star(\rho) - V^{\overline{\pi}^{(t)}}(\rho)\right) \leq \frac{V^\star(d_\rho^{\pi^\star})}{(1-\gamma)T} + \frac{\log|\mathcal{A}|}{\eta T} + \frac{32N\sigma^2\eta^2}{(1-\gamma)^9(1-\sigma)^2} + C_3 \max_{\substack{n\in[N],\\ t\in[T]}}\left\|Q_n^{\pi_n^{(t)}} - q_n^{\pi_n^{(t)}}\right\|_\infty, \tag{24}$$

*where $C_3 := \frac{32\sqrt{N}\sigma\eta}{(1-\gamma)^5(1-\sigma)^2}\left(\frac{\eta\sqrt{N}}{(1-\gamma)^3} + 1\right) + \frac{2}{(1-\gamma)^2}.$*

As long as $\max_{n\in[N],t\in[T]}\left\|Q_n^{\pi_n^{(t)}} - q_n^{\pi_n^{(t)}}\right\|_\infty \leq \frac{\varepsilon}{C_3}$, inexact FedNPG reaches $\frac{1}{T}\sum_{t=0}^{T-1}\left(V^\star(\rho) - V^{\overline{\pi}^{(t)}}(\rho)\right) \leq 2\varepsilon$ at the same iteration complexity as predicted in Corollary 1. Equipped with existing sample complexity bounds on policy evaluation, e.g. Li et al. (2020b), this immediate leads to a sample complexity bound for a federated actor-critic type algorithm for multi-task RL, which scales linearly with respect to the size of the state-action space up to logarithmic factors. We leave further details to Remark **??** in Appendix B.4.

### 4.2 GLOBAL CONVERGENCE OF FEDNPG WITH ENTROPY REGULARIZATION

**Convergence with exact policy evaluation.** Next, we present our global convergence guarantee of entropy-regularized FedNPG with exact policy evaluation (cf. Algorithm 2).

**Theorem 3** (Global linear convergence of exact entropy-regularized FedNPG (informal)). *For any $\gamma \in (0,1)$ and $0 < \tau \leq 1$, there exists $\eta_0 = \min\left\{\frac{1-\gamma}{\tau}, \mathcal{O}\left(\frac{(1-\gamma)^7(1-\sigma)^2\tau}{\sigma^2 N}\right)\right\}$, such that if $0 < \eta \leq \eta_0$, then we have*

$$\left\|\overline{Q}_\tau^{(t)} - Q_\tau^\star\right\|_\infty \leq 2\gamma C_1 \rho(\eta)^t, \quad \left\|\log \pi_\tau^\star - \log \overline{\pi}^{(t)}\right\|_\infty \leq \frac{2C_1}{\tau}\rho(\eta)^t, \tag{25}$$

*where $\overline{Q}_\tau^{(t)} := Q_\tau^{\overline{\pi}^{(t)}}$, $\rho(\eta) \leq \max\{1 - \frac{\tau\eta}{2}, \frac{3+\sigma}{4}\} < 1$, and $C_1$ is some problem-dependent constant.*

The exact expressions of $C_1$ and $\eta_0$ are specified in Appendix B.1. Theorem 3 confirms that entropy-regularized FedNPG converges at a linear rate to the optimal regularized policy, which is almost independent of the size of the state-action space, highlighting the positive role of entropy regularization in federated policy optimization. When the network is fully connected, i.e. $\sigma = 0$, the iteration complexity of entropy-regularized FedNPG reduces to $\mathcal{O}\left(\frac{1}{\eta\tau}\log\frac{1}{\varepsilon}\right)$, matching that of the centralized entropy-regularized NPG established in Cen et al. (2021). When the network is less connected, one needs to be more conservative in the choice of learning rates, leading to a higher iteration complexity, as described in the following corollary.

**Corollary 2** (Iteration complexity of exact entropy-regularized FedNPG). *To reach $\left\|\log \pi_\tau^\star - \log \overline{\pi}^{(t)}\right\|_\infty \leq \varepsilon$, the iteration complexity of entropy-regularized FedNPG is*

$$\widetilde{\mathcal{O}}\left(\max\left\{\frac{2}{\tau\eta}, \frac{4}{1-\sigma}\right\}\log\frac{1}{\varepsilon}\right) \tag{26}$$

*up to logarithmic factors. Especially, when $\eta = \eta_0$, the best iteration complexity becomes*

$$\widetilde{\mathcal{O}}\left(\left(\frac{N\sigma^2}{(1-\gamma)^7(1-\sigma)^2\tau^2} + \frac{1}{1-\gamma}\right)\log\frac{1}{\tau\varepsilon}\right).$$

**Convergence with inexact policy evaluation.** Last but not least, we present the informal convergence results of entropy-regularized FedNPG with inexact policy evaluation, whose formal version can be found in Appendix B.2.

**Theorem 4** (Global linear convergence of inexact entropy-regularized FedNPG (informal)). *Suppose that $q_{\tau,n}^{\pi_n^{(t)}}$ are used in replace of $Q_{\tau,n}^{\pi_n^{(t)}}$ in Algorithm 2. Under the assumptions of Theorem 3, we have*

$$\left\|\overline{Q}_\tau^{(t)} - Q_\tau^\star\right\|_\infty \leq 2\gamma\left(C_1\rho(\eta)^t + C_2 \max_{n\in[N],t\in[T]}\left\|Q_{\tau,n}^{\pi_n^{(t)}} - q_{\tau,n}^{\pi_n^{(t)}}\right\|_\infty\right),$$

$$\left\|\log \pi_\tau^\star - \log \overline{\pi}^{(t)}\right\|_\infty \leq \frac{2}{\tau}\left(C_1\rho(\eta)^t + C_2 \max_{n\in[N],t\in[T]}\left\|Q_{\tau,n}^{\pi_n^{(t)}} - q_{\tau,n}^{\pi_n^{(t)}}\right\|_\infty\right),$$

(27)

*where $\overline{Q}_\tau^{(t)} := Q_\tau^{\overline{\pi}^{(t)}}$, $\rho(\eta) \leq \max\{1 - \frac{\tau\eta}{2}, \frac{3+\sigma}{4}\} < 1$, and $C_1$, $C_2$ are problem-dependent constants specified in Appendix B.2.*

## 5 CONCLUSIONS

This work proposes the first provably efficient federated NPG (FedNPG) methods for solving vanilla and entropy-regularized multi-task RL problems in the fully decentralized setting. The established finite-time global convergence guarantees are almost independent of the size of the state-action space up to some logarithmic factor, and illuminate the impacts of the size and connectivity of the network. Furthermore, the proposed FedNPG methods are robust vis-a-vis inexactness of local policy evaluations, leading to a finite-sample complexity bound of a federated actor-critic method for multi-task RL. When it comes to future directions, it would be of great interest to further explore sample-efficient algorithms and examine if it is possible to go beyond the entrywise approximation error assumption in policy evaluation. Another interesting direction is to extend the analysis of FedNPG to incorporate function approximations.

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

# Appendix

## Table of Contents

## A  FEDERATED NPG (FEDNPG) WITH ENTROPY REGULARIZATION

We record the entropy-regularized FedNPG method here due to space limits.

---

**Algorithm 2** Federated NPG (FedNPG) with entropy regularization

---

1: **Input:** learning rate $\eta > 0$, iteration number $T \in \mathbb{N}_+$, mixing matrix $\boldsymbol{W} \in \mathbb{R}^{N \times N}$, regularization coefficient $\tau > 0$.
2: **Initialize:** $\boldsymbol{\pi}^{(0)}, \boldsymbol{T}^{(0)} = \boldsymbol{Q}_\tau^{(0)}$.
3: **for** $t = 0, 1, \cdots$ **do**
4:    Update the policy for each $(s, a) \in \mathcal{S} \times \mathcal{A}$:

$$\log \boldsymbol{\pi}^{(t+1)}(a|s) = \boldsymbol{W} \left( \left( 1 - \frac{\eta\tau}{1-\gamma} \right) \log \boldsymbol{\pi}^{(t)}(a|s) + \frac{\eta}{1-\gamma} \boldsymbol{T}^{(t)}(s, a) \right) - \log \boldsymbol{z}^{(t)}(s) , \tag{$U_\pi$}$$

   where $\boldsymbol{z}^{(t)}(s) = \sum_{a' \in \mathcal{A}} \exp \left\{ \boldsymbol{W} \left( \left( 1 - \frac{\eta\tau}{1-\gamma} \right) \log \boldsymbol{\pi}^{(t)}(a'|s) + \frac{\eta}{1-\gamma} \boldsymbol{T}^{(t)}(s, a') \right) \right\}$.
5:    Evaluate $\boldsymbol{Q}_\tau^{(t+1)}$.
6:    Update the global Q-function estimate for each $(s, a) \in \mathcal{S} \times \mathcal{A}$:

$$\boldsymbol{T}^{(t+1)}(s, a) = \boldsymbol{W} \Big( \boldsymbol{T}^{(t)}(s, a) + \underbrace{\boldsymbol{Q}_\tau^{(t+1)}(s, a) - \boldsymbol{Q}_\tau^{(t)}(s, a)}_{\text{Q-tracking}} \Big) . \tag{$U_T$}$$

7: **end for**

---

# B  CONVERGENCE ANALYSIS

For technical convenience, we present first the analysis and proof for entropy-regularized FedNPG and then for vanilla FedNPG.

## B.1  ANALYSIS OF ENTROPY-REGULARIZED FEDNPG WITH EXACT POLICY EVALUATION

To facilitate analysis, we introduce several notation below. For all $t \geq 0$, we recall $\overline{\pi}^{(t)}$ as the normalized geometric mean of $\{\pi_n^{(t)}\}_{n \in [N]}$:

$$\overline{\pi}^{(t)} := \mathrm{softmax}\left(\frac{1}{N}\sum_{n=1}^{N} \log \pi_n^{(t)}\right), \tag{28}$$

from which we can easily see that for each $(s,a) \in \mathcal{S} \times \mathcal{A}$, $\overline{\pi}^{(t)}(a|s) \propto \left(\prod_{n=1}^{N} \pi_n^{(t)}(a|s)\right)^{\frac{1}{N}}$. We denote the soft $Q$-functions of $\overline{\pi}^{(t)}$ by $\overline{\boldsymbol{Q}}_\tau^{(t)}$:

$$\overline{\boldsymbol{Q}}_\tau^{(t)} := \begin{pmatrix} Q_{\tau,1}^{\overline{\pi}^{(t)}} \\ \vdots \\ Q_{\tau,N}^{\overline{\pi}^{(t)}} \end{pmatrix}. \tag{29}$$

In addition, we define $\widehat{Q}_\tau^{(t)}, \overline{Q}_\tau^{(t)} \in \mathbb{R}^{|\mathcal{S}||\mathcal{A}|}$ and $\overline{V}_\tau^{(t)} \in \mathbb{R}^{|\mathcal{S}|}$ as follows

$$\widehat{Q}_\tau^{(t)} := \frac{1}{N}\sum_{n=1}^{N} Q_{\tau,n}^{\pi_n^{(t)}}, \tag{30a}$$

$$\overline{Q}_\tau^{(t)} := Q_\tau^{\overline{\pi}^{(t)}} = \frac{1}{N}\sum_{n=1}^{N} Q_{\tau,n}^{\overline{\pi}^{(t)}}. \tag{30b}$$

$$\overline{V}_\tau^{(t)} := V_\tau^{\overline{\pi}^{(t)}} = \frac{1}{N}\sum_{n=1}^{N} V_{\tau,n}^{\overline{\pi}^{(t)}}. \tag{30c}$$

For notational convenience, we also denote

$$\alpha := 1 - \frac{\eta\tau}{1-\gamma}. \tag{31}$$

Following Cen et al. (2022), we introduce the following auxiliary sequence $\{\boldsymbol{\xi}^{(t)} = (\xi_1^{(t)}, \cdots, \xi_N^{(t)})^\top \in \mathbb{R}^{N \times |\mathcal{S}||\mathcal{A}|}\}_{t=0,1,\cdots}$, each recursively defined as

$$\forall (s,a) \in \mathcal{S} \times \mathcal{A}: \quad \boldsymbol{\xi}^{(0)}(s,a) := \frac{\|\exp\left(Q_\tau^\star(s,\cdot)/\tau\right)\|_1}{\left\|\exp\left(\frac{1}{N}\sum_{n=1}^{N}\log\pi_n^{(0)}(\cdot|s)\right)\right\|_1} \cdot \boldsymbol{\pi}^{(0)}(a|s), \tag{32a}$$

$$\log \boldsymbol{\xi}^{(t+1)}(s,a) = \boldsymbol{W}\left(\alpha \log \boldsymbol{\xi}^{(t)}(s,a) + (1-\alpha)\boldsymbol{T}^{(t)}(s,a)/\tau\right), \tag{32b}$$

where $\boldsymbol{T}^{(t)}(s,a)$ is updated via $(U_T^0)$. Similarly, we introduce an averaged auxiliary sequence $\{\overline{\xi}^{(t)} \in \mathbb{R}^{|\mathcal{S}||\mathcal{A}|}\}$ given by

$$\forall (s,a) \in \mathcal{S} \times \mathcal{A}: \quad \overline{\xi}^{(0)}(s,a) := \|\exp\left(Q_\tau^\star(s,\cdot)/\tau\right)\|_1 \cdot \overline{\pi}^{(0)}(a|s), \tag{33a}$$

$$\log \overline{\xi}^{(t+1)}(s,a) = \alpha \log \overline{\xi}^{(t)}(s,a) + (1-\alpha)\widehat{Q}_\tau^{(t)}(s,a)/\tau. \tag{33b}$$

We introduces four error metrics defined as

$$\Omega_1^{(t)} := \left\|u^{(t)}\right\|_\infty, \tag{34a}$$

$$\Omega_2^{(t)} := \left\|v^{(t)}\right\|_\infty, \tag{34b}$$

$$\Omega_3^{(t)} := \left\|Q_\tau^\star - \tau\log\overline{\xi}^{(t)}\right\|_\infty, \tag{34c}$$

$$\Omega_4^{(t)} := \max\left\{0, -\min_{s,a}\left(\overline{Q}_\tau^{(t)}(s,a) - \tau\log\overline{\xi}^{(t)}(s,a)\right)\right\}, \tag{34d}$$

where $u^{(t)}, v^{(t)} \in \mathbb{R}^{|\mathcal{S}||\mathcal{A}|}$ are defined as

$$u^{(t)}(s,a) := \left\| \log \boldsymbol{\xi}^{(t)}(s,a) - \log \overline{\xi}^{(t)}(s,a) \mathbf{1}_N \right\|_2 , \tag{35}$$

$$v^{(t)}(s,a) := \left\| \boldsymbol{T}^{(t)}(s,a) - \widehat{Q}_\tau^{(t)}(s,a) \mathbf{1}_N \right\|_2 . \tag{36}$$

We collect the error metrics above in a vector $\boldsymbol{\Omega}^{(t)} \in \mathbb{R}^4$:

$$\boldsymbol{\Omega}^{(t)} := \left( \Omega_1^{(t)}, \Omega_2^{(t)}, \Omega_3^{(t)}, \Omega_4^{(t)} \right)^\top . \tag{37}$$

With the above preparation, we are ready to state the convergence guarantee of Algorithm 2 in Theorem 5 below, which is the formal version of Theorem 3.

**Theorem 5.** *For any $N \in \mathbb{N}_+, \tau > 0, \gamma \in (0,1)$, there exists $\eta_0 > 0$ which depends only on $N, \gamma, \tau, \sigma, |\mathcal{A}|$, such that if $0 < \eta \le \eta_0$ and $1 - \sigma > 0$, then the updates of Algorithm 2 satisfy*

$$\left\| \overline{Q}_\tau^{(t)} - Q_\tau^\star \right\|_\infty \le 2\gamma \rho(\eta)^t \left\| \boldsymbol{\Omega}^{(0)} \right\|_2 , \tag{38}$$

$$\left\| \log \pi_\tau^\star - \log \overline{\pi}^{(t)} \right\|_\infty \le \frac{2}{\tau} \rho(\eta)^t \left\| \boldsymbol{\Omega}^{(0)} \right\|_2 , \tag{39}$$

*where*

$$\rho(\eta) \le \max \left\{ 1 - \frac{\tau\eta}{2}, \frac{3+\sigma}{4} \right\} < 1 .$$

The dependency of $\eta_0$ on $N, \gamma, \tau, \sigma, |\mathcal{A}|$ is made clear in Lemma 2 that will be presented momentarily in this section. The rest of this section is dedicated to the proof of Theorem 5. We first state a key lemma that tracks the error recursion of Algorithm 2.

**Lemma 1.** *The following linear system holds for all $t \ge 0$:*

$$\boldsymbol{\Omega}^{(t+1)} \le \underbrace{\begin{pmatrix} \sigma\alpha & \frac{\eta\sigma}{1-\gamma} & 0 & 0 \\ S\sigma & \left(1 + \frac{\eta M \sqrt{N}}{1-\gamma}\sigma\right)\sigma & \frac{(2+\gamma)\eta MN}{1-\gamma}\sigma & \frac{\gamma\eta MN}{1-\gamma}\sigma \\ (1-\alpha)M & 0 & (1-\alpha)\gamma + \alpha & (1-\alpha)\gamma \\ \frac{2\gamma+\eta\tau}{1-\gamma}M & 0 & 0 & \alpha \end{pmatrix}}_{=:\boldsymbol{A}(\eta)} \boldsymbol{\Omega}^{(t)} , \tag{40}$$

*where we let*

$$S := M\sqrt{N} \left( 2\alpha + (1-\alpha) \cdot \sqrt{2N} + \frac{1-\alpha}{\tau} \cdot \sqrt{N}M \right) , \tag{41}$$

*and*

$$M := \frac{1 + \gamma + 2\tau(1-\gamma)\log|\mathcal{A}|}{(1-\gamma)^2} \cdot \gamma .$$

*In addition, it holds for all $t \ge 0$ that*

$$\left\| \overline{Q}_\tau^{(t)} - Q_\tau^\star \right\|_\infty \le \gamma \Omega_3^{(t)} + \gamma \Omega_4^{(t)} , \tag{42}$$

$$\left\| \log \overline{\pi}^{(t)} - \log \pi_\tau^\star \right\|_\infty \le \frac{2}{\tau} \Omega_3^{(t)} . \tag{43}$$

*Proof.* See Appendix C.1. $\square$

Let $\rho(\eta)$ denote the spectral norm of $\boldsymbol{A}(\eta)$. As $\boldsymbol{\Omega}^{(t)} \ge 0$, it is immediate from (40) that

$$\left\| \boldsymbol{\Omega}^{(t)} \right\|_2 \le \rho(\eta)^t \left\| \boldsymbol{\Omega}^{(0)} \right\|_2 ,$$

and therefore we have

$$\left\| \overline{Q}_\tau^{(t)} - Q_\tau^\star \right\|_\infty \le 2\gamma \left\| \boldsymbol{\Omega}^{(t)} \right\|_\infty \le 2\gamma \rho(\eta)^t \left\| \boldsymbol{\Omega}^{(0)} \right\|_2 ,$$

and

$$\left\| \log \overline{\pi}^{(t)} - \log \pi_\tau^\star \right\|_\infty \le \frac{2}{\tau} \left\| \boldsymbol{\Omega}^{(t)} \right\|_\infty \le \frac{2}{\tau} \rho(\eta)^t \left\| \boldsymbol{\Omega}^{(0)} \right\|_2 .$$

It remains to bound the spectral radius $\rho(\eta)$, which is achieved by the following lemma.

**Lemma 2** (Bounding the spectral norm of $\boldsymbol{A}(\eta)$). *Let*

$$\zeta := \frac{(1-\gamma)(1-\sigma)^2\tau}{8\left(\tau S_0\sigma^2 + 10Mc\sigma^2/(1-\gamma) + (1-\sigma)^2\tau^2/16\right)}, \tag{44}$$

*where* $S_0 := M\sqrt{N}\left(2 + \sqrt{2N} + \frac{M\sqrt{N}}{\tau}\right)$, $c := MN/(1-\gamma)$. *For any* $N \in \mathbb{N}_+, \tau > 0, \gamma \in (0,1)$, *if*

$$0 < \eta \le \eta_0 := \min\left\{\frac{1-\gamma}{\tau}, \zeta\right\}, \tag{45}$$

*then we have*

$$\rho(\eta) \le \max\left\{\frac{3+\sigma}{4}, \frac{1 + (1-\alpha)\gamma + \alpha}{2}\right\} < 1. \tag{46}$$

*Proof.* See Appendix C.2. □

### B.2 ANALYSIS OF ENTROPY-REGULARIZED FEDNPG WITH INEXACT POLICY EVALUATION

We define the collection of *inexact* Q-function estimates as

$$\boldsymbol{q}_\tau^{(t)} := \left(q_{\tau,1}^{\pi_1^{(t)}}, \cdots, q_{\tau,N}^{\pi_N^{(t)}}\right)^\top,$$

and then the update rule $(U_T)$ should be understood as

$$\boldsymbol{T}^{(t+1)}(s,a) = \boldsymbol{W}\left(\boldsymbol{T}^{(t)}(s,a) + \boldsymbol{q}_\tau^{(t+1)}(s,a) - \boldsymbol{q}_\tau^{(t)}(s,a)\right) \tag{47}$$

in the inexact setting. For notational simplicity, we define $e_n \in \mathbb{R}$ as

$$e_n := \max_{t\in[T]}\left\|Q_{\tau,n}^{\pi_n^{(t)}} - q_{\tau,n}^{\pi_n^{(t)}}\right\|_\infty, \quad n \in [N], \tag{48}$$

and let $\boldsymbol{e} = (e_1, \cdots, e_n)^\top$. Define $\widehat{q}_\tau^{(t)}$, the approximation of $\widehat{Q}_\tau^{(t)}$ as

$$\widehat{q}_\tau^{(t)} := \frac{1}{N}\sum_{n=1}^N q_{\tau,n}^{\pi_n^{(t)}}. \tag{49}$$

With slight abuse of notation, we adapt the auxiliary sequence $\{\overline{\xi}^{(t)}\}_{t=0,\cdots}$ to the inexact updates as

$$\overline{\xi}^{(0)}(s,a) := \left\|\exp\left(Q_\tau^\star(s,\cdot)/\tau\right)\right\|_1 \cdot \overline{\pi}^{(0)}(a|s), \tag{50a}$$

$$\overline{\xi}^{(t+1)}(s,a) := \left[\overline{\xi}^{(t)}(s,a)\right]^\alpha \exp\left((1-\alpha)\frac{\widehat{q}_\tau^{(t)}(s,a)}{\tau}\right), \quad \forall(s,a) \in \mathcal{S} \times \mathcal{A}, \ t \ge 0. \tag{50b}$$

In addition, we define

$$\Omega_1^{(t)} := \left\|u^{(t)}\right\|_\infty, \tag{51a}$$

$$\Omega_2^{(t)} := \left\|v^{(t)}\right\|_\infty, \tag{51b}$$

$$\Omega_3^{(t)} := \left\|Q_\tau^\star - \tau\log\overline{\xi}^{(t)}\right\|_\infty, \tag{51c}$$

$$\Omega_4^{(t)} := \max\left\{0, -\min_{s,a}\left(\overline{q}_\tau^{(t)}(s,a) - \tau\log\overline{\xi}^{(t)}(s,a)\right)\right\}, \tag{51d}$$

where

$$u^{(t)}(s,a) := \left\|\log\boldsymbol{\xi}^{(t)}(s,a) - \log\overline{\xi}^{(t)}(s,a)\boldsymbol{1}_N\right\|_2, \tag{52}$$

$$v^{(t)}(s,a) := \left\|\boldsymbol{T}^{(t)}(s,a) - \widehat{q}_\tau^{(t)}(s,a)\boldsymbol{1}_N\right\|_2. \tag{53}$$

We let $\boldsymbol{\Omega}^{(t)}$ be

$$\boldsymbol{\Omega}^{(t)} := \left(\Omega_1^{(t)}, \Omega_2^{(t)}, \Omega_3^{(t)}, \Omega_4^{(t)}\right)^\top. \tag{54}$$

With the above preparation, we are ready to state the inexact convergence guarantee of Algorithm 2 in Theorem 6 below, which is the formal version of Theorem 4.

**Theorem 6.** *Suppose that $q_{\tau,n}^{\pi_n^{(t)}}$ are used in replace of $Q_{\tau,n}^{\pi_n^{(t)}}$ in Algorithm 2. For any $N \in \mathbb{N}_+, \tau > 0, \gamma \in (0,1)$, there exists $\eta_0 > 0$ which depends only on $N, \gamma, \tau, \sigma, |\mathcal{A}|$, such that if $0 < \eta \le \eta_0$ and $1 - \sigma > 0$, we have*

$$\left\| \overline{Q}_\tau^{(t)} - Q_\tau^\star \right\|_\infty \le 2\gamma \left( \rho(\eta)^t \left\| \boldsymbol{\Omega}^{(0)} \right\|_2 + C_2 \max_{n \in [N], t \in [T]} \left\| Q_{\tau,n}^{\pi_n^{(t)}} - q_{\tau,n}^{\pi_n^{(t)}} \right\|_\infty \right), \tag{55}$$

$$\left\| \log \pi_\tau^\star - \log \overline{\pi}^{(t)} \right\|_\infty \le \frac{2}{\tau} \left( \rho(\eta)^t \left\| \boldsymbol{\Omega}^{(0)} \right\|_2 + C_2 \max_{n \in [N], t \in [T]} \left\| Q_{\tau,n}^{\pi_n^{(t)}} - q_{\tau,n}^{\pi_n^{(t)}} \right\|_\infty \right), \tag{56}$$

*where $\rho(\eta) \le \max\{1 - \frac{\tau\eta}{2}, \frac{3+\sigma}{4}\} < 1$ is the same as in Theorem 5, and $C_2 := \frac{\sigma\sqrt{N}(2(1-\gamma)+M\sqrt{N}\eta)+2\gamma^2+\eta\tau}{(1-\gamma)(1-\rho(\eta))}$.*

From Theorem 6, we can conclude that if

$$\max_{n \in [N], t \in [T]} \left\| Q_{\tau,n}^{\pi_n^{(t)}} - q_{\tau,n}^{\pi_n^{(t)}} \right\|_\infty \le \frac{(1-\gamma)(1-\rho(\eta))\varepsilon}{2\gamma\left(\sigma\sqrt{N}(2(1-\gamma)+M\sqrt{N}\eta)+2\gamma^2+\eta\tau\right)}, \tag{57}$$

then inexact entropy-regularized FedNPG could still achieve $2\varepsilon$-accuracy (i.e. $\left\| \overline{Q}_\tau^{(t)} - Q_\tau^\star \right\|_\infty \le 2\varepsilon$) within $\max\left\{ \frac{2}{\tau\eta}, \frac{4}{1-\sigma} \right\} \log \frac{2\gamma\left\| \boldsymbol{\Omega}^{(0)} \right\|_2}{\varepsilon}$ iterations.

**Remark 1.** *When $\eta = \eta_0$ (cf. (45) and (44)) and $\tau \le 1$, the RHS of (57) is of the order*

$$\mathcal{O}\left( \frac{(1-\gamma)\tau\eta_0\varepsilon}{\gamma(\gamma^2 + \sigma\sqrt{N}(1-\gamma))} \right) = \mathcal{O}\left( \frac{(1-\gamma)^8\tau^2(1-\sigma)^2\varepsilon}{\gamma(\gamma^2 + \sigma\sqrt{N}(1-\gamma))(\gamma^2 N\sigma^2 + (1-\sigma)^2\tau^2(1-\gamma)^6)} \right),$$

*which can be translated into a crude sample complexity bound when using fresh samples to estimate the soft Q-functions in each iteration.*

The rest of this section outlines the proof of Theorem 6. We first state a key lemma that tracks the error recursion of Algorithm 2 with inexact policy evaluation, which is a modified version of Lemma 1.

**Lemma 3.** *The following linear system holds for all $t \ge 0$:*

$$\boldsymbol{\Omega}^{(t+1)} \le \boldsymbol{A}(\eta)\boldsymbol{\Omega}^{(t)} + \underbrace{\begin{pmatrix} 0 \\ \sigma\sqrt{N}\left(2 + \frac{M\sqrt{N}\eta}{1-\gamma}\right) \\ \frac{\eta\tau}{1-\gamma} \\ \frac{2\gamma^2}{1-\gamma} \end{pmatrix}}_{=:\boldsymbol{b}(\eta)} \|\boldsymbol{e}\|_\infty, \tag{58}$$

*where $\boldsymbol{A}(\eta)$ is provided in Lemma 1. In addition, it holds for all $t \ge 0$ that*

$$\left\| \overline{Q}_\tau^{(t)} - Q_\tau^\star \right\|_\infty \le \gamma\Omega_3^{(t)} + \gamma\Omega_4^{(t)}, \tag{59}$$

$$\left\| \log \overline{\pi}^{(t)} - \log \pi_\tau^\star \right\|_\infty \le \frac{2}{\tau}\Omega_3^{(t)}. \tag{60}$$

*Proof.* See Appendix C.3. □

By (58), we have

$$\forall t \in N_+: \quad \boldsymbol{\Omega}^{(t)} \le \boldsymbol{A}(\eta)^t\boldsymbol{\Omega}^{(0)} + \sum_{s=1}^t \boldsymbol{A}(\eta)^{t-s}\boldsymbol{b}(\eta),$$

which gives

$$\left\| \boldsymbol{\Omega}^{(t)} \right\|_2 \le \rho(\eta)^t \left\| \boldsymbol{\Omega}^{(0)} \right\|_2 + \sum_{s=1}^t \rho(\eta)^{t-s} \|\boldsymbol{b}(\eta)\|_2 \|\boldsymbol{e}\|_\infty$$

$$\le \rho(\eta)^t \left\| \boldsymbol{\Omega}^{(0)} \right\|_2 + \frac{\sigma\sqrt{N}(2(1-\gamma)+M\sqrt{N}\eta)+2\gamma^2+\eta\tau}{(1-\gamma)(1-\rho(\eta))} \|\boldsymbol{e}\|_\infty. \tag{61}$$

Here, (61) follows from $\|\boldsymbol{b}(\eta)\|_2 \leq \|\boldsymbol{b}(\eta)\|_1 = \frac{\sigma\sqrt{N}(2(1-\gamma)+M\sqrt{N}\eta)+2\gamma^2+\eta\tau}{1-\gamma} \|\boldsymbol{e}\|_\infty$ and $\sum_{s=1}^t \rho(\eta)^{t-s} \leq 1/(1-\rho(\eta))$. Recall that the bound on $\rho(\eta)$ has already been established in Lemma 2. Therefore we complete the proof of Theorem 6 by combining the above inequality with (59) and (60) in a similar fashion as before. We omit further details for conciseness.

### B.3 ANALYSIS OF FEDNPG WITH EXACT POLICY EVALUATION

We state the formal version of Theorem 1 below.

**Theorem 7.** *Suppose all $\pi_n^{(0)}$ in Algorithm 1 are initialized as uniform distribution. When*

$$0 < \eta \leq \eta_1 := \frac{(1-\sigma)^2(1-\gamma)^3}{8(1+\gamma)\gamma\sqrt{N}\sigma^2},$$

*we have*

$$\frac{1}{T}\sum_{t=0}^{T-1}\left(V^\star(\rho) - V^{\overline{\pi}^{(t)}}(\rho)\right) \leq \frac{V^\star(d_\rho^{\pi^\star})}{(1-\gamma)T} + \frac{\log|\mathcal{A}|}{\eta T} + \frac{8(1+\gamma)^2\gamma^2 N\sigma^2}{(1-\gamma)^9(1-\sigma)^2}\eta^2 \tag{62}$$

*for any fixed state distribution $\rho$.*

The rest of this section is dedicated to prove Theorem 7. Similar to (29), we denote the $Q$-functions of $\overline{\pi}^{(t)}$ by $\overline{\boldsymbol{Q}}^{(t)}$:

$$\overline{\boldsymbol{Q}}^{(t)} := \begin{pmatrix} Q_1^{\overline{\pi}^{(t)}} \\ \vdots \\ Q_N^{\overline{\pi}^{(t)}} \end{pmatrix}. \tag{63}$$

In addition, similar to (30), we define $\widehat{Q}^{(t)}, \overline{Q}^{(t)} \in \mathbb{R}^{|\mathcal{S}||\mathcal{A}|}$ and $\overline{V}^{(t)} \in \mathbb{R}^{|\mathcal{S}|}$ as follows

$$\widehat{Q}^{(t)} := \frac{1}{N}\sum_{n=1}^N Q_n^{\pi_n^{(t)}}, \tag{64a}$$

$$\overline{Q}^{(t)} := Q^{\overline{\pi}^{(t)}} = \frac{1}{N}\sum_{n=1}^N Q_n^{\overline{\pi}^{(t)}}. \tag{64b}$$

$$\overline{V}^{(t)} := V^{\overline{\pi}^{(t)}} = \frac{1}{N}\sum_{n=1}^N V_n^{\overline{\pi}^{(t)}}. \tag{64c}$$

Following the same strategy in the analysis of entropy-regularized FedNPG, we introduce the auxiliary sequence $\{\boldsymbol{\xi}^{(t)} = (\xi_1^{(t)}, \cdots, \xi_N^{(t)})^\top \in \mathbb{R}^{N \times |\mathcal{S}||\mathcal{A}|}\}$ recursively:

$$\boldsymbol{\xi}^{(0)}(s,a) := \frac{1}{\left\|\exp\left(\frac{1}{N}\sum_{n=1}^N \log \pi_n^{(0)}(\cdot|s)\right)\right\|_1} \cdot \boldsymbol{\pi}^{(0)}(a|s), \tag{65a}$$

$$\log \boldsymbol{\xi}^{(t+1)}(s,a) = \boldsymbol{W}\left(\log \boldsymbol{\xi}^{(t)}(s,a) + \frac{\eta}{1-\gamma}\boldsymbol{T}^{(t)}(s,a)\right), \tag{65b}$$

as well as the averaged auxiliary sequence $\{\overline{\xi}^{(t)} \in \mathbb{R}^{|\mathcal{S}||\mathcal{A}|}\}$:

$$\overline{\xi}^{(0)}(s,a) := \overline{\pi}^{(0)}(a|s), \tag{66a}$$

$$\log \overline{\xi}^{(t+1)}(s,a) := \log \overline{\xi}^{(t)}(s,a) + \frac{\eta}{1-\gamma}\widehat{Q}^{(t)}(s,a), \quad \forall (s,a) \in \mathcal{S} \times \mathcal{A}, \ t \geq 0. \tag{66b}$$

As usual, we collect the consensus errors in a vector $\boldsymbol{\Omega}^{(t)} = (\|u^{(t)}\|_\infty, \|v^{(t)}\|_\infty)^\top$, where $u^{(t)}, v^{(t)} \in \mathbb{R}^{|\mathcal{S}||\mathcal{A}|}$ are defined as:

$$u^{(t)}(s,a) := \left\|\log \boldsymbol{\xi}^{(t)}(s,a) - \log \overline{\xi}^{(t)}(s,a)\mathbf{1}_N\right\|_2, \tag{67}$$

$$v^{(t)}(s,a) := \left\|\boldsymbol{T}^{(t)}(s,a) - \widehat{Q}^{(t)}(s,a)\mathbf{1}_N\right\|_2. \tag{68}$$

**Step 1: establishing the error recursion.** The next key lemma establishes the error recursion of Algorithm 1.

**Lemma 4.** *The updates of FedNPG satisfy*

$$\mathbf{\Omega}^{(t+1)} \leq \underbrace{\begin{pmatrix} \sigma & \frac{\eta}{1-\gamma}\sigma \\ J\sigma & \sigma\left(1 + \frac{(1+\gamma)\gamma\sqrt{N}\eta}{(1-\gamma)^3}\sigma\right) \end{pmatrix}}_{=:\boldsymbol{B}(\eta)} \mathbf{\Omega}^{(t)} + \underbrace{\begin{pmatrix} 0 \\ \frac{(1+\gamma)\gamma N\sigma}{(1-\gamma)^4}\eta \end{pmatrix}}_{=:\boldsymbol{d}(\eta)} \tag{69}$$

*for all $t \geq 0$, where*

$$J := \frac{2(1+\gamma)\gamma}{(1-\gamma)^2}\sqrt{N}. \tag{70}$$

*In addition, we have*

$$\phi^{(t+1)}(\eta) \leq \phi^{(t)}(\eta) + \frac{2(1+\gamma)\gamma}{(1-\gamma)^4}\eta\|u^{(t)}\|_\infty - \eta\left(V^\star(\rho) - \overline{V}^{(t)}(\rho)\right), \tag{71}$$

*where*

$$\phi^{(t)}(\eta) := \mathbb{E}_{s\sim d_\rho^{\pi^\star}}\left[\mathsf{KL}\big(\pi^\star(\cdot|s)\,\|\,\overline{\pi}^{(t)}(\cdot|s)\big)\right] - \frac{\eta}{1-\gamma}\overline{V}^{(t)}(d_\rho^{\pi^\star}), \quad \forall t \geq 0. \tag{72}$$

*Proof.* See Appendix C.4. □

**Step 2: bounding the value functions.** Let $\boldsymbol{p} \in \mathbb{R}^2$ be defined as:

$$\boldsymbol{p}(\eta) = \begin{pmatrix} p_1(\eta) \\ p_2(\eta) \end{pmatrix} := \frac{2(1+\gamma)\gamma}{(1-\gamma)^4}\begin{pmatrix} \frac{\sigma(1-\gamma)\left(1-\sigma-(1+\gamma)\gamma\sqrt{N}\sigma\eta/(1-\gamma)^3\right)\eta}{(1-\gamma)\left(1-\sigma-(1+\gamma)\gamma\sqrt{N}\sigma^2\eta/(1-\gamma)^3\right)(1-\sigma)-J\sigma^2\eta} \\ \frac{\sigma\eta^2}{(1-\gamma)\left(1-\sigma-(1+\gamma)\gamma\sqrt{N}\sigma^2\eta/(1-\gamma)^3\right)(1-\sigma)-J\sigma^2\eta} \end{pmatrix}; \tag{73}$$

the rationale for this choice will be made clear momentarily. We define the following Lyapunov function

$$\Phi^{(t)}(\eta) = \phi^{(t)}(\eta) + \boldsymbol{p}(\eta)^\top\mathbf{\Omega}^{(t)}, \quad \forall t \geq 0, \tag{74}$$

which satisfies

$$\begin{aligned}
\Phi^{(t+1)}(\eta) &= \phi^{(t+1)}(\eta) + \boldsymbol{p}(\eta)^\top\mathbf{\Omega}^{(t+1)} \\
&\leq \phi^{(t)}(\eta) + \frac{2(1+\gamma)\gamma}{(1-\gamma)^4}\eta\|u^{(t)}\|_\infty - \eta\left(V^\star(\rho) - \overline{V}^{(t)}(\rho)\right) + \boldsymbol{p}(\eta)^\top\left(\boldsymbol{B}(\eta)\mathbf{\Omega}^{(t)} + \boldsymbol{d}(\eta)\right) \\
&= \Phi^{(t)}(\eta) + \left[\boldsymbol{p}(\eta)^\top(\boldsymbol{B}(\eta) - \boldsymbol{I}) + \left(\frac{2(1+\gamma)\gamma}{(1-\gamma)^4}\eta, 0\right)\right]\mathbf{\Omega}^{(t)} - \eta\left(V^\star(\rho) - \overline{V}^{(t)}(\rho)\right) \\
&\quad + p_2(\eta)\frac{(1+\gamma)\gamma N\sigma}{(1-\gamma)^4}\eta. \tag{75}
\end{aligned}$$

Here, the second inequality follows from (71). One can verify that the second term vanishes due to the choice of $\boldsymbol{p}(\eta)$:

$$\boldsymbol{p}(\eta)^\top(\boldsymbol{B}(\eta) - \boldsymbol{I}) + \left(\frac{2(1+\gamma)\gamma}{(1-\gamma)^4}\eta, 0\right) = (0, 0). \tag{76}$$

Therefore, we conclude that

$$V^\star(\rho) - \overline{V}^{(t)}(\rho) \leq \frac{\Phi^{(t)}(\eta) - \Phi^{(t+1)}(\eta)}{\eta} + p_2(\eta)\frac{(1+\gamma)\gamma N\sigma}{(1-\gamma)^4}.$$

Averaging over $t = 0, \cdots, T-1$,

$$\begin{aligned}
&\frac{1}{T}\sum_{t=0}^{T-1}\left(V^\star(\rho) - \overline{V}^{(t)}(\rho)\right) \\
&\leq \frac{\Phi^{(0)}(\eta) - \Phi^{(T)}(\eta)}{\eta T} + \frac{2(1+\gamma)^2\gamma^2}{(1-\gamma)^8}\cdot\frac{N\sigma^2\eta^2}{(1-\gamma)(1-\sigma-(1+\gamma)\gamma\sqrt{N}\sigma^2\eta/(1-\gamma)^3)(1-\sigma)-\sigma^2 J\eta}. 
\end{aligned} \tag{77}$$

**Step 3: simplifying the expression.** We first upper bound the first term in the RHS of (77). Assuming uniform initialization for all $\pi_n^{(0)}$ in Algorithm 1, we have $\left\|u^{(0)}\right\|_\infty = \left\|v^{(0)}\right\|_\infty = 0$, and

$$\mathbb{E}_{s\sim d_\rho^{\pi^\star}}\left[\mathsf{KL}\big(\pi^\star(\cdot|s)\,\|\,\overline{\pi}^{(0)}(\cdot|s)\big)\right] \leq \log|\mathcal{A}|.$$

Therefore, putting together relations (74) and (154) we have

$$\frac{\Phi^{(0)}(\eta) - \Phi^{(T)}(\eta)}{\eta T} \leq \frac{\log|\mathcal{A}|}{T\eta} + \frac{1}{T}\left(\boldsymbol{p}(\eta)^\top \boldsymbol{\Omega}^{(0)}/\eta + \frac{V^\star(d_\rho^{\pi^\star})}{1-\gamma}\right) = \frac{\log|\mathcal{A}|}{T\eta} + \frac{V^\star(d_\rho^{\pi^\star})}{T(1-\gamma)}\,, \quad (78)$$

To continue, we upper bound the second term in the RHS of (77). Note that

$$\eta \leq \eta_1 \leq \frac{(1-\sigma)(1-\gamma)^3}{2(1+\gamma)\gamma\sqrt{N}\sigma^2}\,,$$

which gives

$$\frac{(1+\gamma)\gamma\sqrt{N}\sigma^2}{(1-\gamma)^3}\eta \leq \frac{1-\sigma}{2}. \quad (79)$$

Thus we have

$$(1-\gamma)(1-\sigma-(1+\gamma)\gamma\sqrt{N}\sigma^2\eta/(1-\gamma)^3)(1-\sigma) - J\sigma^2\eta$$
$$\geq (1-\gamma)(1-\sigma)^2/2 - J\sigma^2\eta_1$$
$$\geq (1-\gamma)(1-\sigma)^2/4\,, \quad (80)$$

where the first inequality follows from (79) and the second inequality follows from the definition of $\eta_1$ and $J$. By (80), we deduce

$$\frac{2(1+\gamma)^2\gamma^2}{(1-\gamma)^8}\cdot\frac{N\sigma^2\eta^2}{(1-\gamma)(1-\sigma-(1+\gamma)\gamma\sqrt{N}\sigma^2\eta/(1-\gamma)^3)(1-\sigma) - J\sigma^2\eta} \leq \frac{8(1+\gamma)^2\gamma^2 N\sigma^2}{(1-\gamma)^9(1-\sigma)^2}\eta^2\,, \quad (81)$$

and our advertised bound (62) thus follows from plugging (78) and (81) into (77).

### B.4 ANALYSIS OF FEDNPG WITH INEXACT POLICY EVALUATION

We state the formal version of Theorem 2 below.

**Theorem 8.** *Suppose that $q_n^{\pi_n^{(t)}}$ are used in replace of $Q_n^{\pi_n^{(t)}}$ in Algorithm 1. Suppose all $\pi_n^{(0)}$ in Algorithm 1 set to uniform distribution. Let*

$$0 < \eta \leq \eta_1 := \frac{(1-\sigma)^2(1-\gamma)^3}{8(1+\gamma)\gamma\sqrt{N}\sigma^2}\,,$$

*we have*

$$\frac{1}{T}\sum_{t=0}^{T-1}\left(V^\star(\rho) - V^{\overline{\pi}^{(t)}}(\rho)\right)$$
$$\leq \frac{V^\star(d_\rho^{\pi^\star})}{(1-\gamma)T} + \frac{\log|\mathcal{A}|}{\eta T} + \frac{8(1+\gamma)^2\gamma^2 N\sigma^2}{(1-\gamma)^9(1-\sigma)^2}\eta^2$$
$$+ \left[\frac{8(1+\gamma)\gamma}{(1-\gamma)^5(1-\sigma)^2}\sqrt{N}\sigma\eta\left(\frac{(1+\gamma)\gamma\eta\sqrt{N}}{(1-\gamma)^3} + 2\right) + \frac{2}{(1-\gamma)^2}\right]\max_{n\in[N],t\in[T]}\left\|Q_n^{\pi_n^{(t)}} - q_n^{\pi_n^{(t)}}\right\|_\infty$$

*for any fixed state distribution $\rho$.*

We next outline the proof of Theorem 8. With slight abuse of notation, we again define $e_n \in \mathbb{R}$ as

$$e_n := \max_{t\in[T]}\left\|Q_n^{\pi_n^{(t)}} - q_n^{\pi_n^{(t)}}\right\|_\infty\,, \quad n\in[N]\,, \quad (82)$$

and let $\boldsymbol{e} = (e_1, \cdots, e_n)^\top$. We define the collection of *inexact* Q-function estimates as

$$\boldsymbol{q}^{(t)} := \left( q_1^{\pi_1^{(t)}}, \cdots, q_N^{\pi_N^{(t)}} \right)^\top,$$

and then the update rule $(U_T^0)$ should be understood as

$$\boldsymbol{T}^{(t+1)}(s,a) = \boldsymbol{W} \left( \boldsymbol{T}^{(t)}(s,a) + \boldsymbol{q}^{(t+1)}(s,a) - \boldsymbol{q}^{(t)}(s,a) \right) \tag{83}$$

in the inexact setting. Define $\widehat{q}^{(t)}$, the approximation of $\widehat{Q}^{(t)}$ as

$$\widehat{q}^{(t)} := \frac{1}{N} \sum_{n=1}^{N} q_n^{\pi_n^{(t)}}, \tag{84}$$

we adapt the averaged auxiliary sequence $\{\overline{\xi}^{(t)} \in \mathbb{R}^{|\mathcal{S}||\mathcal{A}|}\}$ to the inexact updates as follows:

$$\overline{\xi}^{(0)}(s,a) := \overline{\pi}^{(0)}(a|s), \tag{85a}$$

$$\overline{\xi}^{(t+1)}(s,a) := \overline{\xi}^{(t)}(s,a) \exp \left( \frac{\eta}{1-\gamma} \widehat{q}^{(t)}(s,a) \right), \quad \forall (s,a) \in \mathcal{S} \times \mathcal{A}, \ t \geq 0. \tag{85b}$$

As usual, we define the consensus error vector as $\boldsymbol{\Omega}^{(t)} = (\|u^{(t)}\|_\infty, \|v^{(t)}\|_\infty)^\top$, where $u^{(t)}, v^{(t)} \in \mathbb{R}^{|\mathcal{S}||\mathcal{A}|}$ are given by

$$u^{(t)}(s,a) := \left\| \log \boldsymbol{\xi}^{(t)}(s,a) - \log \overline{\xi}^{(t)}(s,a) \mathbf{1}_N \right\|_2, \tag{86}$$

$$v^{(t)}(s,a) := \left\| \boldsymbol{T}^{(t)}(s,a) - \widehat{q}^{(t)}(s,a) \mathbf{1}_N \right\|_2. \tag{87}$$

The following lemma characterizes the dynamics of the error vector $\boldsymbol{\Omega}^{(t)}$, perturbed by additional approximation error.

**Lemma 5.** *The updates of inexact FedNPG satisfy*

$$\boldsymbol{\Omega}^{(t+1)} \leq \boldsymbol{B}(\eta)\boldsymbol{\Omega}^{(t)} + \boldsymbol{d}(\eta) + \underbrace{\left( \begin{array}{c} 0 \\ \sqrt{N}\sigma \left( \frac{(1+\gamma)\gamma\eta\sqrt{N}}{(1-\gamma)^3} + 2 \right) \end{array} \right) \|\boldsymbol{e}\|_\infty}_{=:c(\eta)}. \tag{88}$$

*In addition, we have*

$$\phi^{(t+1)}(\eta) \leq \phi^{(t)}(\eta) + \frac{2(1+\gamma)\gamma}{(1-\gamma)^4} \eta \left\| u^{(t)} \right\|_\infty + \frac{2\eta}{(1-\gamma)^2} \|\boldsymbol{e}\|_\infty - \eta \left( V^\star(\rho) - \overline{V}^{(t)}(\rho) \right), \tag{89}$$

*where $\phi^{(t)}(\eta)$ is defined in (72).*

*Proof.* See Appendix C.5. $\qquad\square$

Similar to (75), we can recursively bound $\Phi^{(t)}(\eta)$ (defined in (74)) as

$$
\begin{aligned}
\Phi^{(t+1)}(\eta) &= \phi^{(t+1)}(\eta) + \boldsymbol{p}(\eta)^\top \boldsymbol{\Omega}^{(t+1)} \\
&\overset{(89)}{\leq} \phi^{(t)}(\eta) + \frac{2(1+\gamma)\gamma}{(1-\gamma)^4} \eta \left\| u^{(t)} \right\|_\infty + \frac{2\eta}{(1-\gamma)^2} \|\boldsymbol{e}\|_\infty - \eta \left( V^\star(\rho) - \overline{V}^{(t)}(\rho) \right) \\
&\quad + \boldsymbol{p}(\eta)^\top \left( \boldsymbol{B}(\eta)\boldsymbol{\Omega}^{(t)} + \boldsymbol{d}(\eta) + \boldsymbol{c}(\eta) \right) \\
&= \Phi^{(t)}(\eta) + \underbrace{\left[ \boldsymbol{p}(\eta)^\top (\boldsymbol{B}(\eta) - \boldsymbol{I}) + \left( \frac{2(1+\gamma)\gamma}{(1-\gamma)^4}\eta, 0 \right) \right]}_{=(0,0) \text{ via (76)}} \boldsymbol{\Omega}^{(t)} - \eta \left( V^\star(\rho) - \overline{V}^{(t)}(\rho) \right) \\
&\quad + p_2(\eta) \frac{(1+\gamma)\gamma N\sigma}{(1-\gamma)^4} \eta + \left[ p_2(\eta)\sqrt{N}\sigma \left( \frac{(1+\gamma)\gamma\eta\sqrt{N}}{(1-\gamma)^3} + 2 \right) + \frac{2\eta}{(1-\gamma)^2} \right] \|\boldsymbol{e}\|_\infty.
\end{aligned}
\tag{90}
$$

From the above expression we know that

$$V^\star(\rho) - \overline{V}^{(t)}(\rho) \le \frac{\Phi^{(t)}(\eta) - \Phi^{(t+1)}(\eta)}{\eta} + p_2(\eta)\frac{(1+\gamma)\gamma N\sigma}{(1-\gamma)^4} + \left[p_2(\eta)\sqrt{N}\sigma\left(\frac{(1+\gamma)\gamma\sqrt{N}}{(1-\gamma)^3} + \frac{2}{\eta}\right) + \frac{2}{(1-\gamma)^2}\right]\|e\|_\infty,$$

which gives

$$\frac{1}{T}\sum_{t=0}^{T-1}\left(V^\star(\rho) - \overline{V}^{(t)}(\rho)\right) \le \frac{\Phi^{(0)}(\eta) - \Phi^{(T)}(\eta)}{\eta T} + p_2(\eta)\frac{(1+\gamma)\gamma N\sigma}{(1-\gamma)^4}$$

$$+ \left[p_2(\eta)\sqrt{N}\sigma\left(\frac{(1+\gamma)\gamma\sqrt{N}}{(1-\gamma)^3} + \frac{2}{\eta}\right) + \frac{2}{(1-\gamma)^2}\right]\|e\|_\infty \quad (91)$$

via telescoping. Combining the above expression with (78), (80) and (81), we have

$$\frac{1}{T}\sum_{t=0}^{T-1}\left(V^\star(\rho) - \overline{V}^{(t)}(\rho)\right) \le \frac{\log|\mathcal{A}|}{T\eta} + \frac{V^\star(d_\rho^{\pi^\star})}{T(1-\gamma)} + \frac{8(1+\gamma)^2\gamma^2 N\sigma}{(1-\gamma)^9(1-\sigma)^2}\eta^2$$

$$+ \left[\frac{8(1+\gamma)\gamma}{(1-\gamma)^5(1-\sigma)^2}\sqrt{N}\sigma\eta\left(\frac{(1+\gamma)\gamma\eta\sqrt{N}}{(1-\gamma)^3} + 2\right) + \frac{2}{(1-\gamma)^2}\right]\|e\|_\infty,$$

$$(92)$$

which establishes (82).

## C  PROOF OF KEY LEMMAS

### C.1  PROOF OF LEMMA 1

Before proceeding, we summarize several useful properties of the auxiliary sequences (cf. (32) and (33)), whose proof is postponed to Appendix D.1.

**Lemma 6** (Properties of auxiliary sequences $\{\overline{\xi}^{(t)}\}$ and $\{\boldsymbol{\xi}^{(t)}\}$). $\{\overline{\xi}^{(t)}\}$ and $\{\boldsymbol{\xi}^{(t)}\}$ have the following properties:

1. $\boldsymbol{\xi}^{(t)}$ can be viewed as an unnormalized version of $\boldsymbol{\pi}^{(t)}$, i.e.,

$$\pi_n^{(t)}(\cdot|s) = \frac{\xi_n^{(t)}(s,\cdot)}{\left\|\xi_n^{(t)}(s,\cdot)\right\|_1}, \quad \forall n \in [N], s \in \mathcal{S}. \quad (93)$$

2. For any $t \ge 0$, $\log\overline{\xi}^{(t)}$ keeps track of the average of $\log\boldsymbol{\xi}^{(t)}$, i.e.,

$$\frac{1}{N}\mathbf{1}_N^\top\log\boldsymbol{\xi}^{(t)} = \log\overline{\xi}^{(t)}. \quad (94)$$

   It follows that

$$\forall s \in \mathcal{S}, t \ge 0: \quad \overline{\pi}^{(t)}(\cdot|s) = \frac{\overline{\xi}^{(t)}(s,\cdot)}{\left\|\overline{\xi}^{(t)}(s,\cdot)\right\|_1}. \quad (95)$$

**Lemma 7** ((Cen et al., 2022, Appendix. A.2)). *For any vector $\theta = [\theta_a]_{a\in\mathcal{A}} \in \mathbb{R}^{|\mathcal{A}|}$, we denote by $\pi_\theta \in \mathbb{R}^{|\mathcal{A}|}$ the softmax transform of $\theta$ such that*

$$\pi_\theta(a) = \frac{\exp(\theta_a)}{\sum_{a'\in\mathcal{A}}\exp(\theta_{a'})}, \quad a \in \mathcal{A}. \quad (96)$$

*For any $\theta_1, \theta_2 \in \mathbb{R}^{|\mathcal{A}|}$, we have*

$$\left|\log(\|\exp(\theta_1)\|_1) - \log(\|\exp(\theta_2)\|_1)\right| \le \|\theta_1 - \theta_2\|_\infty, \quad (97)$$

$$\|\log\pi_{\theta_1} - \log\pi_{\theta_2}\|_\infty \le 2\|\theta_1 - \theta_2\|_\infty. \quad (98)$$

**Step 1: bound** $u^{(t+1)}(s,a) = \left\| \log \boldsymbol{\xi}^{(t+1)}(s,a) - \log \overline{\xi}^{(t+1)}(s,a) \mathbf{1}_N \right\|_2$. By (32b) and (33b) we have

$$
\begin{aligned}
u^{(t+1)}(s,a) &= \left\| \log \boldsymbol{\xi}^{(t+1)}(s,a) - \log \overline{\xi}^{(t+1)}(s,a) \mathbf{1}_N \right\|_2 \\
&= \left\| \alpha \left( \boldsymbol{W} \log \boldsymbol{\xi}^{(t)}(s,a) - \log \overline{\xi}^{(t)}(s,a) \mathbf{1}_N \right) + (1-\alpha) \left( \boldsymbol{W} \boldsymbol{T}^{(t)}(s,a) - \widehat{Q}_\tau^{(t)}(s,a) \mathbf{1}_N \right)/\tau \right\|_2 \\
&\le \sigma\alpha \left\| \log \boldsymbol{\xi}^{(t)}(s,a) - \log \overline{\xi}^{(t)}(s,a) \mathbf{1}_N \right\|_2 + \frac{1-\alpha}{\tau}\sigma \left\| \boldsymbol{T}^{(t)}(s,a) - \widehat{Q}_\tau^{(t)}(s,a) \mathbf{1}_N \right\|_2 \\
&\le \sigma\alpha \left\| u^{(t)} \right\|_\infty + \frac{1-\alpha}{\tau}\sigma \left\| v^{(t)} \right\|_\infty,
\end{aligned}
\tag{99}
$$

where the penultimate step results from the averaging property of $\boldsymbol{W}$ (property (18)). Taking maximum over $(s,a) \in \mathcal{S} \times \mathcal{A}$ establishes the bound on $\Omega_1^{(t+1)}$ in (40).

**Step 2: bound** $v^{(t+1)}(s,a) = \left\| \boldsymbol{T}^{(t+1)}(s,a) - \widehat{Q}_\tau^{(t+1)}(s,a) \mathbf{1}_N \right\|_2$. By $(U_T)$ we have

$$
\begin{aligned}
&\left\| \boldsymbol{T}^{(t+1)}(s,a) - \widehat{Q}_\tau^{(t+1)}(s,a) \mathbf{1}_N \right\|_2 \\
&= \left\| \boldsymbol{W} \left( \boldsymbol{T}^{(t)}(s,a) + \boldsymbol{Q}_\tau^{(t+1)}(s,a) - \boldsymbol{Q}_\tau^{(t)}(s,a) \right) - \widehat{Q}_\tau^{(t+1)}(s,a) \mathbf{1}_N \right\|_2 \\
&= \left\| \left( \boldsymbol{W} \boldsymbol{T}^{(t)}(s,a) - \widehat{Q}_\tau^{(t)}(s,a) \mathbf{1}_N \right) + \boldsymbol{W} \left( \boldsymbol{Q}_\tau^{(t+1)}(s,a) - \boldsymbol{Q}_\tau^{(t)}(s,a) \right) + \left( \widehat{Q}_\tau^{(t)}(s,a) - \widehat{Q}_\tau^{(t+1)}(s,a) \right) \mathbf{1}_N \right\|_2 \\
&\le \sigma \left\| \boldsymbol{T}^{(t)}(s,a) - \widehat{Q}_\tau^{(t)}(s,a) \mathbf{1}_N \right\|_2 + \sigma \left\| \left( \boldsymbol{Q}_\tau^{(t+1)}(s,a) - \boldsymbol{Q}_\tau^{(t)}(s,a) \right) + \left( \widehat{Q}_\tau^{(t)}(s,a) - \widehat{Q}_\tau^{(t+1)}(s,a) \right) \mathbf{1}_N \right\|_2 \\
&\le \sigma \left\| \boldsymbol{T}^{(t)}(s,a) - \widehat{Q}_\tau^{(t)}(s,a) \mathbf{1}_N \right\|_2 + \sigma \left\| \boldsymbol{Q}_\tau^{(t+1)}(s,a) - \boldsymbol{Q}_\tau^{(t)}(s,a) \right\|_2,
\end{aligned}
\tag{100}
$$

where the penultimate step uses property (18), and the last step is due to

$$
\begin{aligned}
&\left\| \left( \boldsymbol{Q}_\tau^{(t+1)}(s,a) - \boldsymbol{Q}_\tau^{(t)}(s,a) \right) + \left( \widehat{Q}_\tau^{(t)}(s,a) - \widehat{Q}_\tau^{(t+1)}(s,a) \right) \mathbf{1}_N \right\|_2^2 \\
&= \left\| \boldsymbol{Q}_\tau^{(t+1)}(s,a) - \boldsymbol{Q}_\tau^{(t)}(s,a) \right\|_2^2 + N \left( \widehat{Q}_\tau^{(t)}(s,a) - \widehat{Q}_\tau^{(t+1)}(s,a) \right)^2 \\
&\quad - 2 \sum_{n=1}^N \left( Q_{\tau,n}^{\pi_n^{(t+1)}}(s,a) - Q_{\tau,n}^{\pi_n^{(t)}}(s,a) \right) \left( \widehat{Q}_\tau^{(t+1)}(s,a) - \widehat{Q}_\tau^{(t)}(s,a) \right) \\
&= \left\| \boldsymbol{Q}_\tau^{(t+1)}(s,a) - \boldsymbol{Q}_\tau^{(t)}(s,a) \right\|_2^2 - N \left( \widehat{Q}_\tau^{(t)}(s,a) - \widehat{Q}_\tau^{(t+1)}(s,a) \right)^2 \\
&\le \left\| \boldsymbol{Q}_\tau^{(t+1)}(s,a) - \boldsymbol{Q}_\tau^{(t)}(s,a) \right\|_2^2.
\end{aligned}
$$

**Step 3: bound** $\left\| Q_\tau^\star - \tau \log \overline{\xi}^{(t+1)} \right\|_\infty$. We decompose the term of interest as

$$
\begin{aligned}
Q_\tau^\star - \tau \log \overline{\xi}^{(t+1)} &= Q_\tau^\star - \tau\alpha \log \overline{\xi}^{(t)} - (1-\alpha)\widehat{Q}_\tau^{(t)} \\
&= \alpha(Q_\tau^\star - \tau \log \overline{\xi}^{(t)}) + (1-\alpha)(Q_\tau^\star - \overline{Q}_\tau^{(t)}) + (1-\alpha)(\overline{Q}_\tau^{(t)} - \widehat{Q}_\tau^{(t)}),
\end{aligned}
$$

which gives

$$
\left\| Q_\tau^\star - \tau \log \overline{\xi}^{(t+1)} \right\|_\infty \le \alpha \left\| Q_\tau^\star - \tau \log \overline{\xi}^{(t)} \right\|_\infty + (1-\alpha) \left\| Q_\tau^\star - \overline{Q}_\tau^{(t)} \right\|_\infty + (1-\alpha) \left\| \overline{Q}_\tau^{(t)} - \widehat{Q}_\tau^{(t)} \right\|_\infty.
\tag{101}
$$

Note that we can upper bound $\left\| \overline{Q}_\tau^{(t)} - \widehat{Q}_\tau^{(t)} \right\|_\infty$ by

$$
\begin{aligned}
\left\| \overline{Q}_\tau^{(t)} - \widehat{Q}_\tau^{(t)} \right\|_\infty &= \left\| \frac{1}{N} \sum_{n=1}^N Q_{\tau,n}^{\pi_n^{(t)}} - \frac{1}{N} \sum_{n=1}^N Q_{\tau,n}^{\overline{\pi}^{(t)}} \right\|_\infty \\
&\le \frac{1}{N} \sum_{n=1}^N \left\| Q_{\tau,n}^{\pi_n^{(t)}} - Q_{\tau,n}^{\overline{\pi}^{(t)}} \right\|_\infty \\
&\le \frac{M}{N} \sum_{n=1}^N \left\| \log \xi_n^{(t)} - \log \overline{\xi}^{(t)} \right\|_\infty \le M \left\| u^{(t)} \right\|_\infty.
\end{aligned}
\tag{102}
$$

The last step is due to $\big| \log \xi_n^{(t)}(s,a) - \log \overline{\xi}^{(t)}(s,a) \big| \leq u^{(t)}(s,a)$, while the penultimate step results from writing

$$\overline{\pi}^{(t)}(\cdot|s) = \mathrm{softmax}\left(\log \overline{\xi}^{(t)}(s,\cdot)\right),$$

$$\pi_n^{(t)}(\cdot|s) = \mathrm{softmax}\left(\log \xi_n^{(t)}(s,\cdot)\right),$$

and applying the following lemma.

**Lemma 8** (Lipschitz constant of soft Q-function). *Assume that* $r(s,a) \in [0,1], \forall (s,a) \in \mathcal{S} \times \mathcal{A}$ *and* $\tau \geq 0$. *For any* $\theta, \theta' \in \mathbb{R}^{|\mathcal{S}||\mathcal{A}|}$, *we have*

$$\left\| Q_\tau^{\pi_{\theta'}} - Q_\tau^{\pi_\theta} \right\|_\infty \leq \underbrace{\frac{1 + \gamma + 2\tau(1-\gamma)\log|\mathcal{A}|}{(1-\gamma)^2} \cdot \gamma}_{=:M} \left\| \theta' - \theta \right\|_\infty . \tag{103}$$

Plugging (102) into (101) gives

$$\left\| Q_\tau^\star - \tau \log \overline{\xi}^{(t+1)} \right\|_\infty \leq \alpha \left\| Q_\tau^\star - \tau \log \overline{\xi}^{(t)} \right\|_\infty + (1-\alpha)\left\| Q_\tau^\star - \overline{Q}_\tau^{(t)} \right\|_\infty + (1-\alpha)M\left\| u^{(t)} \right\|_\infty . \tag{104}$$

**Step 4: bound** $\left\| \boldsymbol{Q}_\tau^{(t+1)}(s,a) - \boldsymbol{Q}_\tau^{(t)}(s,a) \right\|_2$. Let $w^{(t)} : \mathcal{S} \times \mathcal{A} \to \mathbb{R}$ be defined as

$$\forall (s,a) \in \mathcal{S} \times \mathcal{A}: \quad w^{(t)}(s,a) := \left\| \log \boldsymbol{\xi}^{(t+1)}(s,a) - \log \boldsymbol{\xi}^{(t)}(s,a) - (1-\alpha)V_\tau^\star(s)\mathbf{1}_N/\tau \right\|_2 . \tag{105}$$

Again, we treat $w^{(t)}$ as vectors in $\mathbb{R}^{|\mathcal{S}||\mathcal{A}|}$ whenever it is clear from context. For any $(s,a) \in \mathcal{S} \times \mathcal{A}$ and $n \in [N]$, by Lemma 8 it follows that

$$\left| Q_{\tau,n}^{\pi_n^{(t+1)}}(s,a) - Q_{\tau,n}^{\pi_n^{(t)}}(s,a) \right| \leq M \max_{s \in \mathcal{S}} \left\| \log \xi_n^{(t+1)}(s,\cdot) - \log \xi_n^{(t)}(s,\cdot) - (1-\alpha)V_\tau^\star(s)\mathbf{1}_{|\mathcal{A}|}/\tau \right\|_\infty$$

$$\leq M \max_{s \in \mathcal{S}} \max_{a \in \mathcal{A}} w^{(t)}(s,a) \leq M \left\| w^{(t)} \right\|_\infty , \tag{106}$$

and consequently

$$\left\| \boldsymbol{Q}_\tau^{(t+1)}(s,a) - \boldsymbol{Q}_\tau^{(t)}(s,a) \right\|_2 \leq M\sqrt{N} \left\| w^{(t)} \right\|_\infty . \tag{107}$$

It boils down to control $\left\| w^{(t)} \right\|_\infty$. To do so, we first note that for each $(s,a) \in \mathcal{S} \times \mathcal{A}$, we have

$$w^{(t)}(s,a)$$
$$= \left\| \boldsymbol{W}\left( \alpha \log \boldsymbol{\xi}^{(t)}(s,a) + (1-\alpha)\boldsymbol{T}^{(t)}(s,a)/\tau \right) - \log \boldsymbol{\xi}^{(t)}(s,a) - (1-\alpha)V_\tau^\star(s)\mathbf{1}_N/\tau \right\|_2$$
$$\overset{(a)}{=} \left\| \alpha(\boldsymbol{W} - \boldsymbol{I}_N)\left( \log \boldsymbol{\xi}^{(t)}(s,a) - \log \overline{\xi}^{(t)}(s,a)\mathbf{1}_N \right) + (1-\alpha)\left( \boldsymbol{W}\boldsymbol{T}^{(t)}(s,a)/\tau - \log \boldsymbol{\xi}^{(t)}(s,a) - V_\tau^\star(s)\mathbf{1}_N/\tau \right) \right\|_2$$
$$\overset{(b)}{\leq} 2\alpha \left\| \log \boldsymbol{\xi}^{(t)}(s,a) - \log \overline{\xi}^{(t)}(s,a)\mathbf{1}_N \right\|_2 + \frac{1-\alpha}{\tau} \left\| \boldsymbol{W}\boldsymbol{T}^{(t)}(s,a) - \tau \log \boldsymbol{\xi}^{(t)}(s,a) - V_\tau^\star(s)\mathbf{1}_N \right\|_2 \tag{108}$$

where (a) is due to the doubly stochasticity property of $\boldsymbol{W}$ and (b) is from the fact $\left\| \boldsymbol{W} - \boldsymbol{I}_N \right\|_2 \leq 2$. We further bound the second term as follows:

$$\left\| \boldsymbol{W}\boldsymbol{T}^{(t)}(s,a) - \tau \log \boldsymbol{\xi}^{(t)}(s,a) - V_\tau^\star(s)\mathbf{1}_N \right\|_2$$
$$= \left\| \boldsymbol{W}\boldsymbol{T}^{(t)}(s,a) - \tau \log \boldsymbol{\xi}^{(t)}(s,a) - \left( Q_\tau^\star(s,a) - \tau \log \pi_\tau^\star(a|s) \right)\mathbf{1}_N \right\|_2$$
$$\leq \left\| \boldsymbol{W}\boldsymbol{T}^{(t)}(s,a) - Q_\tau^\star(s,a)\mathbf{1}_N \right\|_2 + \tau \left\| \log \boldsymbol{\xi}^{(t)}(s,a) - \log \pi_\tau^\star(a|s)\mathbf{1}_N \right\|_2$$
$$\leq \left\| \boldsymbol{W}\boldsymbol{T}^{(t)}(s,a) - \widehat{Q}_\tau(s,a)\mathbf{1}_N \right\|_2 + \left\| \widehat{Q}_\tau(s,a)\mathbf{1}_N - Q_\tau^\star(s,a)\mathbf{1}_N \right\|_2$$
$$\quad + \tau \left\| \log \boldsymbol{\xi}^{(t)}(s,a) - \log \overline{\pi}^{(t)}(a|s)\mathbf{1}_N \right\|_2 + \tau \left\| \log \overline{\pi}^{(t)}(a|s)\mathbf{1}_N - \log \pi_\tau^\star(a|s)\mathbf{1}_N \right\|_2$$
$$= \sigma \left\| \boldsymbol{T}^{(t)}(s,a) - \widehat{Q}_\tau^{(t)}(s,a)\mathbf{1}_N \right\|_2 + \sqrt{N}\left| \widehat{Q}_\tau^{(t)}(s,a) - Q_\tau^\star(s,a) \right|$$
$$\quad + \tau \left\| \log \boldsymbol{\xi}^{(t)}(s,a) - \log \overline{\pi}^{(t)}(a|s)\mathbf{1}_N \right\|_2 + \tau\sqrt{N}\left| \log \overline{\pi}^{(t)}(a|s) - \log \pi_\tau^\star(a|s) \right| . \tag{109}$$

Here, the first step results from the following relation established in Nachum et al. (2017):

$$\forall (s,a) \in \mathcal{S} \times \mathcal{A}: \quad V_\tau^\star(s) = -\tau \log \pi_\tau^\star(a|s) + Q_\tau^\star(s,a), \tag{110}$$

which also leads to

$$\big\| \log \overline{\pi}^{(t)} - \log \pi_\tau^\star \big\|_\infty \le \frac{2}{\tau} \big\| Q_\tau^\star - \tau \log \overline{\xi}^{(t)} \big\|_\infty \tag{111}$$

by Lemma 7. For the remaining terms in (109), we have

$$\big| \widehat{Q}_\tau^{(t)}(s,a) - Q_\tau^\star(s,a) \big| \le \big\| \widehat{Q}_\tau^{(t)} - \overline{Q}_\tau^{(t)} \big\|_\infty + \big\| \overline{Q}_\tau^{(t)} - Q_\tau^\star \big\|_\infty, \tag{112}$$

and

$$\begin{aligned}
\big\| \log \boldsymbol{\xi}^{(t)}(s,a) - \log \overline{\pi}^{(t)}(a|s) \mathbf{1}_N \big\|_2 &= \sqrt{\sum_{n=1}^N \Big( \log \xi_n^{(t)}(s,a) - \log \overline{\pi}^{(t)}(a|s) \Big)^2} \\
&\le \sqrt{\sum_{n=1}^N 2 \big\| \log \xi_n^{(t)} - \log \overline{\xi}^{(t)} \big\|_\infty^2} \\
&\le \sqrt{\sum_{n=1}^N 2 \big\| u^{(t)} \big\|_\infty^2} = \sqrt{2N} \big\| u^{(t)} \big\|_\infty, \tag{113}
\end{aligned}$$

where the first inequality again results from Lemma 7. Plugging (111), (112), (113) into (109) and using the definition of $u^{(t)}, v^{(t)}$, we arrive at

$$\begin{aligned}
w^{(t)}(s,a) &\le \Big( 2\alpha + (1-\alpha) \cdot \sqrt{2N} \Big) \big\| u^{(t)} \big\|_\infty + \frac{1-\alpha}{\tau} \big\| v^{(t)} \big\|_\infty + \frac{1-\alpha}{\tau} \cdot \sqrt{N} \Big( \big\| \widehat{Q}_\tau^{(t)} - \overline{Q}_\tau^{(t)} \big\|_\infty + \big\| \overline{Q}_\tau^{(t)} - Q_\tau^\star \big\|_\infty \Big) \\
&\quad + \frac{1-\alpha}{\tau} \cdot 2\sqrt{N} \big\| Q_\tau^\star - \tau \log \overline{\xi}^{(t)} \big\|_\infty.
\end{aligned}$$

Using previous display, we can write (107) as

$$\begin{aligned}
\big\| \boldsymbol{Q}_\tau^{(t+1)}(s,a) &- \boldsymbol{Q}_\tau^{(t)}(s,a) \big\|_2 \\
&\le M\sqrt{N} \bigg\{ \Big( 2\alpha + (1-\alpha) \cdot \sqrt{2N} \Big) \big\| u^{(t)} \big\|_\infty + \frac{1-\alpha}{\tau} \sigma \big\| v^{(t)} \big\|_\infty \\
&\quad + \frac{1-\alpha}{\tau} \cdot \sqrt{N} \Big( M \big\| u^{(t)} \big\|_\infty + \big\| \overline{Q}_\tau^{(t)} - Q_\tau^\star \big\|_\infty \Big) + \frac{1-\alpha}{\tau} \cdot 2\sqrt{N} \big\| Q_\tau^\star - \tau \log \overline{\xi}^{(t)} \big\|_\infty \bigg\}.
\end{aligned} \tag{114}$$

Combining (100) with the above expression (114), we get

$$\begin{aligned}
\big\| v^{(t+1)} \big\|_\infty &\le \sigma \left( 1 + \frac{\eta M \sqrt{N}}{1-\gamma} \sigma \right) \big\| v^{(t)} \big\|_\infty + \sigma M \sqrt{N} \bigg\{ \left( 2\alpha + (1-\alpha) \cdot \sqrt{2N} + \frac{1-\alpha}{\tau} \cdot \sqrt{N} M \right) \big\| u^{(t)} \big\|_\infty \\
&\quad + \frac{1-\alpha}{\tau} \cdot \sqrt{N} \big\| \overline{Q}_\tau^{(t)} - Q_\tau^\star \big\|_\infty + \frac{1-\alpha}{\tau} \cdot 2\sqrt{N} \big\| Q_\tau^\star - \tau \log \overline{\xi}^{(t)} \big\|_\infty \bigg\}.
\end{aligned} \tag{115}$$

**Step 5: bound $\big\| \overline{Q}_\tau^{(t+1)} - Q_\tau^\star \big\|_\infty$.** For any state-action pair $(s,a) \in \mathcal{S} \times \mathcal{A}$, we observe that

$$\begin{aligned}
Q_\tau^\star(s,a) &- \overline{Q}_\tau^{(t+1)}(s,a) \\
&= r(s,a) + \gamma \mathop{\mathbb{E}}_{s' \sim P(\cdot|s,a)} [V_\tau^\star(s')] - \left( r(s,a) + \gamma \mathop{\mathbb{E}}_{s' \sim P(\cdot|s,a)} \left[ V_\tau^{\overline{\pi}^{(t+1)}}(s') \right] \right) \\
&= \gamma \mathop{\mathbb{E}}_{s' \sim P(\cdot|s,a)} \left[ \tau \log \left( \left\| \exp \left( \frac{Q_\tau^\star(s',\cdot)}{\tau} \right) \right\|_1 \right) \right] - \gamma \mathop{\mathbb{E}}_{\substack{s' \sim P(\cdot|s,a), \\ a' \sim \overline{\pi}^{(t+1)}(\cdot|s')}} \left[ \overline{Q}_\tau^{(t+1)}(s',a') - \tau \log \overline{\pi}^{(t+1)}(a'|s') \right],
\end{aligned} \tag{116}$$

where the first step invokes the definition of $Q_\tau$ (cf. (7a)), and the second step is due to the following expression of $V_\tau^\star$ established in Nachum et al. (2017):

$$V_\tau^\star(s) = \tau \log \left( \left\| \exp \left( \frac{Q_\tau^\star(s, \cdot)}{\tau} \right) \right\|_1 \right) . \tag{117}$$

To continue, note that by (95) and (33b) we have

$$\log \overline{\pi}^{(t+1)}(a|s) = \log \overline{\xi}^{(t+1)}(s, a) - \log \left( \left\| \overline{\xi}^{(t+1)}(s, \cdot) \right\|_1 \right)$$

$$= \alpha \log \overline{\xi}^{(t)}(s, a) + (1 - \alpha) \frac{\widehat{Q}_\tau^{(t)}(s, a)}{\tau} - \log \left( \left\| \overline{\xi}^{(t+1)}(s, \cdot) \right\|_1 \right) . \tag{118}$$

Plugging (118) into (116) and (114) establishes the bounds on

$$Q_\tau^\star(s, a) - \overline{Q}_\tau^{(t+1)}(s, a) = \gamma \mathop{\mathbb{E}}_{s' \sim P(\cdot|s,a)} \left[ \tau \log \left( \left\| \exp \left( \frac{Q_\tau^\star(s', \cdot)}{\tau} \right) \right\|_1 \right) - \tau \log \left( \left\| \overline{\xi}^{(t+1)}(s', \cdot) \right\|_1 \right) \right]$$

$$- \gamma \mathop{\mathbb{E}}_{\substack{s' \sim P(\cdot|s,a), \\ a' \sim \overline{\pi}^{(t+1)}(\cdot|s')}} \left[ \overline{Q}_\tau^{(t+1)}(s', a') - \tau \underbrace{\left( \alpha \log \overline{\xi}^{(t)}(s', a') + (1 - \alpha) \frac{\widehat{Q}_\tau^{(t)}(s', a')}{\tau} \right)}_{= \log \overline{\xi}^{(t+1)}(s', a')} \right] \tag{119}$$

for any $(s, a) \in \mathcal{S} \times \mathcal{A}$. In view of property (97), the first term on the right-hand side of (119) can be bounded by

$$\tau \log \left( \left\| \exp \left( \frac{Q_\tau^\star(s', \cdot)}{\tau} \right) \right\|_1 \right) - \tau \log \left( \left\| \overline{\xi}^{(t+1)}(s', \cdot) \right\|_1 \right) \leq \left\| Q_\tau^\star - \tau \log \overline{\xi}^{(t+1)} \right\|_\infty .$$

Plugging the above expression into (119), we have

$$0 \leq Q_\tau^\star(s, a) - \overline{Q}_\tau^{(t+1)}(s, a) \leq \gamma \left\| Q_\tau^\star - \tau \log \overline{\xi}^{(t+1)} \right\|_\infty - \gamma \min_{s,a} \left( \overline{Q}_\tau^{(t+1)}(s, a) - \tau \log \overline{\xi}^{(t+1)}(s, a) \right) ,$$

which gives

$$\left\| Q_\tau^\star - \overline{Q}_\tau^{(t+1)} \right\|_\infty \leq \gamma \left\| Q_\tau^\star - \tau \log \overline{\xi}^{(t+1)} \right\|_\infty + \gamma \max \left\{ 0, - \min_{s,a} \left( \overline{Q}_\tau^{(t+1)}(s, a) - \tau \log \overline{\xi}^{(t+1)}(s, a) \right) \right\} . \tag{120}$$

Plugging the above inequality into (104) and (115) establishes the bounds on $\Omega_3^{(t+1)}$ and $\Omega_2^{(t+1)}$ in (40), respectively.

**Step 6: bound** $- \min_{s,a} \left( \overline{Q}_\tau^{(t+1)}(s, a) - \tau \log \overline{\xi}^{(t+1)}(s, a) \right)$. We need the following lemma which is adapted from Lemma 1 in Cen et al. (2022):

**Lemma 9** (Performance improvement of FedNPG with entropy regularization). *Suppose* $0 < \eta \leq (1 - \gamma)/\tau$. *For any state-action pair* $(s_0, a_0) \in \mathcal{S} \times \mathcal{A}$, *one has*

$$\overline{V}_\tau^{(t+1)}(s_0) - \overline{V}_\tau^{(t)}(s_0) \geq \frac{1}{\eta} \mathop{\mathbb{E}}_{s \sim d_{s_0}^{\overline{\pi}^{(t+1)}}} \left[ \alpha \mathsf{KL}\big( \overline{\pi}^{(t+1)}(\cdot|s_0) \,\|\, \overline{\pi}^{(t)}(\cdot|s_0) \big) + \mathsf{KL}\big( \overline{\pi}^{(t)}(\cdot|s_0) \,\|\, \overline{\pi}^{(t+1)}(\cdot|s_0) \big) \right]$$

$$- \frac{2}{1 - \gamma} \left\| \widehat{Q}_\tau^{(t)} - \overline{Q}_\tau^{(t)} \right\|_\infty , \tag{121}$$

$$\overline{Q}_\tau^{(t+1)}(s_0, a_0) - \overline{Q}_\tau^{(t)}(s_0, a_0) \geq - \frac{2\gamma}{1 - \gamma} \left\| \widehat{Q}_\tau^{(t)} - \overline{Q}_\tau^{(t)} \right\|_\infty . \tag{122}$$

*Proof.* See Appendix D.3. ∎

Using (122), we have

$$\overline{Q}_\tau^{(t+1)}(s, a) - \tau \left( \alpha \log \overline{\xi}^{(t)}(s, a) + (1 - \alpha) \frac{\widehat{Q}_\tau^{(t)}(s, a)}{\tau} \right)$$

$$\geq \overline{Q}_\tau^{(t)}(s, a) - \tau \left( \alpha \log \overline{\xi}^{(t)}(s, a) + (1 - \alpha) \frac{\widehat{Q}_\tau^{(t)}(s, a)}{\tau} \right) - \frac{2\gamma}{1 - \gamma} \left\| \widehat{Q}_\tau^{(t)} - \overline{Q}_\tau^{(t)} \right\|_\infty$$

$$\geq \alpha \left( \overline{Q}_\tau^{(t)}(s, a) - \tau \log \overline{\xi}^{(t)}(s, a) \right) - \frac{2\gamma + \eta\tau}{1 - \gamma} \left\| \widehat{Q}_\tau^{(t)} - \overline{Q}_\tau^{(t)} \right\|_\infty , \tag{123}$$

which gives

$$
\begin{aligned}
&-\min_{s,a}\left(\overline{Q}_\tau^{(t+1)}(s,a) - \tau\log\overline{\xi}^{(t+1)}(s,a)\right)\\
&\leq -\alpha\min_{s,a}\left(\overline{Q}_\tau^{(t)}(s,a) - \tau\log\overline{\xi}^{(t)}(s,a)\right) + \frac{2\gamma+\eta\tau}{1-\gamma}M\|u^{(t)}\|_\infty\\
&\leq \alpha\max\left\{0, \min_{s,a}\left(\overline{Q}_\tau^{(t)}(s,a) - \tau\log\overline{\xi}^{(t)}(s,a)\right)\right\} + \frac{2\gamma+\eta\tau}{1-\gamma}M\|u^{(t)}\|_\infty.
\end{aligned}
\tag{124}
$$

This establishes the bounds on $\Omega_4^{(t+1)}$ in (40).

## C.2 PROOF OF LEMMA 2

Let $f(\lambda)$ denote the characteristic function. In view of some direct calculations, we obtain

$$
\begin{aligned}
f(\lambda) = (\lambda-\alpha)\bigg\{ &\underbrace{(\lambda-\sigma\alpha)(\lambda-\sigma(1+\sigma b\eta))(\lambda-(1-\alpha)\gamma-\alpha)}_{=:f_0(\lambda)}\\
&- \frac{\eta\sigma^2}{1-\gamma}\underbrace{[S(\lambda-(1-\alpha)\gamma-\alpha) + \gamma cdM\eta + (1-\alpha)(2+\gamma)Mc\eta]}_{=:f_1(\lambda)}\bigg\}\\
&- \frac{\tau\eta^3\gamma}{(1-\gamma)^2}\cdot 2cdM\sigma^2,
\end{aligned}
\tag{125}
$$

where, for the notation simplicity, we let

$$
b := \frac{M\sqrt{N}}{1-\gamma},
\tag{126a}
$$

$$
c := \frac{MN}{1-\gamma} = \sqrt{N}b,
\tag{126b}
$$

$$
d := \frac{2\gamma+\eta\tau}{1-\gamma}.
\tag{126c}
$$

Note that among all these new notation we introduce, $S, d$ are dependent of $\eta$. To decouple the dependence, we give their upper bounds as follows

$$
d_0 := \frac{1+\gamma}{1-\gamma} \geq d,
\tag{127}
$$

$$
S_0 := M\sqrt{N}\left(2 + \sqrt{2N} + \frac{M\sqrt{N}}{\tau}\right) \geq S,
\tag{128}
$$

where (127) follows from $\eta \leq (1-\gamma)/\tau$, and (128) uses the fact that $\alpha \leq 1$ and $1-\alpha \leq 1$.

Let

$$
\lambda^\star := \max\left\{\frac{3+\sigma}{4}, \frac{1+(1-\alpha)\gamma+\alpha}{2}\right\}.
\tag{129}
$$

Since $\boldsymbol{A}(\rho)$ is a nonnegative matrix, by Perron-Frobenius Theorem (see Horn & Johnson (2012), Theorem 8.3.1), $\rho(\eta)$ is an eigenvalue of $\boldsymbol{A}(\rho)$. So to verify (46), it suffices to show that $f(\lambda) > 0$ for any $\lambda \in [\lambda^\star, \infty)$. To do so, in the following we first show that $f(\lambda^\star) > 0$, and then we prove that $f$ is non-decreasing on $[\lambda^\star, \infty)$.

- *Showing $f(\lambda^\star) > 0$.* We first lower bound $f_0(\lambda^\star)$. Since $\lambda^\star \geq \frac{3+\sigma}{4}$, we have

$$
\lambda^\star - \sigma(1+\sigma b\eta) \geq \frac{1-\sigma}{4},
\tag{130}
$$

and from $\lambda^\star \geq \frac{1+(1-\alpha)\gamma+\alpha}{2}$ we deduce

$$
\lambda^\star - (1-\alpha)\gamma - \alpha \geq \frac{(1-\gamma)(1-\alpha)}{2}
\tag{131}
$$

and

$$\lambda^\star > \frac{1+\alpha}{2} \,, \tag{132}$$

which gives

$$\lambda^\star - \sigma\alpha \geq \frac{1+\alpha}{2} - \sigma\alpha \,. \tag{133}$$

Combining (133), (130), (131), we have that

$$f_0(\lambda^\star) \geq \frac{1-\sigma}{8} \left( \frac{1+\alpha}{2} - \sigma\alpha \right) \eta\tau \,. \tag{134}$$

To continue, we upper bound $f_1(\lambda^\star)$ as follows.

$$
\begin{aligned}
f_1(\lambda^\star) &\leq S\tau\eta + \gamma cdM\eta + \frac{2+\gamma}{1-\gamma}cM\tau\eta^2 \\
&= \eta \left( \tau \left( S + \frac{2+\gamma}{1-\gamma}Mc\eta \right) + \gamma cdM \right) \,.
\end{aligned}
\tag{135}
$$

Plugging (134),(135) into (125) and using (132), we have

$$
\begin{aligned}
f(\lambda^\star) &> \frac{1-\alpha}{2} \left( f_0(\lambda^\star) - \frac{\eta\sigma^2}{1-\gamma}f_1(\lambda^\star) \right) - \frac{\tau\eta^3\gamma}{(1-\gamma)^2} \cdot 2cdM\sigma^2 \\
&\geq \frac{\tau\eta^2}{2(1-\gamma)} \left[ \frac{1-\sigma}{8}\tau \left( 1-\sigma + (1-\alpha)(\sigma - \tfrac{1}{2}) \right) - \frac{\eta\sigma^2}{1-\gamma} \left( \tau \left( S + \frac{2+\gamma}{1-\gamma}Mc\eta \right) + 5\gamma cdM \right) \right] \\
&= \frac{\tau\eta^2}{2(1-\gamma)} \left[ \frac{(1-\sigma)^2}{8}\tau - \frac{\eta}{1-\gamma} \left( S\tau\sigma^2 + \frac{2+\gamma}{1-\gamma}Mc\sigma^2\tau\eta + \tau^2 \left( \tfrac{1}{2} - \sigma^2 \right) \cdot \frac{1-\sigma}{8} + 5\gamma cdM\sigma^2 \right) \right] \\
&\geq \frac{\tau\eta^2}{2(1-\gamma)} \left[ \frac{(1-\sigma)^2}{8}\tau - \frac{\eta}{1-\gamma} \left( S_0\tau\sigma^2 + \frac{(1-\sigma)^2}{16}\tau^2 + (2+\gamma+5\gamma d_0)cM\sigma^2 \right) \right] \geq 0 \,,
\end{aligned}
$$

where the penultimate inequality uses $\frac{1}{2} - \sigma \leq \frac{1-\sigma}{2}$, and the last inequality follows from the definition of $\zeta$ (cf. (44)).

- *Proving $f$ is non-decreasing on $[\lambda^\star, \infty)$.* Note that

$$\eta \leq \zeta \leq \frac{(1-\gamma)(1-\sigma)^2}{8S_0\sigma^2} \,,$$

thus we have

$$\forall \lambda \geq \lambda^\star : \quad f_0'(\lambda) - \frac{\eta\sigma^2}{1-\gamma}f_1'(\lambda) \geq (\lambda - \sigma\alpha)(\lambda - \sigma(1+\sigma b\eta)) - \frac{\eta}{1-\gamma}S\sigma^2 \geq 0 \,,$$

which indicates that $f_0 - f_1$ is non-decreasing on $[\lambda^\star, \infty)$. Therefore, $f$ is non-decreasing on $[\lambda^\star, \infty)$.

### C.3   PROOF OF LEMMA 3

Note that bounding $u^{(t+1)}(s,a)$ is identical to the proof in Appendix C.1 and shall be omitted. The rest of the proof also follows closely that of Lemma 1, and we only highlight the differences due to approximation error for simplicity.

**Step 2: bound** $v^{(t+1)}(s,a) = \left\| \boldsymbol{T}^{(t+1)}(s,a) - \widehat{q}_\tau^{(t+1)}(s,a)\mathbf{1}_N \right\|_2$**.** Let $\boldsymbol{q}_\tau^{(t)} := \left( q_{\tau,1}^{\pi_1^{(t)}}, \cdots, q_{\tau,N}^{\pi_N^{(t)}} \right)^\top$. Similar to (100) we have

$$
\begin{aligned}
&\left\| \boldsymbol{T}^{(t+1)}(s,a) - \widehat{q}_\tau^{(t+1)}(s,a)\mathbf{1}_N \right\|_2 \\
&\leq \sigma \left\| \boldsymbol{T}^{(t)}(s,a) - \widehat{q}_\tau^{(t)}(s,a)\mathbf{1}_N \right\|_2 + \sigma \left\| \boldsymbol{q}_\tau^{(t+1)}(s,a) - \boldsymbol{q}_\tau^{(t)}(s,a) \right\|_2 \\
&\leq \sigma \left\| \boldsymbol{T}^{(t)}(s,a) - \widehat{q}_\tau^{(t)}(s,a)\mathbf{1}_N \right\|_2 + \sigma \left\| \boldsymbol{Q}_\tau^{(t+1)}(s,a) - \boldsymbol{Q}_\tau^{(t)}(s,a) \right\|_2 + 2\sigma \left\| e \right\|_2 \,. \tag{136}
\end{aligned}
$$

**Step 3: bound** $\left\|Q_\tau^\star - \tau \log \overline{\xi}^{(t+1)}\right\|_\infty$**.** In the context of inexact updates, (101) writes

$$\left\|Q_\tau^\star - \tau \log \overline{\xi}^{(t+1)}\right\|_\infty \le \alpha \left\|Q_\tau^\star - \tau \log \overline{\xi}^{(t)}\right\|_\infty + (1-\alpha)\left\|Q_\tau^\star - \overline{Q}_\tau^{(t)}\right\|_\infty + (1-\alpha)\left\|\overline{Q}_\tau^{(t)} - \widehat{q}_\tau^{(t)}\right\|_\infty .$$

For the last term, following a similar argument in (102) leads to

$$\left\|\overline{Q}_\tau^{(t)} - \widehat{q}_\tau^{(t)}\right\|_\infty = \left\| \frac{1}{N}\sum_{n=1}^N Q_{\tau,n}^{\pi_n^{(t)}} - \frac{1}{N}\sum_{n=1}^N Q_{\tau,n}^{\overline{\pi}^{(t)}} \right\|_\infty + \left\| \frac{1}{N}\sum_{n=1}^N \left( Q_{\tau,n}^{\pi_n^{(t)}} - q_{\tau,n}^{\pi_n^{(t)}} \right) \right\|_\infty$$

$$\le M \cdot \frac{1}{N}\sum_{n=1}^N \left\| \log \xi_n^{(t)} - \log \overline{\xi}^{(t)} \right\|_\infty + \frac{1}{N}\sum_{n=1}^N e_n$$

$$\le M \left\|u^{(t)}\right\|_\infty + \left\|e\right\|_\infty .$$

Combining the above two inequalities, we obtain

$$\left\|Q_\tau^\star - \tau \log \overline{\xi}^{(t+1)}\right\|_\infty \le \alpha \left\|Q_\tau^\star - \tau \log \overline{\xi}^{(t)}\right\|_\infty + (1-\alpha)\left\|Q_\tau^\star - \overline{Q}_\tau^{(t)}\right\|_\infty + (1-\alpha)\left( M\left\|u^{(t)}\right\|_\infty + \left\|e\right\|_\infty \right) . \tag{137}$$

**Step 4: bound** $\left\|\boldsymbol{Q}_\tau^{(t+1)}(s,a) - \boldsymbol{Q}_\tau^{(t)}(s,a)\right\|_2$**.** We remark that the bound established in (107) still holds in the inexact setting, with the same definition for $w^{(t)}$:

$$\left\|\boldsymbol{Q}_\tau^{(t+1)}(s,a) - \boldsymbol{Q}_\tau^{(t)}(s,a)\right\|_2 \le M\sqrt{N}\left\|w^{(t)}\right\|_\infty . \tag{138}$$

To deal with the approximation error, we rewrite (109) as

$$\left\| \boldsymbol{WT}^{(t)}(s,a) - \tau \log \boldsymbol{\xi}^{(t)}(s,a) - V_\tau^\star(s)\mathbf{1}_N \right\|_2$$

$$= \left\| \boldsymbol{WT}^{(t)}(s,a) - \tau \log \boldsymbol{\xi}^{(t)}(s,a) - \left( Q_\tau^\star(s,a) - \tau \log \pi_\tau^\star(a|s) \right)\mathbf{1}_N \right\|_2$$

$$\le \left\| \boldsymbol{WT}^{(t)}(s,a) - Q_\tau^\star(s,a)\mathbf{1}_N \right\|_2 + \tau \left\| \log \boldsymbol{\xi}^{(t)}(s,a) - \log \pi_\tau^\star(a|s)\mathbf{1}_N \right\|_2$$

$$\le \left\| \boldsymbol{WT}^{(t)}(s,a) - \widehat{q}_\tau(s,a)\mathbf{1}_N \right\|_2 + \left\| \widehat{q}_\tau(s,a)\mathbf{1}_N - Q_\tau^\star(s,a)\mathbf{1}_N \right\|_2$$

$$\quad + \tau \left\| \log \boldsymbol{\xi}^{(t)}(s,a) - \log \overline{\pi}^{(t)}(a|s)\mathbf{1}_N \right\|_2 + \tau \left\| \log \overline{\pi}^{(t)}(a|s)\mathbf{1}_N - \log \pi_\tau^\star(a|s)\mathbf{1}_N \right\|_2$$

$$\le \sigma \left\| \boldsymbol{T}^{(t)}(s,a) - \widehat{q}_\tau^{(t)}(s,a)\mathbf{1}_N \right\|_2 + \sqrt{N}\left|\widehat{q}_\tau^{(t)}(s,a) - Q_\tau^\star(s,a)\right|$$

$$\quad + \tau \left\| \log \boldsymbol{\xi}^{(t)}(s,a) - \log \overline{\pi}^{(t)}(a|s)\mathbf{1} \right\|_2 + \tau\sqrt{N}\left| \log \overline{\pi}^{(t)}(a|s) - \log \pi_\tau^\star(a|s) \right| , \tag{139}$$

where the second term can be upper-bounded by

$$\left|\widehat{q}_\tau^{(t)}(s,a) - Q_\tau^\star(s,a)\right| \le \left\|\widehat{Q}_\tau^{(t)} - \overline{Q}_\tau^{(t)}\right\|_\infty + \left\|\overline{Q}_\tau^{(t)} - Q_\tau^\star\right\|_\infty + \left\|\widehat{q}_\tau^{(t)}(s,a) - \widehat{Q}_\tau^{(t)}(s,a)\right\|_\infty$$

$$\le \left\|\widehat{Q}_\tau^{(t)} - \overline{Q}_\tau^{(t)}\right\|_\infty + \left\|\overline{Q}_\tau^{(t)} - Q_\tau^\star\right\|_\infty + \left\|e\right\|_\infty . \tag{140}$$

Combining (140), (139) and the established bounds in (108), (111), (113) leads to

$$w^{(t)}(s,a) \le \left( 2\alpha + (1-\alpha)\cdot\sqrt{2N} \right)\left\|u^{(t)}\right\|_\infty + \frac{1-\alpha}{\tau}\left\|v^{(t)}\right\|_\infty$$

$$+ \frac{1-\alpha}{\tau}\cdot\sqrt{N}\left( \left\|\widehat{Q}_\tau^{(t)} - \overline{Q}_\tau^{(t)}\right\|_\infty + \left\|\overline{Q}_\tau^{(t)} - Q_\tau^\star\right\|_\infty + \left\|e\right\|_\infty \right) + \frac{1-\alpha}{\tau}\cdot 2\sqrt{N}\left\|Q_\tau^\star - \tau \log \overline{\xi}^{(t)}\right\|_\infty .$$

Combining the above inequality with (138) and (136) gives

$$\left\|v^{(t+1)}\right\|_\infty \le \sigma\left( 1 + \frac{\eta M\sqrt{N}}{1-\gamma}\sigma \right)\left\|v^{(t)}\right\|_\infty + \sigma M\sqrt{N}\Bigg\{ \left( 2\alpha + (1-\alpha)\cdot\sqrt{2N} + \frac{1-\alpha}{\tau}\cdot\sqrt{N}M \right)\left\|u^{(t)}\right\|_\infty$$

$$+ \frac{1-\alpha}{\tau}\cdot\sqrt{N}\left( \left\|\overline{Q}_\tau^{(t)} - Q_\tau^\star\right\|_\infty + \left\|e\right\|_\infty \right) + \frac{1-\alpha}{\tau}\cdot 2\sqrt{N}\left\|Q_\tau^\star - \tau \log \overline{\xi}^{(t)}\right\|_\infty \Bigg\} + 2\sigma\sqrt{N}\left\|e\right\|_\infty . \tag{141}$$

**Step 5: bound** $\left\|\overline{Q}_\tau^{(t+1)} - Q_\tau^\star\right\|_\infty$**.** It is straightforward to verify that (120) applies to the inexact updates as well:

$$\left\|Q_\tau^\star - \overline{Q}_\tau^{(t+1)}\right\|_\infty \le \gamma \left\|Q_\tau^\star - \tau \log \overline{\xi}^{(t+1)}\right\|_\infty + \gamma \left(-\min_{s,a}\left(\overline{Q}_\tau^{(t+1)}(s,a) - \tau \log \overline{\xi}^{(t+1)}(s,a)\right)\right) .$$

Plugging the above inequality into (137) and (141) establishes the bounds on $\Omega_3^{(t+1)}$ and $\Omega_2^{(t+1)}$ in (58), respectively.

**Step 6: bound** $-\min_{s,a}\left(\overline{Q}_\tau^{(t+1)}(s,a) - \tau \log \overline{\xi}^{(t+1)}(s,a)\right)$**.** We obtain the following lemma by interpreting the approximation error $e$ as part of the consensus error $\left\|\widehat{Q}_\tau^{(t)} - \overline{Q}_\tau^{(t)}\right\|_\infty$ in Lemma 9.

**Lemma 10** (inexact version of Lemma 9). *Suppose* $0 < \eta \le (1-\gamma)/\tau$. *For any state-action pair* $(s_0, a_0) \in \mathcal{S} \times \mathcal{A}$, *one has*

$$\overline{V}_\tau^{(t+1)}(s_0) - \overline{V}_\tau^{(t)}(s_0) \ge \frac{1}{\eta} \mathop{\mathbb{E}}_{s \sim d_{s_0}^{\overline{\pi}^{(t+1)}}} \left[\alpha \mathsf{KL}\left(\overline{\pi}^{(t+1)}(\cdot|s_0) \,\|\, \overline{\pi}^{(t)}(\cdot|s_0)\right) + \mathsf{KL}\left(\overline{\pi}^{(t)}(\cdot|s_0) \,\|\, \overline{\pi}^{(t+1)}(\cdot|s_0)\right)\right]$$

$$- \frac{2}{1-\gamma}\left(\left\|\widehat{Q}_\tau^{(t)} - \overline{Q}_\tau^{(t)}\right\|_\infty + \|e\|_\infty\right) , \tag{142}$$

$$\overline{Q}_\tau^{(t+1)}(s_0, a_0) - \overline{Q}_\tau^{(t)}(s_0, a_0) \ge -\frac{2\gamma}{1-\gamma}\left(\left\|\widehat{Q}_\tau^{(t)} - \overline{Q}_\tau^{(t)}\right\|_\infty + \|e\|_\infty\right) . \tag{143}$$

Using (143), we have

$$\overline{Q}_\tau^{(t+1)}(s,a) - \tau\left(\alpha \log \overline{\xi}^{(t)}(s,a) + (1-\alpha)\frac{\widehat{Q}_\tau^{(t)}(s,a)}{\tau}\right)$$

$$\ge \overline{Q}_\tau^{(t)}(s,a) - \tau\left(\alpha \log \overline{\xi}^{(t)}(s,a) + (1-\alpha)\frac{\widehat{Q}_\tau^{(t)}(s,a)}{\tau}\right) - \frac{2\gamma}{1-\gamma}\left(\left\|\widehat{Q}_\tau^{(t)} - \overline{Q}_\tau^{(t)}\right\|_\infty + \|e\|_\infty\right)$$

$$\ge \alpha\left(\overline{Q}_\tau^{(t)}(s,a) - \tau \log \overline{\xi}^{(t)}(s,a)\right) - \frac{2\gamma + \eta\tau}{1-\gamma}\left\|\widehat{Q}_\tau^{(t)} - \overline{Q}_\tau^{(t)}\right\|_\infty - \frac{2\gamma}{1-\gamma}\|e\|_\infty , \tag{144}$$

which gives

$$-\min_{s,a}\left(\overline{Q}_\tau^{(t+1)}(s,a) - \tau \log \overline{\xi}^{(t+1)}(s,a)\right)$$
$$\le -\alpha \min_{s,a}\left(\overline{Q}_\tau^{(t)}(s,a) - \tau \log \overline{\xi}^{(t)}(s,a)\right) + \frac{2\gamma + \eta\tau}{1-\gamma}M\left\|u^{(t)}\right\|_\infty + \frac{2\gamma}{1-\gamma}\|e\|_\infty . \tag{145}$$

### C.4 Proof of Lemma 4

**Step 1: bound** $u^{(t+1)}(s,a) = \left\|\log \boldsymbol{\xi}^{(t+1)}(s,a) - \log \overline{\xi}^{(t+1)}(s,a)\mathbf{1}_N\right\|_2$**.** Following the same strategy in establishing (99), we have

$$\left\|\log \boldsymbol{\xi}^{(t+1)}(s,a) - \log \overline{\xi}^{(t+1)}(s,a)\mathbf{1}_N\right\|_2$$

$$= \left\|\left(\boldsymbol{W}\log \boldsymbol{\xi}^{(t)}(s,a) - \log \overline{\xi}^{(t)}(s,a)\mathbf{1}_N\right) + \frac{\eta}{1-\gamma}\left(\boldsymbol{W}\boldsymbol{T}^{(t)}(s,a) - \widehat{Q}^{(t)}(s,a)\mathbf{1}_N\right)\right\|_2$$

$$\le \sigma\left\|\log \boldsymbol{\xi}^{(t)}(s,a) - \log \overline{\xi}^{(t)}(s,a)\mathbf{1}_N\right\|_2 + \frac{\eta}{1-\gamma}\sigma\left\|\boldsymbol{T}^{(t)}(s,a) - \widehat{Q}^{(t)}(s,a)\mathbf{1}_N\right\|_2 , \tag{146}$$

or equivalently

$$\left\|u^{(t+1)}\right\|_\infty \le \sigma\left\|u^{(t)}\right\|_\infty + \frac{\eta}{1-\gamma}\sigma\left\|v^{(t)}\right\|_\infty . \tag{147}$$

**Step 2: bound** $v^{(t+1)}(s,a) = \left\|\boldsymbol{T}^{(t+1)}(s,a) - \widehat{Q}^{(t+1)}(s,a)\mathbf{1}_N\right\|_2$**.** In the same vein of establishing (100), we have

$$\left\|\boldsymbol{T}^{(t+1)}(s,a) - \widehat{Q}^{(t+1)}(s,a)\mathbf{1}_N\right\|_2$$

$$\le \sigma\left\|\boldsymbol{T}^{(t)}(s,a) - \widehat{Q}^{(t)}(s,a)\mathbf{1}_N\right\|_2 + \sigma\left\|\boldsymbol{Q}^{(t+1)}(s,a) - \boldsymbol{Q}^{(t)}(s,a)\right\|_2 , \tag{148}$$

The term $\left\|\boldsymbol{Q}^{(t+1)}(s,a) - \boldsymbol{Q}^{(t)}(s,a)\right\|_2$ can be bounded in a similar way in (107):

$$\left\|\boldsymbol{Q}^{(t+1)}(s,a) - \boldsymbol{Q}^{(t)}(s,a)\right\|_2 \le \frac{(1+\gamma)\gamma}{(1-\gamma)^2}\sqrt{N}\big\|w_0^{(t)}\big\|_\infty, \tag{149}$$

where the coefficient $\frac{(1+\gamma)\gamma}{(1-\gamma)^2}$ comes from $M$ in Lemma 8 when $\tau = 0$, and $w_0^{(t)} \in \mathbb{R}^{|\mathcal{S}||\mathcal{A}|}$ is defined as

$$\forall (s,a) \in \mathcal{S} \times \mathcal{A}: \quad w_0^{(t)}(s,a) := \left\|\log\boldsymbol{\xi}^{(t+1)}(s,a) - \log\boldsymbol{\xi}^{(t)}(s,a) - \frac{\eta}{1-\gamma}V^\star(s)\mathbf{1}_N\right\|_2. \tag{150}$$

It remains to bound $\big\|w_0^{(t)}\big\|_\infty$. Towards this end, we rewrite (108) as

$$
\begin{aligned}
&w_0^{(t)}(s,a)\\
&= \left\|\boldsymbol{W}\left(\log\boldsymbol{\xi}^{(t)}(s,a) + \frac{\eta}{1-\gamma}\boldsymbol{T}^{(t)}(s,a)\right) - \log\boldsymbol{\xi}^{(t)}(s,a) - \frac{\eta}{1-\gamma}V^\star(s)\mathbf{1}_N\right\|_2\\
&= \left\|(\boldsymbol{W}-\boldsymbol{I})\left(\log\boldsymbol{\xi}^{(t)}(s,a) - \log\overline{\xi}^{(t)}(s,a)\mathbf{1}_N\right) + \frac{\eta}{1-\gamma}\left(\boldsymbol{W}\boldsymbol{T}^{(t)}(s,a) - V^\star(s)\mathbf{1}_N\right)\right\|_2\\
&\le 2\left\|\log\boldsymbol{\xi}^{(t)}(s,a) - \log\overline{\xi}^{(t)}(s,a)\mathbf{1}_N\right\|_2 + \frac{\eta}{1-\gamma}\left\|\boldsymbol{W}\boldsymbol{T}^{(t)}(s,a) - V^\star(s)\mathbf{1}_N\right\|_2\\
&\le 2\left\|\log\boldsymbol{\xi}^{(t)}(s,a) - \log\overline{\xi}^{(t)}(s,a)\mathbf{1}_N\right\|_2 + \frac{\eta}{1-\gamma}\left\|\boldsymbol{W}\boldsymbol{T}^{(t)}(s,a) - \widehat{Q}^{(t)}(s,a)\mathbf{1}_N\right\|_2 + \frac{\eta}{1-\gamma}\cdot\sqrt{N}\big|\widehat{Q}^{(t)}(s,a) - V^\star(s)\big|.
\end{aligned}
\tag{151}
$$

Note that it holds for all $(s,a) \in \mathcal{S} \times \mathcal{A}$:

$$\big|\widehat{Q}^{(t)}(s,a) - V^\star(s)\big| \le \frac{1}{1-\gamma}$$

since $\widehat{Q}^{(t)}(s,a)$ and $V^\star(s)$ are both in $[0, 1/(1-\gamma)]$. This along with (151) gives

$$w_0^{(t)}(s,a) \le 2\big\|u^{(t)}\big\|_\infty + \frac{\eta}{1-\gamma}\big\|v^{(t)}\big\|_\infty + \frac{\eta\sqrt{N}}{(1-\gamma)^2}.$$

Combining the above inequality with (149) and (148), we arrive at

$$\big\|v^{(t+1)}\big\|_\infty \le \sigma\left(1 + \frac{(1+\gamma)\gamma\sqrt{N}\eta}{(1-\gamma)^3}\sigma\right)\big\|v^{(t)}\big\|_\infty + \frac{(1+\gamma)\gamma}{(1-\gamma)^2}\sqrt{N}\sigma\left\{2\big\|u^{(t)}\big\|_\infty + \frac{\eta}{(1-\gamma)^2}\cdot\sqrt{N}\right\}. \tag{152}$$

**Step 3: establish the descent equation.** The following lemma characterizes the improvement in $\phi^{(t)}(\eta)$ for every iteration of Algorithm 1, with the proof postponed to Appendix D.4.

**Lemma 11** (Performance improvement of exact FedNPG). *For all starting state distribution $\rho \in \Delta(\mathcal{S})$, we have the iterates of FedNPG satisfy*

$$\phi^{(t+1)}(\eta) \le \phi^{(t)}(\eta) + \frac{2\eta}{(1-\gamma)^2}\big\|\widehat{Q}^{(t)} - \overline{Q}^{(t)}\big\|_\infty - \eta\left(V^\star(\rho) - \overline{V}^{(t)}(\rho)\right), \tag{153}$$

*where*

$$\phi^{(t)}(\eta) := \mathbb{E}_{s\sim d_\rho^{\pi^\star}}\left[\mathsf{KL}\big(\pi^\star(\cdot|s)\,\|\,\overline{\pi}^{(t)}(\cdot|s)\big)\right] - \frac{\eta}{1-\gamma}\overline{V}^{(t)}(d_\rho^{\pi^\star}), \quad \forall t \ge 0. \tag{154}$$

It remains to control the term $\big\|\overline{Q}^{(t)} - \widehat{Q}^{(t)}\big\|_\infty$. Similar to (102), for all $t \ge 0$, we have

$$
\begin{aligned}
\big\|\overline{Q}^{(t)} - \widehat{Q}^{(t)}\big\|_\infty &= \left\|\frac{1}{N}\sum_{n=1}^N Q_n^{\pi_n^{(t)}} - \frac{1}{N}\sum_{n=1}^N Q_n^{\overline{\pi}^{(t)}}\right\|_\infty\\
&\overset{(a)}{\le} \frac{(1+\gamma)\gamma}{(1-\gamma)^2}\cdot\frac{1}{N}\sum_{n=1}^N\big\|\log\xi_n^{(t)} - \log\overline{\xi}^{(t)}\big\|_\infty\\
&\overset{(b)}{\le} \frac{(1+\gamma)\gamma}{(1-\gamma)^2}\big\|u^{(t)}\big\|_\infty,
\end{aligned}
\tag{155}
$$

where (a) invokes Lemma 8 with $\tau = 0$ and (b) stems from the definition of $u^{(t)}$. This along with (153) gives

$$\phi^{(t+1)}(\eta) \leq \phi^{(t)}(\eta) + \frac{2(1+\gamma)\gamma}{(1-\gamma)^4}\eta\|u^{(t)}\|_\infty - \eta\left(V^\star(\rho) - \overline{V}^{(t)}(\rho)\right).$$

### C.5 Proof of Lemma 5

The bound on $u^{(t+1)}(s, a)$ is already established in Step 1 in Appendix C.1 and shall be omitted. As usual we only highlight the key differences with the proof of Lemma 4 due to approximation error.

**Step 1: bound** $v^{(t+1)}(s, a) = \left\|\mathbf{T}^{(t+1)}(s, a) - \widehat{q}^{(t+1)}(s, a)\mathbf{1}_N\right\|_2$. Let $\mathbf{q}^{(t)} := \left(q_1^{\pi_1^{(t)}}, \cdots, q_N^{\pi_N^{(t)}}\right)^\top$.
From (83), we have

$$\left\|\mathbf{T}^{(t+1)}(s, a) - \widehat{q}^{(t+1)}(s, a)\mathbf{1}_N\right\|_2$$
$$= \left\|\mathbf{W}\left(\mathbf{T}^{(t)}(s, a) + \mathbf{q}^{(t+1)}(s, a) - \mathbf{q}^{(t)}(s, a)\right) - \widehat{q}^{(t+1)}(s, a)\mathbf{1}_N\right\|_2$$
$$= \left\|\left(\mathbf{W}\mathbf{T}^{(t)}(s, a) - \widehat{q}^{(t)}(s, a)\mathbf{1}_N\right) + \mathbf{W}\left(\mathbf{q}^{(t+1)}(s, a) - \mathbf{q}^{(t)}(s, a)\right) + \left(\widehat{q}^{(t)}(s, a) - \widehat{q}^{(t+1)}(s, a)\right)\mathbf{1}_N\right\|_2$$
$$\leq \sigma\left\|\mathbf{T}^{(t)}(s, a) - \widehat{q}^{(t)}(s, a)\mathbf{1}_N\right\|_2 + \sigma\left\|\left(\mathbf{q}^{(t+1)}(s, a) - \mathbf{q}^{(t)}(s, a)\right) + \left(\widehat{q}^{(t)}(s, a) - \widehat{q}^{(t+1)}(s, a)\right)\mathbf{1}_N\right\|_2$$
$$\leq \sigma\left\|\mathbf{T}^{(t)}(s, a) - \widehat{q}^{(t)}(s, a)\mathbf{1}_N\right\|_2 + \sigma\left\|\mathbf{q}^{(t+1)}(s, a) - \mathbf{q}^{(t)}(s, a)\right\|_2$$
$$\leq \sigma\left\|\mathbf{T}^{(t)}(s, a) - \widehat{q}^{(t)}(s, a)\mathbf{1}_N\right\|_2 + \sigma\left\|\mathbf{Q}^{(t+1)}(s, a) - \mathbf{Q}^{(t)}(s, a)\right\|_2 + 2\sigma\sqrt{N}\|\mathbf{e}\|_\infty. \quad (156)$$

Note that (149) still holds for inexact FedNPG:

$$\left\|\mathbf{Q}^{(t+1)}(s, a) - \mathbf{Q}^{(t)}(s, a)\right\|_2 \leq \frac{(1+\gamma)\gamma}{(1-\gamma)^2}\sqrt{N}\left\|w_0^{(t)}\right\|_\infty, \quad (157)$$

where $w_0^{(t)}$ is defined in (150). We rewrite (151), the bound on $w_0^{(t)}(s, a)$, as

$$w_0^{(t)}(s, a) \leq 2\left\|\log\boldsymbol{\xi}^{(t)}(s, a) - \log\overline{\xi}^{(t)}(s, a)\mathbf{1}_N\right\|_2$$
$$+ \frac{\eta}{1-\gamma}\left\|\mathbf{T}^{(t)}(s, a) - \widehat{q}^{(t)}(s, a)\mathbf{1}_N\right\|_2 + \frac{\eta\sigma}{1-\gamma} \cdot \sqrt{N}\left|\widehat{q}^{(t)}(s, a) - V^\star(s)\right|. \quad (158)$$

With the following bound

$$\forall(s, a) \in \mathcal{S} \times \mathcal{A}: \quad \left|\widehat{q}^{(t)}(s, a) - V^\star(s)\right| \leq \left\|\widehat{q}^{(t)} - \overline{Q}^{(t)}\right\|_\infty + \frac{1}{1-\gamma}$$

in mind, we write (151) as

$$w_0^{(t)}(s, a) \leq 2\|u^{(t)}\|_\infty + \frac{\eta\sigma}{1-\gamma}\|v^{(t)}\|_\infty + \frac{\eta}{1-\gamma} \cdot \sqrt{N}\left(\left\|\widehat{q}^{(t)} - \overline{q}^{(t)}\right\|_\infty + \frac{1}{1-\gamma}\right).$$

Putting all pieces together, we obtain

$$\left\|v^{(t+1)}\right\|_\infty \leq \sigma\left(1 + \frac{(1+\gamma)\gamma\sqrt{N}\eta}{(1-\gamma)^3}\sigma\right)\left\|v^{(t)}\right\|_\infty$$
$$+ \frac{(1+\gamma)\gamma}{(1-\gamma)^2}\sqrt{N}\sigma\left\{2\|u^{(t)}\|_\infty + \frac{\eta\sqrt{N}}{(1-\gamma)^2} + \frac{\eta\sqrt{N}}{1-\gamma}\|\mathbf{e}\|_\infty\right\} \quad (159)$$
$$+ 2\sigma\sqrt{N}\|\mathbf{e}\|_\infty.$$

**Step 2: establish the descent equation.** Note that Lemma 11 directly applies by replacing $\widehat{Q}^{(t)}$ with $\widehat{q}^{(t)}$:

$$\phi^{(t+1)}(\eta) \leq \phi^{(t)}(\eta) + \frac{2\eta}{(1-\gamma)^2}\left\|\widehat{q}^{(t)} - \overline{Q}^{(t)}\right\|_\infty - \eta\left(V^\star(\rho) - \overline{V}^{(t)}(\rho)\right).$$

To bound the middle term, for all $t \geq 0$, we have

$$
\begin{aligned}
\left\| \overline{Q}^{(t)} - \widehat{q}^{(t)} \right\|_\infty &= \left\| \frac{1}{N} \sum_{n=1}^{N} Q_n^{\pi_n^{(t)}} - \frac{1}{N} \sum_{n=1}^{N} Q_n^{\overline{\pi}^{(t)}} \right\|_\infty + \frac{1}{N} \left\| \sum_{n=0}^{N} \left( q_n^{\pi_n^{(t)}} - Q_n^{\pi_n^{(t)}} \right) \right\|_\infty \\
&\leq \frac{(1+\gamma)\gamma}{(1-\gamma)^2} \cdot \frac{1}{N} \sum_{n=1}^{N} \left\| \log \xi_n^{(t)} - \log \overline{\xi}^{(t)} \right\|_\infty + \frac{1}{N} \sum_{n=1}^{N} e_n \\
&\leq \frac{(1+\gamma)\gamma}{(1-\gamma)^2} \left\| u^{(t)} \right\|_\infty + \| \boldsymbol{e} \|_\infty .
\end{aligned}
\tag{160}
$$

Hence, (89) is established by combining the above two inequalities.

## D  PROOF OF AUXILIARY LEMMAS

### D.1  PROOF OF LEMMA 6

The first claim is easily verified as $\log \xi_n^{(t)}(s, \cdot)$ always deviate from $\log \pi_n^{(t)}(\cdot|s)$ by a global constant shift, as long as it holds for $t = 0$:

$$
\begin{aligned}
\log \xi_n^{(t+1)}(s, \cdot) &= \sum_{n'=1}^{N} [W]_{n,n'} \left( \alpha \log \xi_{n'}^{(t)}(s, \cdot) + (1-\alpha) T_n^{(t)}(s, \cdot)/\tau \right) \\
&= \alpha \sum_{n'=1}^{N} [W]_{n,n'} \left( \alpha \left( \log \pi_{n'}^{(t)}(s, \cdot) + c_{n'}^{(t)}(s) \mathbf{1}_{|\mathcal{A}|} \right) + (1-\alpha) T_n^{(t)}(s, \cdot)/\tau \right) \\
&= \alpha \sum_{n'=1}^{N} [W]_{n,n'} \left( \alpha \log \pi_{n'}^{(t)}(s, \cdot) + (1-\alpha) T_n^{(t)}(s, \cdot)/\tau \right) - \log z_n^{(t)}(s) \mathbf{1}_{|\mathcal{A}|} + c_n^{(t+1)}(s) \mathbf{1}_{|\mathcal{A}|} \\
&= \log \pi_n^{(t+1)}(\cdot|s) + c_n^{(t+1)}(s) \mathbf{1}_{|\mathcal{A}|},
\end{aligned}
$$

where $z_n^{(t)}$ is the normalization term (cf. line 5, Algorithm 2) and $\{c_n^{(t)}(s)\}$ are some constants. To prove the second claim, $\forall t \geq 0, \forall (s, a) \in \mathcal{S} \times \mathcal{A}$, let

$$
\overline{T}^{(t)}(s, a) := \frac{1}{N} \mathbf{1}^\top \boldsymbol{T}^{(t)}(s, a) .
\tag{161}
$$

Taking inner product with $\frac{1}{N}\mathbf{1}$ for both sides of $(U_T)$ and using the double stochasticity property of $\boldsymbol{W}$, we get

$$
\overline{T}^{(t+1)}(s, a) = \overline{T}^{(t)}(s, a) + \widehat{Q}_\tau^{(t+1)}(s, a) - \widehat{Q}_\tau^{(t)}(s, a) .
\tag{162}
$$

By the choice of $\boldsymbol{T}^{(0)}$ (line 2 of Algorithm 2), we have $\overline{T}^{(0)} = \widehat{Q}_\tau^{(0)}$ and hence by induction

$$
\forall t \geq 0 : \quad \overline{T}^{(t)} = \widehat{Q}_\tau^{(t)} .
\tag{163}
$$

This implies

$$
\begin{aligned}
\log \overline{\xi}^{(t+1)}(s, a) - \alpha \log \overline{\xi}^{(t)}(s, a) &= (1-\alpha) \widehat{Q}_\tau^{(t)}(s, a)/\tau \\
&= (1-\alpha) \overline{T}^{(t)}(s, a)/\tau \\
&= \frac{1}{N} \mathbf{1}^\top \log \boldsymbol{\xi}^{(t+1)}(s, a) - \alpha \frac{1}{N} \mathbf{1}^\top \log \boldsymbol{\xi}^{(t)}(s, a).
\end{aligned}
$$

Therefore, to prove (94), it suffices to verify the claim for $t = 0$:

$$
\frac{1}{N} \mathbf{1}^\top \log \boldsymbol{\xi}^{(0)}(s, a) = \log \| \exp \left( Q_\tau^\star(s, \cdot)/\tau \right) \|_1 + \frac{1}{N} \mathbf{1}^\top \log \boldsymbol{\pi}^{(0)}(a|s) - \log \left\| \exp \left( \frac{1}{N} \sum_{n=1}^{N} \log \pi_n^{(0)}(\cdot|s) \right) \right\|_1
$$

$$
= \log \| \exp \left( Q_\tau^\star(s, \cdot)/\tau \right) \|_1 + \log \overline{\pi}^{(0)}(a|s) = \log \overline{\xi}^{(0)}(s, a) .
$$

By taking logarithm over both sides of the definition of $\overline{\pi}^{(t+1)}$ (cf. $(U_\pi)$), we get

$$
\log \overline{\pi}^{(t+1)}(a|s) = \alpha \log \overline{\pi}^{(t)}(a|s) + (1-\alpha) \widehat{Q}^{(t)}(s, a)/\tau - z^{(t)}(s)
\tag{164}
$$

for some constant $z^{(t)}(s)$, which deviate from the update rule of $\log \overline{\xi}^{(t+1)}$ by a global constant shift and hence verifies (95).

## D.2 PROOF OF LEMMA 8

For notational simplicity, we let $Q_\tau^{\theta'}$ and $Q_\tau^\theta$ denote $Q_\tau^{\pi_{\theta'}}$ and $Q_\tau^{\pi_\theta}$, respectively. From (7a) we immediately know that to bound $\left\| Q_\tau^{\theta'} - Q_\tau^\theta \right\|_\infty$, it suffices to control $\left| V_\tau^\theta(s) - V_\tau^{\theta'}(s) \right|$ for each $s \in \mathcal{S}$. By (4) we have

$$\left| V_\tau^\theta(s) - V_\tau^{\theta'}(s) \right| \le \left| V^\theta(s) - V^{\theta'}(s) \right| + \tau \left| \mathcal{H}(s, \pi_\theta) - \mathcal{H}(s, \pi_{\theta'}) \right|, \tag{165}$$

so in the following we bound both terms in the RHS of (165).

**Step 1: bounding** $\left| \mathcal{H}(s, \pi_\theta) - \mathcal{H}(s, \pi_{\theta'}) \right|$. We first bound $\left| \mathcal{H}(s, \pi_\theta) - \mathcal{H}(s, \pi_{\theta'}) \right|$ using the idea in the proof of Lemma 14 in Mei et al. (2020). We let

$$\theta_t = \theta + t(\theta' - \theta), \quad \forall t \in \mathbb{R}, \tag{166}$$

and let $h_t \in \mathbb{R}^{|\mathcal{S}|}$ be

$$\forall s \in \mathcal{S}: \quad h_t(s) := -\sum_{a \in \mathcal{A}} \pi_{\theta_t}(a|s) \log \pi_{\theta_t}(a|s). \tag{167}$$

Note that $\|h_t\|_\infty \le \log |\mathcal{A}|$. We also denote $H_t : \mathcal{S} \to \mathbb{R}^{|\mathcal{A}| \times |\mathcal{A}|}$ by:

$$\forall s \in \mathcal{S}: \quad H_t(s) := \left. \frac{\partial \pi_\theta(\cdot|s)}{\partial \theta} \right|_{\theta = \theta_t} = \mathrm{diag}\{\pi_{\theta_t}(\cdot|s)\} - \pi_{\theta_t}(\cdot|s) \pi_{\theta_t}(\cdot|s)^\top, \tag{168}$$

then we have

$$\forall s \in \mathcal{S}: \quad \left| \frac{dh_t(s)}{dt} \right| = \left| \left\langle \frac{\partial h_t(s)}{\partial \theta_t(\cdot|s)}, \theta'(s, \cdot) - \theta(s, \cdot) \right\rangle \right|$$
$$= \left| \langle H_t(s) \log \pi_{\theta_t}(\cdot|s), \theta'(s, \cdot) - \theta(s, \cdot) \rangle \right|$$
$$\le \| H_t(s) \log \pi_{\theta_t}(\cdot|s) \|_1 \| \theta'(s, \cdot) - \theta(s, \cdot) \|_\infty, \tag{169}$$

where $\frac{\partial h_t(s)}{\partial \theta_t(\cdot|s)}$ stands for $\frac{\partial h_t(s)}{\partial \theta(\cdot|s)} \big|_{\theta = \theta_t}$. The first term in (169) is further upper bounded as

$$\| H_t(s) \log \pi_{\theta_t}(\cdot|s) \|_1 = \sum_{a \in \mathcal{A}} \pi_{\theta_t}(a|s) \left| \log \pi_{\theta_t}(a|s) - \pi_{\theta_t}(\cdot|s)^\top \log \pi_{\theta_t}(\cdot|s) \right|$$
$$\le \sum_{a \in \mathcal{A}} \pi_{\theta_t}(a|s) \left( |\log \pi_{\theta_t}(a|s)| + \left| \pi_{\theta_t}(\cdot|s)^\top \log \pi_{\theta_t}(\cdot|s) \right| \right)$$
$$= -2 \sum_{a \in \mathcal{A}} \pi_{\theta_t}(a, s) \log \pi_{\theta_t}(a|s) \le 2 \log |\mathcal{A}|.$$

By Lagrange mean value theorem, there exists $t \in (0, 1)$ such that

$$|h_1(s) - h_0(s)| = \left| \frac{dh_t(s)}{dt} \right| \le 2 \log |\mathcal{A}| \| \theta'(s, \cdot) - \theta(s, \cdot) \|_\infty,$$

where the inequality follows from (169) and the above inequality. Combining (5) with the above inequality, we arrive at

$$\left| \mathcal{H}(s, \pi_\theta) - \mathcal{H}(s, \pi_{\theta'}) \right| \le \frac{2 \log |\mathcal{A}|}{1 - \gamma} \| \theta' - \theta \|_\infty. \tag{170}$$

**Step 2: bounding** $\left| V^\theta(s) - V^{\theta'}(s) \right|$. Similar to the previous proof, we bound $\left| V^\theta(s) - V^{\theta'}(s) \right|$ by bounding $\left| \frac{dV^{\theta_t}}{dt}(s) \right|$. By Bellman's consistency equation, the value function of $\pi_{\theta_t}$ is given by

$$V^{\theta_t}(s) = \sum_{a \in \mathcal{A}} \pi_{\theta_t}(a|s) r(s, a) + \gamma \sum_a \pi_{\theta_\alpha}(a|s) \sum_{s' \in \mathcal{S}} \mathcal{P}(s'|s, a) V^{\theta_t}(s'),$$

which can be represented in a matrix-vector form as

$$V^{\theta_t}(s) = e_s^\top M_t r_t, \tag{171}$$

where $e_s \in \mathbb{R}^{|\mathcal{S}|}$ is a one-hot vector whose $s$-th entry is 1,

$$M_t := (I - \gamma P_t)^{-1},\tag{172}$$

with $P_t \in \mathbb{R}^{|\mathcal{S}| \times |\mathcal{S}|}$ denoting the induced state transition matrix by $\pi_{\theta_t}$

$$P_t(s, s') = \sum_{a \in \mathcal{A}} \pi_{\theta_t}(a|s)\mathcal{P}(s'|s, a),\tag{173}$$

and $r_t \in \mathbb{R}^{|\mathcal{S}|}$ is given by

$$\forall s \in \mathcal{S}: \quad r_t(s) := \sum_{a \in \mathcal{A}} \pi_{\theta_t}(a|s)r(s, a).\tag{174}$$

Taking derivative w.r.t. $t$ in (171), we obtain (**?**)

$$\frac{dV^{\theta_t}(s)}{dt} = \gamma \cdot e_s^\top M_t \frac{dP_t}{dt} M_t r_t + e_s^\top M_t \frac{dr_t}{dt}.\tag{175}$$

We now calculate each term respectively.

- For the first term, it follows that

$$\left| \gamma \cdot e_s^\top M_t \frac{dP_t}{dt} M_t r_t \right| \le \gamma \left\| M_t \frac{dP_t}{dt} M_t r_t \right\|_\infty$$

$$\le \frac{\gamma}{1-\gamma} \left\| \frac{dP_t}{dt} M_t r_t \right\|_\infty$$

$$\le \frac{2\gamma}{1-\gamma} \| M_t r_t \|_\infty \| \theta' - \theta \|_\infty \tag{176}$$

$$\le \frac{2\gamma}{(1-\gamma)^2} \| r_t \|_\infty \| \theta' - \theta \|_\infty$$

$$\le \frac{2\gamma}{(1-\gamma)^2} \| \theta' - \theta \|_\infty.\tag{177}$$

where the second and fourth lines use the fact $\|M_t\|_1 \le 1/(1-\gamma)$ (Li et al., 2020b, Lemma 10), and the last line follow from

$$\|r_t\|_\infty = \max_{s \in \mathcal{S}} \left| \sum_{a \in \mathcal{A}} \pi_{\theta_t}(a|s)r(s, a) \right| \le 1.$$

We defer the proof of (176) to the end of proof.

- For the second term, it follows that

$$\left| e_s^\top M_t \frac{dr_t}{dt} \right| \le \frac{1}{1-\gamma} \left\| \frac{dr_t}{dt} \right\|_\infty \le \frac{1}{1-\gamma} \| \theta' - \theta \|_\infty.\tag{178}$$

where the first inequality follows again from $\|M_t\|_1 \le 1/(1-\gamma)$, and the second inequality follows from

$$\left\| \frac{dr_t}{dt} \right\|_\infty = \max_{s \in \mathcal{S}} \left| \frac{dr_t(s)}{dt} \right| = \max_{s \in \mathcal{S}} \left| \left\langle \frac{\partial \pi_{\theta_t}(\cdot|s)^\top r(s, \cdot)}{\partial \theta_t(s, \cdot)}, \theta'(s, \cdot) - \theta(s, \cdot) \right\rangle \right|$$

$$\le \max_{s \in \mathcal{S}} \left\| \frac{\partial \pi_{\theta_t}(\cdot|s)^\top}{\partial \theta_t(s, \cdot)} r(s, \cdot) \right\|_1 \| \theta'(s, \cdot) - \theta(s, \cdot) \|_\infty$$

$$= \max_{s \in \mathcal{S}} \left( \sum_{a \in \mathcal{A}} \pi_{\theta_t}(a|s) \left| r(s, a) - \pi_{\theta_t}(\cdot|s)^\top r(s, \cdot) \right| \right) \| \theta'(s, \cdot) - \theta(s, \cdot) \|_\infty$$

$$\le \max_{s \in \mathcal{S}} \underbrace{\max_{a \in \mathcal{A}} \left| r(s, a) - \pi_{\theta_t}(\cdot|s)^\top r(s, \cdot) \right|}_{\le 1 \text{ since } r(s,a) \in [0,1]} \| \theta'(s, \cdot) - \theta(s, \cdot) \|_\infty$$

$$\le \max_{s \in \mathcal{S}} \| \theta'(s, \cdot) - \theta(s, \cdot) \|_\infty = \| \theta' - \theta \|_\infty.$$

Plugging the above two inequalities into (175) and using Lagrange mean value theorem, we have

$$\left|V^\theta(s) - V^{\theta'}(s)\right| \le \frac{1+\gamma}{(1-\gamma)^2} \|\theta' - \theta\|_\infty \ . \tag{179}$$

**Step 3: sum up.** Combining (179), (170) and (165), we have

$$\forall s \in \mathcal{S}: \quad \left|V_\tau^\theta(s) - V_\tau^{\theta'}(s)\right| \le \frac{1 + \gamma + 2\tau(1-\gamma)\log|\mathcal{A}|}{(1-\gamma)^2} \|\log\pi - \log\pi'\|_\infty \ . \tag{180}$$

Combining (180) and (7a), (103) immediately follows.

**Proof of** (176). For any vector $x \in \mathbb{R}^{|\mathcal{S}|}$, we have

$$\left[\frac{d\boldsymbol{P}_t}{dt}x\right]_s = \sum_{s'\in\mathcal{S}}\sum_{a\in\mathcal{A}} \frac{d\pi_{\theta_t}(a|s)}{dt} \mathcal{P}(s'|s,a)x(s') \, ,$$

from which we can bound the $l_\infty$ norm as

$$\left\|\frac{d\boldsymbol{P}_t}{dt}x\right\|_\infty \le \max_s \sum_{a\in\mathcal{A}}\sum_{s'\in\mathcal{S}} \mathcal{P}(s'|s,a)\left|\frac{d\pi_{\theta_t}(a|s)}{dt}\right| \|x\|_\infty$$

$$= \max_s \sum_{a\in\mathcal{A}} \left|\frac{d\pi_{\theta_t}(a|s)}{dt}\right| \|x\|_\infty$$

$$\le 2 \|\theta' - \theta\|_\infty \|x\|_\infty$$

as desired, where the last line follows from the following fact:

$$\sum_{a\in\mathcal{A}}\left|\frac{d\pi_{\theta_t}(a|s)}{dt}\right| = \sum_{a\in\mathcal{A}}\left|\left\langle\frac{\partial\pi_{\theta_t}(a|s)}{\partial\theta_t}, \theta'-\theta\right\rangle\right|$$

$$= \sum_{a\in\mathcal{A}}\left|\left\langle\frac{\partial\pi_{\theta_t}(a|s)}{\partial\theta_t(s,\cdot)}, \theta'(s,\cdot)-\theta(s,\cdot)\right\rangle\right|$$

$$= \sum_{a\in\mathcal{A}}\pi_{\theta_t}(a|s)\left|(\theta'(s,a)-\theta(s,a)) - \pi_{\theta_t}(\cdot|s)^\top(\theta'(s,\cdot)-\theta(s,\cdot))\right|$$

$$\le \max_a |\theta'(s,a)-\theta(s,a)| + \left|\pi_{\theta_t}(\cdot|s)^\top(\theta'(s,\cdot)-\theta(s,\cdot))\right|$$

$$\le 2\|\theta'-\theta\|_\infty \ .$$

### D.3 PROOF OF LEMMA 9

To simplify the notation, we denote

$$\delta^{(t)} := \widehat{Q}_\tau^{(t)} - \overline{Q}_\tau^{(t)} \ . \tag{181}$$

We first rearrange the terms of (164) and obtain

$$-\tau\log\overline{\pi}^{(t)}(a|s) + \left(\overline{Q}_\tau^{(t)}(s,a) + \delta^{(t)}(s,a)\right) = \frac{1-\gamma}{\eta}\left(\log\overline{\pi}^{(t+1)}(a|s) - \log\overline{\pi}^{(t)}(a|s)\right) + \frac{1-\gamma}{\eta}z^{(t)}(s) \ . \tag{182}$$

This in turn allows us to express $\overline{V}_\tau^{(t)}(s_0)$ for any $s_0 \in \mathcal{S}$ as follows

$$\overline{V}_\tau^{(t)}(s_0) = \mathop{\mathbb{E}}_{a_0\sim\overline{\pi}^{(t)}(\cdot|s_0)}\left[-\tau\log\overline{\pi}^{(t)}(a_0|s_0) + \overline{Q}_\tau^{(t)}(s_0,a_0)\right]$$

$$= \mathop{\mathbb{E}}_{a_0\sim\overline{\pi}^{(t)}(\cdot|s_0)}\left[\frac{1-\gamma}{\eta}z^{(t)}(s_0)\right] + \mathop{\mathbb{E}}_{a_0\sim\overline{\pi}^{(t)}(\cdot|s_0)}\left[\frac{1-\gamma}{\eta}\left(\log\overline{\pi}^{(t+1)}(a_0|s_0) - \log\overline{\pi}^{(t)}(a_0|s_0)\right) - \delta^{(t)}(s_0,a_0)\right]$$

$$= \frac{1-\gamma}{\eta}z^{(t)}(s_0) - \frac{1-\gamma}{\eta}\mathsf{KL}\big(\overline{\pi}^{(t)}(\cdot|s_0)\,\|\,\overline{\pi}^{(t+1)}(\cdot|s_0)\big) - \mathop{\mathbb{E}}_{a_0\sim\overline{\pi}^{(t)}(\cdot|s_0)}\left[\delta^{(t)}(s_0,a_0)\right]$$

$$= \mathop{\mathbb{E}}_{a_0\sim\overline{\pi}^{(t+1)}(\cdot|s_0)}\left[\frac{1-\gamma}{\eta}z^{(t)}(s_0)\right] - \frac{1-\gamma}{\eta}\mathsf{KL}\big(\overline{\pi}^{(t)}(\cdot|s_0)\,\|\,\overline{\pi}^{(t+1)}(\cdot|s_0)\big) - \mathop{\mathbb{E}}_{a_0\sim\overline{\pi}^{(t)}(\cdot|s_0)}\left[\delta^{(t)}(s_0,a_0)\right] \, , \tag{183}$$

where the first identity makes use of (7b), the second line follows from (182). Invoking (7b) again to rewrite the $z(s_0)$ appearing in the first term of (183), we reach

$$
\begin{aligned}
\overline{V}_\tau^{(t)}&(s_0)\\
=&\mathop{\mathbb{E}}_{a_0\sim\overline{\pi}^{(t+1)}(\cdot|s_0)}\left[-\tau\log\overline{\pi}^{(t+1)}(a_0|s_0)+\overline{Q}_\tau^{(t)}(s_0,a_0)+\left(\tau-\frac{1-\gamma}{\eta}\right)\left(\log\overline{\pi}^{(t+1)}(a_0|s_0)-\log\overline{\pi}^{(t)}(a|s)\right)\right]\\
&-\frac{1-\gamma}{\eta}\mathsf{KL}\big(\overline{\pi}^{(t)}(\cdot|s_0)\,\|\,\overline{\pi}^{(t+1)}(\cdot|s_0)\big)-\mathop{\mathbb{E}}_{a_0\sim\overline{\pi}^{(t)}(\cdot|s_0)}\left[\delta^{(t)}(s_0,a_0)\right]+\mathop{\mathbb{E}}_{a_0\sim\overline{\pi}^{(t+1)}(\cdot|s_0)}\left[\delta^{(t)}(s_0,a_0)\right]\\
=&\mathop{\mathbb{E}}_{\substack{a_0\sim\overline{\pi}^{(t+1)}(\cdot|s_0),\\s_1\sim P(\cdot|s_0,a_0)}}\left[-\tau\log\overline{\pi}^{(t+1)}(a_0|s_0)+r(s_0,a_0)+\gamma\overline{V}_\tau^{(t)}(s_0)\right]\\
&-\left(\frac{1-\gamma}{\eta}-\tau\right)\mathsf{KL}\big(\overline{\pi}^{(t+1)}(\cdot|s_0)\,\|\,\overline{\pi}^{(t)}(\cdot|s_0)\big)-\frac{1-\gamma}{\eta}\mathsf{KL}\big(\overline{\pi}^{(t)}(\cdot|s_0)\,\|\,\overline{\pi}^{(t+1)}(\cdot|s_0)\big)\\
&-\mathop{\mathbb{E}}_{a_0\sim\overline{\pi}^{(t)}(\cdot|s_0)}\left[\delta^{(t)}(s_0,a_0)\right]+\mathop{\mathbb{E}}_{a_0\sim\overline{\pi}^{(t+1)}(\cdot|s_0)}\left[\delta^{(t)}(s_0,a_0)\right].
\end{aligned}
\tag{184}
$$

Note that for any $(s_0,a_0)\in\mathcal{S}\times\mathcal{A}$, we have

$$
\begin{aligned}
&-\mathop{\mathbb{E}}_{a_0\sim\overline{\pi}^{(t)}(\cdot|s_0)}\left[\delta^{(t)}(s_0,a_0)\right]+\mathop{\mathbb{E}}_{a_0\sim\overline{\pi}^{(t+1)}(\cdot|s_0)}\left[\delta^{(t)}(s_0,a_0)\right]\\
&=\sum_{a_0\in\mathcal{A}}\left(\overline{\pi}^{(t+1)}(a_0|s_0)-\overline{\pi}^{(t)}(a_0|s_0)\right)\delta^{(t)}(s_0,a_0)\\
&\leq\left\|\overline{\pi}^{(t+1)}(\cdot|s_0)-\overline{\pi}^{(t)}(\cdot|s_0)\right\|_1\left\|\delta^{(t)}\right\|_\infty\leq 2\left\|\delta^{(t)}\right\|_\infty.
\end{aligned}
\tag{185}
$$

To finish up, applying (184) recursively to expand $\overline{V}_\tau^{(t)}(s_i)$, $i\geq 1$ and making use of (185), we arrive at

$$
\begin{aligned}
\overline{V}_\tau^{(t)}&(s_0)\\
\leq&\sum_{i=1}^\infty\gamma^i\cdot 2\left\|\delta^{(t)}\right\|_\infty+\mathop{\mathbb{E}}_{\substack{a_i\sim\overline{\pi}^{(t+1)}(\cdot|s_i),\\s_{i+1}\sim P(\cdot|s_i,a_i),\forall i\geq 0}}\left[\sum_{i=1}^\infty\gamma^i\left\{r(s_i,a_i)-\tau\log\overline{\pi}^{(t+1)}(a_i|s_i)\right\}\right.\\
&\left.-\sum_{i=1}^\infty\gamma^i\left\{\left(\frac{1-\gamma}{\eta}-\tau\right)\mathsf{KL}\big(\overline{\pi}^{(t+1)}(\cdot|s_i)\,\|\,\overline{\pi}^{(t)}(\cdot|s_i)\big)+\frac{1-\gamma}{\eta}\mathsf{KL}\big(\overline{\pi}^{(t)}(\cdot|s_i)\,\|\,\overline{\pi}^{(t+1)}(\cdot|s_i)\big)\right\}\right]\\
=&\frac{2}{1-\gamma}\left\|\delta^{(t)}\right\|_\infty+\overline{V}_\tau^{(t+1)}(s_0)\\
&-\mathop{\mathbb{E}}_{s\sim d_{s_0}^{\overline{\pi}^{(t+1)}}}\left[\left(\frac{1}{\eta}-\frac{\tau}{1-\gamma}\right)\mathsf{KL}\big(\overline{\pi}^{(t+1)}(\cdot|s_i)\,\|\,\overline{\pi}^{(t)}(\cdot|s_i)\big)+\frac{1}{\eta}\mathsf{KL}\big(\overline{\pi}^{(t)}(\cdot|s_i)\,\|\,\overline{\pi}^{(t+1)}(\cdot|s_i)\big)\right],
\end{aligned}
\tag{186}
$$

where the third line follows since $\overline{V}_\tau^{(t+1)}$ can be viewed as the value function of $\overline{\pi}^{(t+1)}$ with adjusted rewards $\overline{r}^{(t+1)}(s,a):=r(s,a)-\tau\log\overline{\pi}^{(t+1)}(s|a)$. And (121) follows immediately from the above inequality (186). By (7a) we can easily see that (122) is a consequence of (121).

### D.4 Proof of Lemma 11

We first introduce the famous performance difference lemma which will be used in our proof.

**Lemma 12** (Performance difference lemma). *For all policies $\pi,\pi'$ and state $s_0$, we have*

$$
V^\pi(s_0)-V^{\pi'}(s_0)=\frac{1}{1-\gamma}\mathbb{E}_{s\sim d_{s_0}^\pi}\mathbb{E}_{a\sim\pi(\cdot|s)}\left[A^{\pi'}(s,a)\right].
\tag{187}
$$

The proof of Lemma 12 can be found in, for example, Appendix A of Agarwal et al. (2021).

For all $t \geq 0$, we define the advantage function $\overline{A}^{(t)}$ as:

$$\forall (s, a) \in \mathcal{S} \times \mathcal{A}: \quad \overline{A}^{(t)}(s, a) \coloneqq \overline{Q}^{(t)}(s, a) - \overline{V}^{(t)}(s). \tag{188}$$

Then for Alg. 1, the update rule of $\overline{\pi}$ (Eq. (164)) can be written as

$$\log \overline{\pi}^{(t+1)}(a|s) = \log \overline{\pi}^{(t)}(a|s) + \frac{\eta}{1-\gamma} \left( \overline{A}^{(t)}(s, a) + \delta^{(t)}(s, a) \right) - \log \widehat{z}^{(t)}(s), \tag{189}$$

where $\delta^{(t)}$ is defined in (181) and

$$\begin{aligned}
\log \widehat{z}^{(t)}(s) &= \log \sum_{a' \in \mathcal{A}} \overline{\pi}^{(t)}(a'|s) \exp \left\{ \frac{\eta}{1-\gamma} \left( \overline{A}^{(t)}(s, a') + \delta^{(t)}(s, a') \right) \right\} \\
&\geq \sum_{a' \in \mathcal{A}} \overline{\pi}^{(t)}(a'|s) \log \exp \left\{ \frac{\eta}{1-\gamma} \left( \overline{A}^{(t)}(s, a') + \delta^{(t)}(s, a') \right) \right\} \\
&= \frac{\eta}{1-\gamma} \sum_{a' \in \mathcal{A}} \overline{\pi}^{(t)}(a'|s) \left( \overline{A}^{(t)}(s, a') + \delta^{(t)}(s, a') \right) \\
&= \frac{\eta}{1-\gamma} \sum_{a' \in \mathcal{A}} \overline{\pi}^{(t)}(a'|s) \delta^{(t)}(s, a') \geq -\frac{\eta}{1-\gamma} \left\| \delta^{(t)} \right\|_\infty,
\end{aligned} \tag{190}$$

where the first inequality follows by Jensen's inequality on the concave function $\log x$ and the last equality uses $\sum_{a' \in \mathcal{A}} \overline{\pi}^{(t)}(a'|s) \overline{A}^{(t)}(s, a') = 0$.

For all starting state distribution $\mu$, we use $d^{(t+1)}$ as shorthand for $d_\mu^{\overline{\pi}^{(t+1)}}$, the performance difference lemma (Lemma 12) implies:

$$\begin{aligned}
&\overline{V}^{(t+1)}(\mu) - \overline{V}^{(t)}(\mu) \\
&= \frac{1}{1-\gamma} \mathbb{E}_{s \sim d^{(t+1)}} \sum_{a \in \mathcal{A}} \overline{\pi}^{(t+1)}(a|s) \left( \overline{A}^{(t)}(s, a) + \delta^{(t)}(s, a) \right) - \frac{1}{1-\gamma} \mathbb{E}_{s \sim d^{(t+1)}} \mathbb{E}_{a \sim \overline{\pi}^{(t+1)}(\cdot|s)} \left[ \delta^{(t)}(s, a) \right] \\
&= \frac{1}{\eta} \mathbb{E}_{s \sim d^{(t+1)}} \sum_{a \in \mathcal{A}} \overline{\pi}^{(t+1)}(a|s) \log \frac{\overline{\pi}^{(t+1)}(a|s) \widehat{z}^{(t)}(s)}{\overline{\pi}^{(t)}(a|s)} - \frac{1}{1-\gamma} \mathbb{E}_{s \sim d^{(t+1)}} \mathbb{E}_{a \sim \overline{\pi}^{(t+1)}(\cdot|s)} \left[ \delta^{(t)}(s, a) \right] \\
&= \frac{1}{\eta} \mathbb{E}_{s \sim d^{(t+1)}} \mathsf{KL} \left( \overline{\pi}^{(t+1)}(\cdot|s) \,\|\, \overline{\pi}^{(t)}(\cdot|s) \right) + \frac{1}{\eta} \mathbb{E}_{s \sim d^{(t+1)}} \log \widehat{z}^{(t)}(s) - \frac{1}{1-\gamma} \mathbb{E}_{s \sim d^{(t+1)}} \mathbb{E}_{a \sim \overline{\pi}^{(t+1)}(\cdot|s)} \left[ \delta^{(t)}(s, a) \right] \\
&\geq \frac{1}{\eta} \mathbb{E}_{s \sim d^{(t+1)}} \left( \log \widehat{z}^{(t)}(s) + \frac{\eta}{1-\gamma} \left\| \delta^{(t)} \right\|_\infty \right) - \frac{2}{1-\gamma} \left\| \delta^{(t)} \right\|_\infty,
\end{aligned}$$

from which we can see that

$$\overline{V}^{(t+1)}(\mu) - \overline{V}^{(t)}(\mu) \geq -\frac{2}{1-\gamma} \left\| \delta^{(t)} \right\|_\infty, \tag{191}$$

where we use (190), and that

$$\overline{V}^{(t+1)}(\mu) - \overline{V}^{(t)}(\mu) \geq \frac{1-\gamma}{\eta} \mathbb{E}_{s \sim \mu} \left( \log \widehat{z}^{(t)}(s) + \frac{\eta}{1-\gamma} \left\| \delta^{(t)} \right\|_\infty \right) - \frac{2}{1-\gamma} \left\| \delta^{(t)} \right\|_\infty, \tag{192}$$

which follows from $d^{(t+1)} = d_\mu^{\overline{\pi}^{(t+1)}} \geq (1-\gamma)\mu$ and the fact that $\log \widehat{z}^{(t)}(s) + \frac{\eta}{1-\gamma} \left\| \delta^{(t)} \right\|_\infty \geq 0$ (by (190)).

For any fixed $\rho$, we use $d^\star$ as shorthand for $d_\rho^{\pi^\star}$. By the performance difference lemma (Lemma 12),

$$
\begin{aligned}
&V^\star(\rho) - \overline{V}^{(t)}(\rho) \\
&= \frac{1}{1-\gamma}\mathbb{E}_{s\sim d^\star} \sum_{a\in\mathcal{A}} \pi^\star(a|s) \left(\overline{A}^{(t)}(s,a) + \delta^{(t)}(s,a)\right) - \frac{1}{1-\gamma}\mathbb{E}_{s\sim d^\star}\mathbb{E}_{a\sim\pi^\star(\cdot|s)}\left[\delta^{(t)}(s,a)\right] \\
&= \frac{1}{\eta}\mathbb{E}_{s\sim d^\star} \sum_{a\in\mathcal{A}} \pi^\star(a|s) \log \frac{\overline{\pi}^{(t+1)}(a|s)\widehat{z}^{(t)}(s)}{\overline{\pi}^{(t)}(a|s)} - \frac{1}{1-\gamma}\mathbb{E}_{s\sim d^\star}\mathbb{E}_{a\sim\pi^\star(\cdot|s)}\left[\delta^{(t)}(s,a)\right] \\
&= \frac{1}{\eta}\mathbb{E}_{s\sim d^\star} \left(\mathsf{KL}\big(\pi^\star(\cdot|s)\,\|\,\overline{\pi}^{(t)}(\cdot|s)\big) - \mathsf{KL}\big(\pi^\star(\cdot|s)\,\|\,\overline{\pi}^{(t+1)}(\cdot|s)\big) + \log\widehat{z}^{(t)}(s)\right) - \frac{1}{1-\gamma}\mathbb{E}_{s\sim d^\star}\mathbb{E}_{a\sim\pi^\star(\cdot|s)}\left[\delta^{(t)}(s,a)\right] \\
&\leq \frac{1}{\eta}\mathbb{E}_{s\sim d^\star} \left(\mathsf{KL}\big(\pi^\star(\cdot|s)\,\|\,\overline{\pi}^{(t)}(\cdot|s)\big) - \mathsf{KL}\big(\pi^\star(\cdot|s)\,\|\,\overline{\pi}^{(t+1)}(\cdot|s)\big) + \left(\log\widehat{z}^{(t)}(s) + \frac{\eta}{1-\gamma}\big\|\delta^{(t)}\big\|_\infty\right)\right),
\end{aligned}
$$
$$(193)$$

where we use (189) in the second equality.

By applying (192) with $\mu = d^\star$ as the initial state distribution, we have

$$
\frac{1}{\eta}\mathbb{E}_{s\sim\mu}\left(\log\widehat{z}^{(t)}(s) + \frac{\eta}{1-\gamma}\big\|\delta^{(t)}\big\|_\infty\right) \leq \frac{1}{1-\gamma}\left(\overline{V}^{(t+1)}(d^\star) - \overline{V}^{(t)}(d^\star)\right) + \frac{2}{(1-\gamma)^2}\big\|\delta^{(t)}\big\|_\infty.
$$

Plugging the above equation into (193), we obtain

$$
\begin{aligned}
V^\star(\rho) - \overline{V}^{(t)}(\rho) \leq{}& \frac{1}{\eta}\mathbb{E}_{s\sim d^\star}\left(\mathsf{KL}\big(\pi^\star(\cdot|s)\,\|\,\overline{\pi}^{(t)}(\cdot|s)\big) - \mathsf{KL}\big(\pi^\star(\cdot|s)\,\|\,\overline{\pi}^{(t+1)}(\cdot|s)\big)\right) \\
&+ \frac{1}{1-\gamma}\left(\overline{V}^{(t+1)}(d^\star) - \overline{V}^{(t)}(d^\star)\right) + \frac{2}{(1-\gamma)^2}\big\|\delta^{(t)}\big\|_\infty,
\end{aligned}
$$

which gives Lemma 11.

# E    EXPERIMENTS

We study the empirical performance of FedNPG (Algorithm 1) and entropy-regularized FedNPG (Algorithm 2) on a $K \times K$ GridWorld problem. To be specific, the collective goal of $N$ agents is to learn a global optimal policy to follow a predetermined path which starts at the top left corner and ends at the bottom right corner. However, each agent only has access to partial information about the whole map: in Figure 1 (where we take $N = 3$ and $K = 9$ as an example), agent $n$ explores on map $n$, $n \in [N]$. After taking an action, the agent gets reward 1 only when it is at the shaded positions, otherwise it gets reward 0. We stipulate the action space of all agents to be $\mathcal{A} = \{\text{right}, \text{down}\}$, i.e. movement is allowed only in two directions right or down. If an agent takes an action that will lead it out of the boarder of the map, we stipulate the agent's state doesn't change and receive reward 0. Each agent starts at the top left corner. To learn a shared policy to follow the path, we aim to maximize the average value function of all agents.

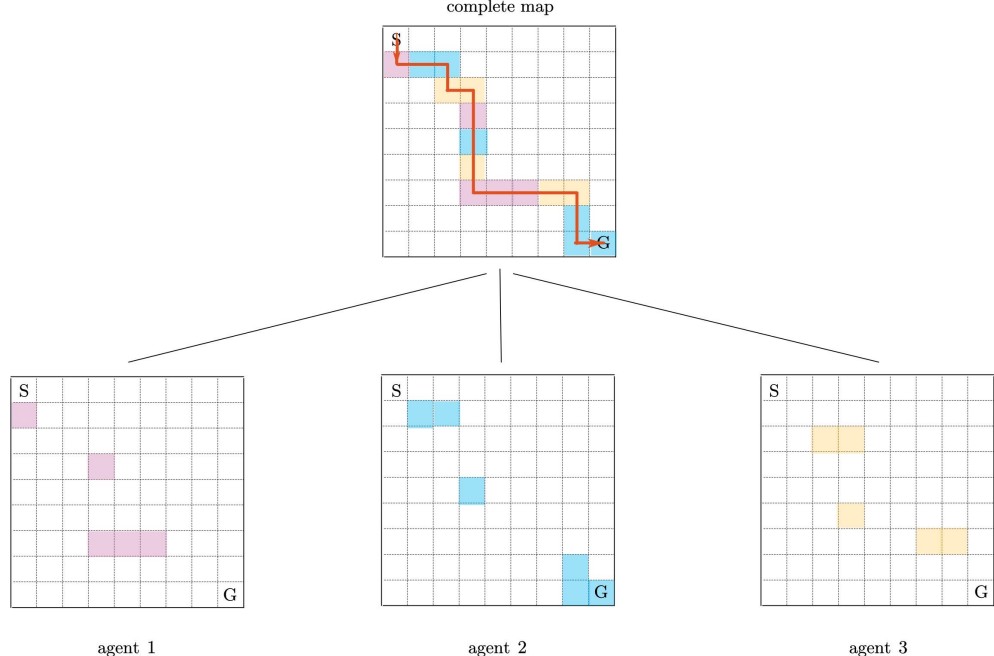

Figure 1: Gridworld experiment. $N$ agents ($N = 3$ in this illustration) aim to learn a shared policy to follow a predetermined path, which is the red line in the complete map. Each agent only has access to partial information about the path and gets reward 1 only at the shaded positions and 0 at other positions. Each agent starts at the top left corner.

In the following, we provide empirical results of our algorithms. In all the experiments, we fix the discounted factor $\gamma = 0.99$. We validate the effectiveness of vanilla FedNPG and entropy-regularized FedNPG across different map size $K$, where we set $\tau = 0, 0.005, 0.05$, $\eta = 0.1$, and $N = 10$. We use a *standard ring graph*, where agent $n$ receives information from agent $n + 1$ for $n \in [N - 1]$, and agent $N$ receives information from agent 1, with all the weights on each edge of the communication graph set to be 0.5. The corresponding mixing matrix of the standard ring graph is as follows:

$$\boldsymbol{W} = \begin{pmatrix} 0.5 & 0.5 & 0 & 0 & \cdots & 0 & 0 \\ 0 & 0.5 & 0.5 & 0 & \cdots & 0 & 0 \\ 0 & 0 & 0.5 & 0.5 & \cdots & 0 & 0 \\ \vdots & \vdots & \vdots & \vdots & & \vdots & \vdots \\ 0 & 0 & 0 & 0 & \cdots & 0.5 & 0.5 \\ 0.5 & 0 & 0 & 0 & \cdots & 0 & 0.5 \end{pmatrix}. \tag{194}$$

Here, $\boldsymbol{W}$ in (194) satisfies the double stochasticity assumption but is not symmetric, hence doesn't strictly adhere to Assumption 1

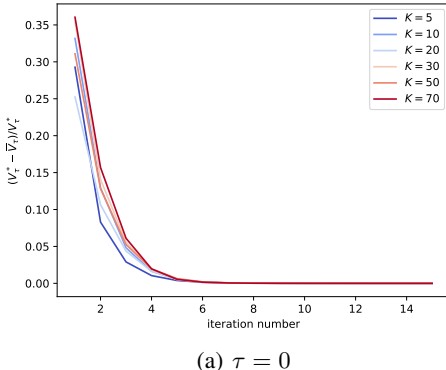 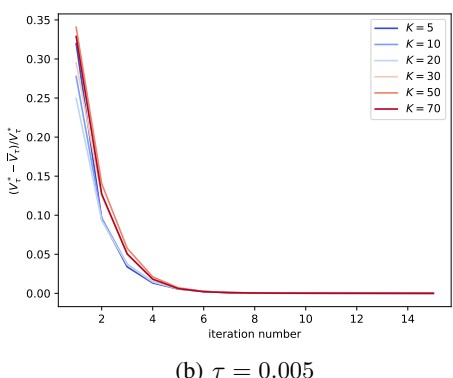

(a) $\tau = 0$                (b) $\tau = 0.005$

Figure 2: **Changing map size $K$.** We let $\tau = 0, 0.005$ and change $K$ for each $\tau$. We plot $V_\tau^\star - \overline{V}_\tau^{(t)}$ with respect to the number of iterations $t$. Both vanilla and entropy-regularized NPG converges to the optimal value function in a few iterations, and the convergence speed is almost the same across different $K$.

Figure 2 plots $V_\tau^\star - \overline{V}_\tau^{(t)}$ with respect to the number of iterations $t$ for $\tau = 0$ (vanilla) and $\tau = 0.005$ (regularized). We can see that both vanilla and entropy-regularized FedNPG converge to the optimal value function in a few iterations, and the convergence speed is almost the same across different $K$, i.e. the impact of $K$ on the convergence speed is minimal.

Figure 3 shows the performance of our algorithm when the number of agents $N$ varies. We set $K = 10, 20, 30$, $\tau = 0.005$, and the communication graph to be the standard ring graph. We can see that the convergence speed decreases as $N$ increases. Same as before, the convergence speed is insensitive to the change of $K$.

In Figure 4, we illustrate the impact of the network topology to our algorithms. To be specific, we change the number of neighbors of each agent (i.e., the number of non-zero entries in each row of $\boldsymbol{W}$), and (i) randomly generalize the weights of the graph such that each row of $\boldsymbol{W}$ sum up to 1, i.e.,$\boldsymbol{W}\mathbf{1} = \mathbf{1}$, see Figure 4(a); (ii) set the non-zero entries in each row of $\boldsymbol{W}$ all to be $\frac{1}{\text{number of neighbors}}$, see Figure 4(b). We fix $\eta = 0.1$, $K = 10$, $\tau = 0.005$. We plot the curves of value functions changing with respect to the number of iterations. The green dashed line represents the optimal value. For both 4(a) and 4(b), the convergence speed increase as number of neighbors of each agent increases. FedNPG performs better when using equal weights.

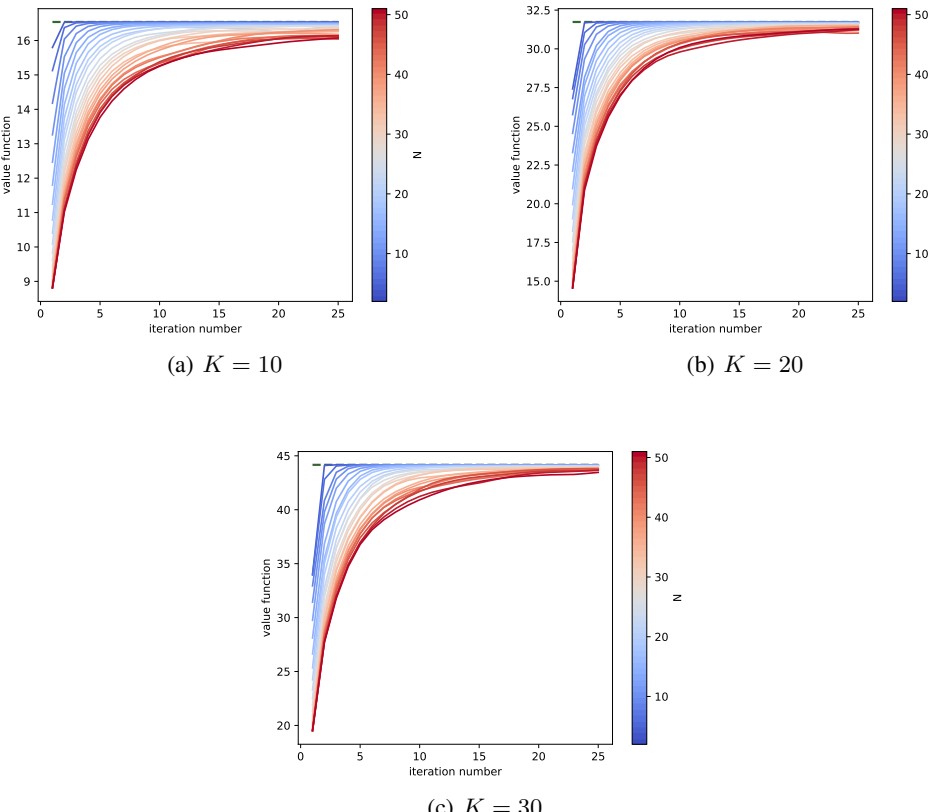

(a) $K = 10$    (b) $K = 20$

(c) $K = 30$

Figure 3: **Changing number of agents $N$.** we let $K = 10, 20, 30$ and change $N$ for each $K$. We plot the curves of value functions changing with the iteration number. The green dashed line represents the optimal value. We can see that the convergence speed decreases as $N$ increases. Same as before, the convergence speed is insensitive to the change of $K$.

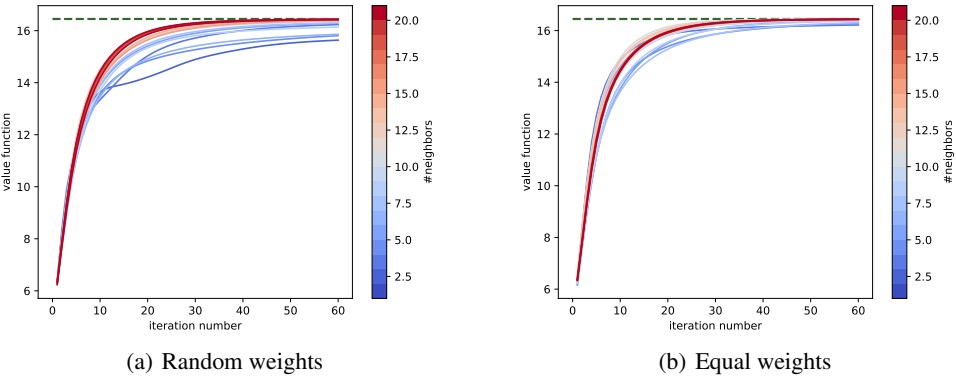

(a) Random weights    (b) Equal weights

Figure 4: **Changing communication network topology.** We change the number of neighbors of each agent. In Figure 4(a), we randomly generalize the weights of the graph such that each row of $\boldsymbol{W}$ sum up to 1; (ii) In Figure 4(b), we set the non-zero entries in each row of $\boldsymbol{W}$ all to be $\frac{1}{\text{number of neighbors}}$. We plot the curves of value functions changing with the iteration number. The green dashed line represents the optimal value. For both 4(a) and 4(b), the convergence speed increase as number of neighbors increases. FedNPG performs better when using equal weights.

