# OpenReview forum: "Federated Natural Policy Gradient Methods for Multi-task Reinforcement Learning"
_ICLR.cc/2024/Conference — Submitted to ICLR 2024_

### Official Review · Reviewer_GXb4 · 2023-10-21

**Soundness:** 2 fair
**Presentation:** 3 good
**Contribution:** 2 fair
**Rating:** 6
**Confidence:** 3

**Summary:**

This paper analyzes the decentralized federated natural policy gradient method for multi-tasks in the infinite-horizon tabular setting. Precisely, all agents have the same transaction matrix but different rewards. Agents communicate with neighbors in a prescribed graph topology. Both federated vanilla and entropy-regularized NPG methods are analyzed with global convergence rates. With exact policy evaluation, non-asymptotic global convergence is guaranteed. With imperfect policy evaluation, convergence rates remain the same when the infinite norms of approximation errors are small.

**Strengths:**

1.	To the best of my knowledge, this is the first work on decentralized FedNPG with convergence analysis.
2.	Without trajectory transmission, the results show that the convergence rates do not show down a lot.
3.	With function approximation, the communication complexity of vanilla FedNPG would be very high. But the natural policy gradient update in the tabular setting has a simple form. It is wise to choose this as no higher-order matrix is involved.

**Weaknesses:**

1.	Only the tabular setting is studied. The action and state space are discrete and finite.
2.	It is good enough to give convergence performances for previously proposed algorithms (or with minor changes). However, the decentralized FedNPG algorithm is quite new. Some simulations are needed to verify the proposed algorithms.
3.	In practice, it is very hard to be synchronous in each iteration with fully distributed settings. Especially, each agent randomly (categorical distribution) selects one agent to communicate.

**Questions:**

1.	Should the mixing matrix $\mathbf{W}$ be ergodic? I cannot find a related assumption or discussion, which confuses me with the statement “illuminate the impacts of network size and connectivity”. Can each agent compute independently and do a one-shot average? There is a connectivity rule in (Nedic & Ozdaglar, 2009), but not here.
2.	As the local update is a key point in federated learning, is it possible to compute locally for several iterations without communication?
3.	Is it possible to show some simulation results?
4.	(Clarification) Are the reward functions deterministic?
5.	(Motivation) I personally like this topic, and would like to know more about the motivation. As each agent has its own reward function, why don’t they simply use the local policies instead of the global policy? Does it make sense to force them to use the same policy?

This work is generally good. I promise to raise the score if questions 1 - 3 are fairly (or partially) addressed.

---

> ### Author Response · Authors · 2023-11-20
> **Response to Reviewer GXb4**
>
> We thank the reviewer for the useful feedback.
>
> > Should the mixing matrix $\mathbf{W}$ be ergodic? I cannot find a related assumption or discussion, which confuses me with the statement “illuminate the impacts of network size and connectivity”. Can each agent compute independently and do a one-shot average? There is a connectivity rule in (Nedic & Ozdaglar, 2009), but not here.
>
> Thanks for bringing this to our attention! Yes, we need $\mathbf{W}$ to be ergodic to guarantee our eq (17) holds, as $\mathbf{W}$ being non-ergodic would lead to $\sigma = 1$ (footnote 3 in [3]). We have specified the communication graph is connected in the paragraph above Assumption 1 in our revised paper.
>
> > As the local update is a key point in federated learning, is it possible to compute locally for several iterations without communication?
>
> Thanks for the question! As each agent has its own set of reward, performing multiple local updates would lead to large consensus error, which would require increasing communication rounds accordingly to ensure convergence. Verifying the similar linear speedup in [1] for stochastic policy updates with homogeneous agents would be in important direction that worth further investigation.
>
> > It is good enough to give convergence performances for previously proposed algorithms (or with minor changes). However, the decentralized FedNPG algorithm is quite new. Some simulations are needed to verify the proposed algorithms. Is it possible to show some simulation results?
>
> We’ve added the experiments in Appendix E in our paper, where we distributedly train the agents to learn a shared policy to follow a predetermined trajectory in the Gridworld. See also our global response for more details.
>
> > (Clarification) Are the reward functions deterministic?
>
> We have assumed the reward functions to be deterministic for simplicity. However, our analysis for inexact policy evaluation is based on estimates of Q-function, and thus directly extend to the stochastic reward setting as long as we have reasonable estimate of the expected Q-function.
>
> > (Motivation) I personally like this topic, and would like to know more about the motivation. As each agent has its own reward function, why don’t they simply use the local policies instead of the global policy? Does it make sense to force them to use the same policy?
>
> We remark that in the multi-task learning setting, the local task shaped by each agent's own reward may not reflect the collective goal in its entirety, due to task design, privacy concern, distribution shift, etc. In our numerical experiment, a global task is decomposed into multiple local tasks, and the local policies would fail miserably when the current state falls out of the coverage of local rewards, i.e., misalignment between the local task and the global task. Intuitively speaking, we would like all agents collaborate to find a policy that performs reasonably well across different tasks.
>
> > Only the tabular setting is studied. The action and state space are discrete and finite.
>
> - Firstly, to the best of our knowledge, no other algorithms are currently available to find the global optimal policy with non-asymptotic convergence guarantees even for tabular scenario. We emphasize that this is the *first* time that global convergence is established for federated multi-task RL using policy optimization, as is pointed out in the abstract and in Section 1.
> - Secondly, it's essential to note that there is a substantial body of theoretical work in the policy gradient literature that delves into the intricacies of the tabular scenario. References [1-5] are just a few examples of the theoretical foundations laid in this area. Analyzing algorithms in a tabular setting provides a robust understanding of their dynamics and serves as a foundational step towards their generalization to more complex scenarios.
> - Lastly, it's straightforward to generalize our algorithms to scenarios where the state or action space are infinite or continuous by introducing function approximation and , which we leave as future work, as is mentioned in Section 5.
>
> > In practice, it is very hard to be synchronous in each iteration with fully distributed settings. Especially, each agent randomly (categorical distribution) selects one agent to communicate.
>
> We agree that a fixed choice of mixing matrix can slow down the system with a limited communication bandwidth. A straightforward extension is to adopt stochastic mixing matrices (e.g., [2]) whose average satisfies the standard mixing matrix properties. We believe it is possible to incorporate this aspect in our analysis too, which we leave for future investigation.

---

> ### Author Response · Authors · 2023-11-20
> **Reference**
>
> [1] Stich, Sebastian U. "Local SGD converges fast and communicates little."
>
> [2] Wang, Jianyu, Anit Kumar Sahu, Gauri Joshi, and Soummya Kar. "Matcha: A matching-based link scheduling strategy to speed up distributed optimization." IEEE Transactions on Signal Processing 70 (2022): 5208-5221.
>
> [3] Qu G, Li N. Harnessing smoothness to accelerate distributed optimization[J]. IEEE Transactions on Control of Network Systems, 2017, 5(3): 1245-1260.

---

> ### Comment · Reviewer_GXb4 · 2023-11-22
> **Thank you for the responses.**
>
> Thank the authors for the responses. Most of my concerns are addressed. I raised my score to 6.
>
> 1. As for the experiment, without more descriptions, I may take it as "the transition kernel is known for policy evaluation." (partially like planning.) I suggest describing how policies are evaluated in experiments. But I do not think this hurts a lot.
>
> 2. Does W = [0, 1; 1, 0] satisfy the setting in this paper? According to eq (17), what is its radius then? I think the setting needs further checking.

---

### Official Review · Reviewer_tCM8 · 2023-11-01

**Soundness:** 3 good
**Presentation:** 3 good
**Contribution:** 2 fair
**Rating:** 5
**Confidence:** 4

**Summary:**

The study employs the federated NPG method to address a multi-task reinforcement learning challenge. In their setup, there is reward heterogeneity among various agents, though they share identical transition kernels. Two algorithms, namely the vanilla and entropy-regularized FedNPG, are introduced to tackle the decentralized FRL issue within a graph topology. Additionally, the authors offer theoretical assurances for these algorithms.

**Strengths:**

1.The paper is well-written and clearly presented
2. The authors provide a clear comparison of their findings with existing results.
3. The analyses are solid.

**Weaknesses:**

* The study lacks simulation results that would validate the efficacy of the presented algorithms.

* The framework presented is relatively simplistic, being limited to the tabular scenario with deterministic gradients. There's no consideration for function approximation or the presence of noise.

* A notable omission is the lack of multiple local updates in the algorithms, which are the key features in Federated Learning (FL). Heterogeneity only exists  when there are more than one local updates. Consequently, the authors did not examine the influence of heterogeneity between agents, since their algorithms do not incorporate the multiple local update steps.

**Questions:**

* How would the algorithms behave if the transition kernels differ between agents?

* Regarding agents' motivation to participate in the federation, prior studies [1][2][3][4] have explored the incentives in terms of linear or sublinear speedup. Do the proposed algorithms match this expected speedup in convergence rate as the number of agents increases?

[1] Fan, Xiaofeng, Yining Ma, Zhongxiang Dai, Wei Jing, Cheston Tan, and Bryan Kian Hsiang Low. "Fault-tolerant federated reinforcement learning with theoretical guarantee." Advances in Neural Information Processing Systems 34 (2021): 1007-1021.

[2] Khodadadian, Sajad, Pranay Sharma, Gauri Joshi, and Siva Theja Maguluri. "Federated reinforcement learning: Linear speedup under markovian sampling." In International Conference on Machine Learning, pp. 10997-11057. PMLR, 2022.

[3] Wang, Han, Aritra Mitra, Hamed Hassani, George J. Pappas, and James Anderson. "Federated temporal difference learning with linear function approximation under environmental heterogeneity." arXiv preprint arXiv:2302.02212 (2023).

[4] Shen, Han, Kaiqing Zhang, Mingyi Hong, and Tianyi Chen. "Towards Understanding Asynchronous Advantage Actor-critic: Convergence and Linear Speedup." IEEE Transactions on Signal Processing (2023).

---

> ### Author Response · Authors · 2023-11-20
> **Response to Reviewer tCM8**
>
> Thank you for reading our paper and your comment.
>
> > The study lacks simulation results that would validate the efficacy of the presented algorithms.
>
> We’ve added the experiments in Appendix E in our paper, where we distributedly train the agents to learn a shared policy to follow a predetermined trajectory in the Gridworld. See also our global response for more details.
>
> > The framework presented is relatively simplistic, being limited to the tabular scenario with deterministic gradients. There's no consideration for function approximation or the presence of noise.
>
> - Firstly, to the best of our knowledge, no other algorithms are currently available to find the global optimal policy with non-asymptotic convergence guarantees even for tabular scenario. We emphasize that this is the *first* time that global convergence is established for federated multi-task RL using policy optimization, as is pointed out in the abstract and in Section 1.
> - Secondly, it's essential to note that there is a substantial body of theoretical work in the policy gradient literature that delves into the intricacies of the tabular scenario. References [1-5] are just a few examples of the theoretical foundations laid in this area. Analyzing algorithms in a tabular setting provides a robust understanding of their dynamics and serves as a foundational step towards their generalization to more complex scenarios.
> - Lastly, it's straightforward to generalize our algorithms to scenarios where the state or action space are infinite or continuous by introducing function approximation and to sample version where gradients are stochastic, which we leave as future work, as is mentioned in Section 5.
>
> > A notable omission is the lack of multiple local updates in the algorithms, which are the key features in Federated Learning (FL). Heterogeneity only exists when there are more than one local updates. Consequently, the authors did not examine the influence of heterogeneity between agents, since their algorithms do not incorporate the multiple local update steps.
>
> In this work, we consider the fully decentralized setting, where each agent only exchanges information with its neighbors, and it may take several iterations for one agent to acquire information of another agent that's not its neighbor through the communication network. This leads to a high degree of heterogeneity even without multiple local updates. Note that there're a lot of works in the fully decentralized federated learning literature that do not have multiple local updates in their algorithms, see [6-9] for instance.
>
> > How would the algorithms behave if the transition kernels differ between agents?
>
> When the transition kernels are different across the agents, the algorithms would likely fail to find the optimal policy. Since the preconditioner used by the natural policy gradient method is agent-dependent, the average update is not guaranteed to align with the gradient of the average value function. We remark that the theoretical underpinnings of solving mixture of MDPs remains limited even for centralized learning, and believe this is an important direction that worth further investigation.
>
> > Prior studies [r1][r2][r3][r4] have explored the incentives in terms of linear or sublinear speedup. Do the proposed algorithms match this expected speedup in convergence rate as the number of agents increases?
>
> Thanks for comment! We remark that the aforementioned works assume homogeneous agents, with the exception of [r4] which assumes bounded heterogeneity and includes the bound in the final convergence results. The speedup in convergence essentially stems from distributing required samples among agents, whereas our work focus on understanding the optimization aspect of multi-task learning (e.g., when agents use policy optimization) where agents hold heterogeneous reward functions. Therefore, we believe such linear/sublinear speedup may not be amenable in our result.

---

> > ### Author Response · Authors · 2023-11-20
> > **Reference**
> >
> > [1] Agarwal et al. On the theory of policy gradient methods: Optimality, approximation, and distribution shift. Journal of Machine Learning Research.
> >
> > [2] Lin Xiao. On the convergence rates of policy gradient methods. Journal of Machine Learning Research.
> >
> > [3] Mei et al. On the global convergence rates of softmax policy gradient methods. In Proceedings of the 37th International Conference on Machine Learning.
> >
> > [4] Yuan et al. A general sample complexity analysis of vanilla policy gradient. In Proceedings of The 25th International Conference on Artificial Intelligence and Statistics, volume 151 of Proceedings of Machine Learning Research.
> >
> > [5] Leonardos et al. Global convergence of multi-agent policy gradient in Markov potential games. In International Conference on Learning Representations, 2022.
> >
> > [6] Lalitha et al., 2018. Fully decentralized federated learning. In Proceedings of NeurIPS.
> >
> > [7] Pei et al. Decentralized federated graph neural networks. In International Workshop on Federated and Transfer Learning for Data Sparsity and Confidentiality in Conjunction with IJCAI (2021).
> >
> > [8] Bellet et al. D-Cliques: Compensating for data heterogeneity with topology in decentralized federated learning. In 41st International Symposium on Reliable Distributed Systems proceedings, 2022.
> >
> > [9] Bars et al. Refined convergence and topology learning under heterogeneous data. Proceedings of The 26th International Conference on Artificial Intelligence and Statistics, PMLR 206:1672-1702, 2023.

---

> > ### Comment · Reviewer_tCM8 · 2023-11-22
> > **Thank you for the responses.**
> >
> > I thank the authors for the responses. I chose to maintain my score. Thanks.

---

### Official Review · Reviewer_svj3 · 2023-11-02

**Soundness:** 3 good
**Presentation:** 3 good
**Contribution:** 2 fair
**Rating:** 3
**Confidence:** 2

**Summary:**

This paper proposes federated vanilla and entropy-regularized natural policy gradient methods under softmax parameterization. Some extensibility properties are given or proven, including global convergence and etc. Overall, this paper is well-written, and it has clearly expressed their work and the author's importance.

**Strengths:**

Complete work with well-designed algorithms and theoretical analysis. The authors have considered a less common but easily thought of issue, i.e., multi-task RL.

**Weaknesses:**

It is easy to be considered as a combination of multiple existing works with not clearly discussed motivation. The most important issue is the lack of numercial experiments which could prove the efficiency of the proposed algorithms. The proposed theoretical results are overclaimed a bit, for the reason that there should be some assumpotions on the the structural form of the policy $pi$, like (107), in order to obtain the global covergence.

**Questions:**

1. More experiments to show the efficiency of the proposed algorithms;
2. The convergence results should be improved, or the contributions should be properly clarified;
3. More comparison with distributed optimization methods should be discussed, especially some convergence results. For the reason that maybe there are already some global convergence results for general distributed optimization problems (with multi-task RL as a special case).

**Details Of Ethics Concerns:**

~

---

> ### Author Response · Authors · 2023-11-20
> **Response to Reviewer svj3 (1/2)**
>
> Thank you for reading our paper and for your feedback. According to your comment, we believe your biggest concern is our lack of experiments. We've added the experiments in Appendix E in our paper, where we distributedly train the agents to learn a globally optimal policy to follow a predetermined trajectory in the Gridworld. See also our global response for more details.
>
> Below we address your other questions and comments.
>
> > not clearly discussed motivation.
>
> - To provide a clearer explanation of our motivation, we first restate the problem we study and some of the possible applications here (see Section 1 for more details):
>   - we consider a multi-task setting in which each agent has its own private reward function corresponding to different tasks, while sharing the same transition kernel of the environment. The collective goal is to learn a shared policy that maximizes the total rewards accumulated from all the agents.
> - There are a lot of application scenarios where our setting becomes highly relevant. In Section 1, we give two such scenarios as follows:
>   - in healthcare[1], different hospitals may be interested in finding an optimal treatment for all patients without disclosing private data, where the effectiveness of the treatment can vary across different hospitals due to demographical differences.
>   - to enhance ChatGPT’s performance across different tasks or domains[2,3], one might consult domain experts to chat and rate ChatGPT’s outputs for solving different tasks, and train ChatGPT in a federated manner without exposing private data or feedback of each expert.
> - In addition, our setting is especially suitable for the multi-task problems where each agent only have partial access of the "global" task. There are a lot of such problems.
>   - An example is the problem we consider in our experiments (see Appendix E), where we distributedly train the agents to learn a shared policy to follow a predetermined trajectory while each agent only has partial information of this trajectory.
>   - The above problem could be seen as a simplified version of the Unmanned Aerial Vehicle (UAV) Patrol Mission, each unmanned aerial vehicle (UAV) patrols only in a specific area, and they need to collectively train a strategy utilizing information from the entire patrol range.
>   - In the game setting, different agents aim to train a character to perform well in multiple tasks, and each agent trains on one task.
>
> - Despite the promise, provably efficient algorithms for federated multi-task RL remain substantially under-explored, especially in the fully decentralized setting. Our work is the first to provide efficient algrithms with global convergence guarantees for federated multi-task RL.
>
> > It is easy to be considered as a combination of multiple existing works.
>
> We remark that while our work is built upon the algorithmic ideas in the distributed learning, reinforcement learning and optimization literature, it is not a strightforward combination and the theoretical analysis is by no means trivial.
>
> One key difficulty is to estimate the global Q-functions using only neighboring information and local data. To address this issue, we invoke the "Q-tracking" step (see Algorithm 1,2), which is inspired by the gradient tracking method in decentralized optimization. Note that this generalization is highly non-trivial: to the best of our knowledge, the utility of gradient tracking has not been exploited in policy optimization, and the intrinsic nonconcavity issue, together with the use of natural gradients, prevents us from directly using the results from decentralized optimization. It is thus of great value to study if the combination of NPG and gradient tracking could lead to fast globally convergent algorithms as in the standard decentralized optimization literature despite the nonconcavity.
>
> Besides, due to the lack of global information sharing, care needs to be taken to judiciously balance the use of neighboring information (to facilitate consensus) and local data (to facilitate learning) when updating the policy. Compared to the centralized version of our proposed algorithms, a much more delicate theoretical analysis is required to prove our convergence results. For example, the key step to establish the convergence rate of the single-agent exact entropy-regularized NPG is to form the 2nd-order linear system in Eq.(46) in Cen et al., 2022 ([4]), while in our corresponding analysis, a 4th-order linear system in Lemma 1 is needed, where the inequality in each line is non-trivial and requires the introduction of some intricate and novel auxiliary lemmas, see appendix C.

---

> ### Author Response · Authors · 2023-11-20
> **Response to Reviewer svj3 (2/2)**
>
> > The proposed theoretical results are overclaimed a bit, for the reason that there should be some assumpotions on the the structural form of the policy , like (107), in order to obtain the global covergence.
>
> We do not overclaim the theoretical results and have made it clear that our algorithms are under softmax parameterization in, for example, Abstract, Section 1.1, 1.2, 2.3 and 3.2.
> Note that softmax parameterization is popular in practice and is often adopted in the policy optimization literature, see [4-8] for instance.
>
> > Q2: The convergence results should be improved, or the contributions should be properly clarified.
>
> We assume this question corresponds to the above comment about the softmax parameterization we use. Hope our above explanation addresses your concern, and please let us know otherwise.
>
> > More comparison with distributed optimization methods should be discussed, especially some convergence results. For the reason that maybe there are already some global convergence results for general distributed optimization problems (with multi-task RL as a special case).
>
> To the best of our knowledge, no other algorithms for federated multi-task RL are currently available to find the global optimal policy with non-asymptotic convergence guarantees even for tabular infinite-horizon Markov decision processes. Our work is the first to provide algorithms with finite-time global convergence guarantees for federated multi-task RL using policy optimization. We have specified this in the abstract as well as in Section 1.
>
> On the other end, for generic nonconvex decentralized minimization, typically only first-order convergence guarantees are available. Therefore, they do not lead to global convergence guarantees when specialized to the multi-task RL problem. The reason that we can achieve global convergence is by leveraging specific problem structures in the Markov decision processes.
>
> Note that we do compare our algorithms with their centralized counterparts in Table 1, where we specify that when the network is fully connected (i.e., $\sigma=0$), the convergence results of our algorithms recover that of the corresponding centralized versions. See also the descriptions right below Theorem 1 and 3 for more details.
>
> ---
>
> [1] Zerka et al. Systematic review of privacy-preserving
> distributed machine learning from federated databases in health care. JCO clinical cancer informatics.
>
> [2] Muneer M Alshater. Exploring the role of artificial intelligence in enhancing academic performance: A case study of chatgpt. Available at SSRN, 2022.
>
> [3] Rahman et al. Chatgpt and academic research: a review and recommendations based on practical examples. Journal of Education, Management and Development Studies.
>
> [4] Cen et al. Fast global convergence of natural policy gradient methods with entropy regularization.
>
> [5] Agarwal et al. On the theory of policy gradient methods: Optimality, approximation, and distribution shift. The Journal of Machine Learning Research.
>
> [6] Mei et al. On the global convergence rates of softmax policy gradient methods. In Proceedings of the 37th International Conference on Machine Learning.
>
> [7] Yuan et al. Linear convergence of natural policy gradient methods with log-linear policies. arXiv preprint arXiv:2210.01400, 2022a.
>
> [8] Li et al. Softmax policy gradient methods can take exponential time to converge. In Proceedings of Thirty Fourth Conference on Learning Theory.

---

### Official Review · Reviewer_P1H3 · 2023-11-07

**Soundness:** 3 good
**Presentation:** 3 good
**Contribution:** 2 fair
**Rating:** 5
**Confidence:** 3

**Summary:**

This paper studies the federated RL problem with multi-task objectives. It develops NPG based algorithms and provides non-asymptotic convergence guarantees under exact policy evaluation.

**Strengths:**

- The paper is well written. The problem setting and formulation are clearly presented, and the ideas are well explained.

**Weaknesses:**

- From the algorithm description, it seems that the agents would need to communicate and share their information with others. This seems to be different from the motivation of using a federated algorithm, where usually agents share parameters with a central entity for aggregation. Please elaborate.
- The technical results need more explanation. Right now it is quite dry, in the sense that there is not much discussions.

**Questions:**

- From the algorithm description, it seems that the agents would need to communicate and share their information with others. This seems to be different from the motivation of using a federated algorithm, where usually agents share parameters with a central entity for aggregation. Please elaborate.
- The technical results need more explanation. Right now it is quite dry, in the sense that there is not much discussions.

---

> ### Author Response · Authors · 2023-11-20
> **Response to Reviewer P1H3**
>
> Thanks for your time reviewing our paper and for your feedback.
>
> > From the algorithm description, it seems that the agents would need to communicate and share their information with others. This seems to be different from the motivation of using a federated algorithm, where usually agents share parameters with a central entity for aggregation.
>
> We remark that we consider the more general federated setting, i.e., the decentralized setting, where the agents communication over a predetermined network topology. This encompasses the server-client setting as a special case, by simply setting the mixing matrix as a rescaled all-one matrix.
>
> We also highlight that in our setting, each agent performs local computation using its own private reward or task information, and only exchanges policy and global Q-function estimate with its neighbors, as is made clear in our algorithms and Section 1.1.
>
> > The technical results need more explanation. Right now it is quite dry, in the sense that there is not much discussions.
>
> In Section 4.1 we first give the $\mathcal O (1/T^{2/3})$ global convergence rate of exact FedNPG (Algorithm 1) in Theorem 1, and provide the $\mathcal O (1/\varepsilon^{3/2})$ iteration complexity in Corolary 1. The convergence results we provide are nearly independent of the size of the state-action space and illuminate the impacts of network size and connectivity. See the description right below Theorem 1 for a more detailed analysis of our results.
>
> Then in Themrem 2 we show that when the  Q-functions are only estimated approximately, inexact FedNPG has the same iteration complexity to reach an $\varepsilon$-approximation solution w.r.t. the value function, as long as the approximation error of the Q-functions are properly bounded. This illustrates that the convergence behavior of FedNPG is robust against inexactness of policy evaluation.
>
> Similarly, in Section 4.2 we provide the linear convergence rate of exact entropy-regularized FedNPG (Algorithm 2) in Theorem 2 and its iteration complexity in Corollary 2, and justifies the robostness of the results against the Q-function approximation error in Theorem 4.
>
> To the best of our knowledge, the proposed federated NPG methods are the first policy optimization methods for multi-task RL that achieve explicit non-asymptotic global convergence guarantees, allowing for fully decentralized communication without any need to share local reward/task information. Especially, when the network is fully connected (i.e., $\sigma=0$), the convergence results we provide for Algorithm 1 and 2 recover that of their centralized counterparts, as is stressed right below Theorem 1 and 3, and in Table 1.

---

> > ### Comment · Reviewer_P1H3 · 2023-11-22
> > **Thank you for the responses.**
> >
> > I thank the authors for the responses. However, the response about the general federated setting is not convincing. The authors should try to provide a more concrete motivating scenario. Thus, I will keep my score. Thanks.

---

### Author Response · Authors · 2023-11-20
**Experiments are added.**

Since the main concern of most of the reviewers are our lack of simulations, we add the experiment section (See Appendix E in our updated paper), where we conduct Gridworld experiments, aiming at distributedly training a globally optimal policy to follow a predetermined trajectory when each of the agent only has partial information of the trajectory. We consider the inexact Q-function approximation case and use different network topology, map size and number of agents in our experiments, and our results (see Figure 2-4) validate the fast convergence of our proposed algorithms.

If the experiments along with our response resolve your concerns, we sincerely hope you reevaluate the scores.

---

### Meta-Review · Area_Chair_wgcH · 2023-12-05

**Metareview:**

This paper studies the federated RL problem with multi-task objectives. It develops NPG based algorithms and provides non-asymptotic convergence guarantees under exact policy evaluation.

Strengths:
This is the first work for showing global convergence for federated multi-task RL using policy optimization.

Weaknesses:
1. In Reinforcement Learning literature, sample complexity is more important measure as compared to iteration complexity. The state of the art papers for centralized case achieves a sample complexity of O(1/\epsilon^2), while it is unclear what is the result in this paper. The paper needs more investigation with explicit mentions of the sample complexity results per user.
2. The results for NPG have been studied for general parametrization - where the state of the art sample complexity for global optima is O(1/\epsilon^2). The extension to general parametrization will be interesting.
3. Explanation of results, with comparison to centralized counterparts, would help.
4. A practical use case for the setup would help the paper. Mentioning "sensor networks, UAV, robot teams" does not help the application, since it is not clear why multi-task RL objective as in this paper needs to be used. Why cannot each agent use their local policies? It is not clear why we force the policy of each agent to be the same in any of these applications. Can we have some task attribute in practice as part of state? Also, many of the examples like robot teams will need collaborative MARL and not clear if they are examples here.

**Justification For Why Not Higher Score:**

The paper seems to need more investigation to be accepted at a top-tier ML venue.

**Justification For Why Not Lower Score:**

N/A

---

### Decision · Program_Chairs · 2024-01-16

Reject